# First of a line or last of a dynasty? *Parabos tigneresi* and the evolution of eurasian bovinae in the early pliocene

Leonardo Sorbelli[1,2]*, Faysal Bibi[1], Joan Madurell-Malapeira[3]*, Federica Grandi[4,5], Elena Moreno-Ribas[4,5], Oriol Oms[6], Gerard Campeny[4,5], Bruno Gómez de Soler[4,5]*

1 Museum für Naturkunde, Leibniz Institute for Evolution and Biodiversity Science, Berlin, Germany, 2 Institut Català de Paleontologia Miquel Crusafont (ICP-CERCA), Universitat Autònoma de Barcelona, Edifici ICTA-ICP, Campus de la UAB, Barcelona, Spain, 3 Earth Science Department, University of Florence, Florence, Italy, 4 Institut Català de Paleoecologia Humana i Evolució Social (IPHES-CERCA), Tarragona, Spain, 5 Departament d'Història i Història de l'Art, Universitat Rovira i Virgili (URV), Tarragona, Spain, 6 Departament de Geologia, Universitat Autònoma de Barcelona (UAB), Barcelona, Spain

* leonardo.sorbelli@mfn.berlin (LS); joan.madurellmalapeira@unifi.it (JMM); bgomez@iphes.cat (BGS)

## Abstract

The Early Pliocene maar deposits of Camp dels Ninots (Catalonia, NE Iberia) have produced an exceptionally well-preserved fossil assemblage. Among the large mammals, a large bovid stands out, represented by 14 individuals, including 8 nearly complete and partially articulated skeletons. Here we provide a comprehensive description and systematic evaluation of this material, previously referred to *Alephis tigneresi*, including an in-depth comparison with specimens assigned to *Parabos*, *Alephis*, and other large Late Miocene and Early Pliocene Eurasian and African bovids. We revise the genera *Alephis* and *Parabos*, recognizing five valid species across both groups, and refer the Camp dels Ninots bovid to *Parabos tigneresi*. We further explore the possible relationships of *Parabos* and *Alephis* within Bovinae, finding that they could either be the earliest European representatives of stem Bovini or among the last surviving members of the Miocene Tragoportacini radiation. Additionally, the exceptionally preserved skeletons of *P. tigneresi* provide paleoecological evidence of adaptation to humid, vegetated environments, confirming reconstruction of the paleoenvironment of the Camp dels Ninots maar lake, while also showing traits associated with mixed to open habitats, suggesting potentially broad habitat preferences for *P. tigneresi*.

## Introduction

Pliocene continental deposits with abundant, well-preserved and chronologically well-constrained large mammals are rare in Europe. As a result, except for a few regional and local studies, the mammalian faunas of this period remain poorly known, hampered by their scarce record [1]. This issue hindered the study of the evolution

**Data availability statement:** All relevant data are within the paper and its Supporting Information file.

**Funding:** This study was materially supported by the Caldes de Malavella Town Hall (https://caldesdemalavella.cat/index.php/en/town-hall-en-gb) which helped in the excavation of the fossil material analyzed in this work for the CN project. This study was also financially supported by the Catalan Government (Generalitat de Catalunya) through the Departament de Cultura (https://cultura.gencat.cat/ca/inici) in the form of a project award received by LS, JMM, FG, EMR, OO, GC, and BGS (CLT009/22/000043). This study was further financially supported by the Agència de Gestió d'Ajuts Universitaris i de Recerca (AGAUR) (https://agaur.gencat.cat/ca/inici) in the form of a research group grant (2021 SGR 01238 (AGAUR)) received by FG, EMR, GC, and BGS. This study was also financially supported by the Universitat Rovira i Virgili (https://www.urv.cat/en) in the form of a grant received by FG, EMR, GC, and BGS (2023-URV-01238). This study received additional financial support from the Ministry of Science and Innovation of the Spanish Government (https://www.ciencia.gob.es/en/) in the form of a project grant received by EMR, GC, and BGS (PID2024-157622NB-I00). This study received additional financial support from the Alexander von Humboldt Foundation (AvH) (https://www.humboldt-foundation.de) in the form of a Humboldt Research Fellowship grant received by LS for postdoctoral research. The funders had no undeclared role in study design, data collection and analysis, decision to publish, or preparation of the manuscript.

**Competing interests:** The authors have declared that no competing interests exist.

of bovids which are fundamental elements of the Neogene assemblages of Western Palaearctic, especially in the framework of the Neogene and Quaternary faunal turnovers and the debate surrounding the first appearance of Bovini in Europe [2, 3, 4, 5, among others]. For this reason, the discovery of rich Pliocene fossiliferous localities such as Camp dels Ninots is of the utmost importance to comprehend the evolution of the European large mammal assemblages and, in particular, of bovids.

The Camp dels Ninots maar (CN) is situated in the La Selva Depression in the town of Caldes de Malavella, Girona (NE Iberian Peninsula, 41°50′00.9" N, 2°47′57.6" W; Fig 1) and was formed during the Pliocene as part of the Catalan Volcanic Zone [6]. The maar lake originated from a phreatomagmatic explosion at the junction of a regional fault and Palaeozoic groundwater [7]. The geological formations at the base of the paleolake include Late Carboniferous-Permian granites and schists, along with pre-volcanic porous sands, clays, and gravels associated with the alluvial fan system of the Selva Basin [6]. Groundwater from the Pliocene aquifer subsequently filled the crater forming the lake which had an estimated maximum diameter of 600 meters and a depth of 72 meters [7]. This led to the gradual deposition of post-volcanic Pliocene clays, interbedded with sandstones and diatomites. Within this environment, the water column stratified, with an anoxic layer forming at the lake bed which allowed the preservation of a remarkable Pliocene fossil assemblage [8 and references therein].

Initial biochronological and magnetostratigraphic analyses dated the site to the Late Pliocene, approximately 3.3–3.1 Ma [6,7,9,10]. However, recent studies combining Ar/Ar dating, revised magnetostratigraphy, and multiproxy analyses have re-dated the site to 4.41 Ma [11]. This makes Camp dels Ninots one of the few well-dated European sites documenting the continental Early Pliocene. The importance of CN lies not only in its unusual chronology but also in the exceptional preservation of its fossil specimens [9]. The vertebrate collection, accumulated over more than 20 years of systematic excavation, includes nearly complete and articulated skeletons of freshwater fishes, amphibians, reptiles, birds, and mammals [9, 12, 13, 14, among others]. The large mammals, in particular, are represented by three taxa: *Stephanorhinus cf. jeanvireti*, *Tapirus arvernensis*, and a large bovid referred to as *Alephis tigneresi* [9,14]. Despite their relatively low taxonomic diversity, the exceptional preservation of the CN large mammal's collection makes it unique in the context of European Pliocene megafauna studies.

The genus *Alephis* Gromolard (1980 [15]) and the affine *Parabos* Arambourg and Piveteau (1929 [16]) are enigmatic bovids known from numerous but often fragmentary remains coming from the Early Pliocene (MN13-MN15) of Europe (Fig 2) [15–22].

The classification of both *Alephis* and *Parabos* has puzzled scholars, with uncertainty over whether they represent the last of the Boselaphini/Tragoportacini or the first of the Bovini in Europe [2, 15, 22, 23, 24, among others]. At present, the few finds attributable to these genera has left their diagnostic traits—and therefore their position within Bovidae—largely unresolved. The discovery at Camp dels Ninots of several nearly complete and articulated skeletons attributable to this group (Fig 3) presents a rare opportunity for their comprehensive re-evaluation.

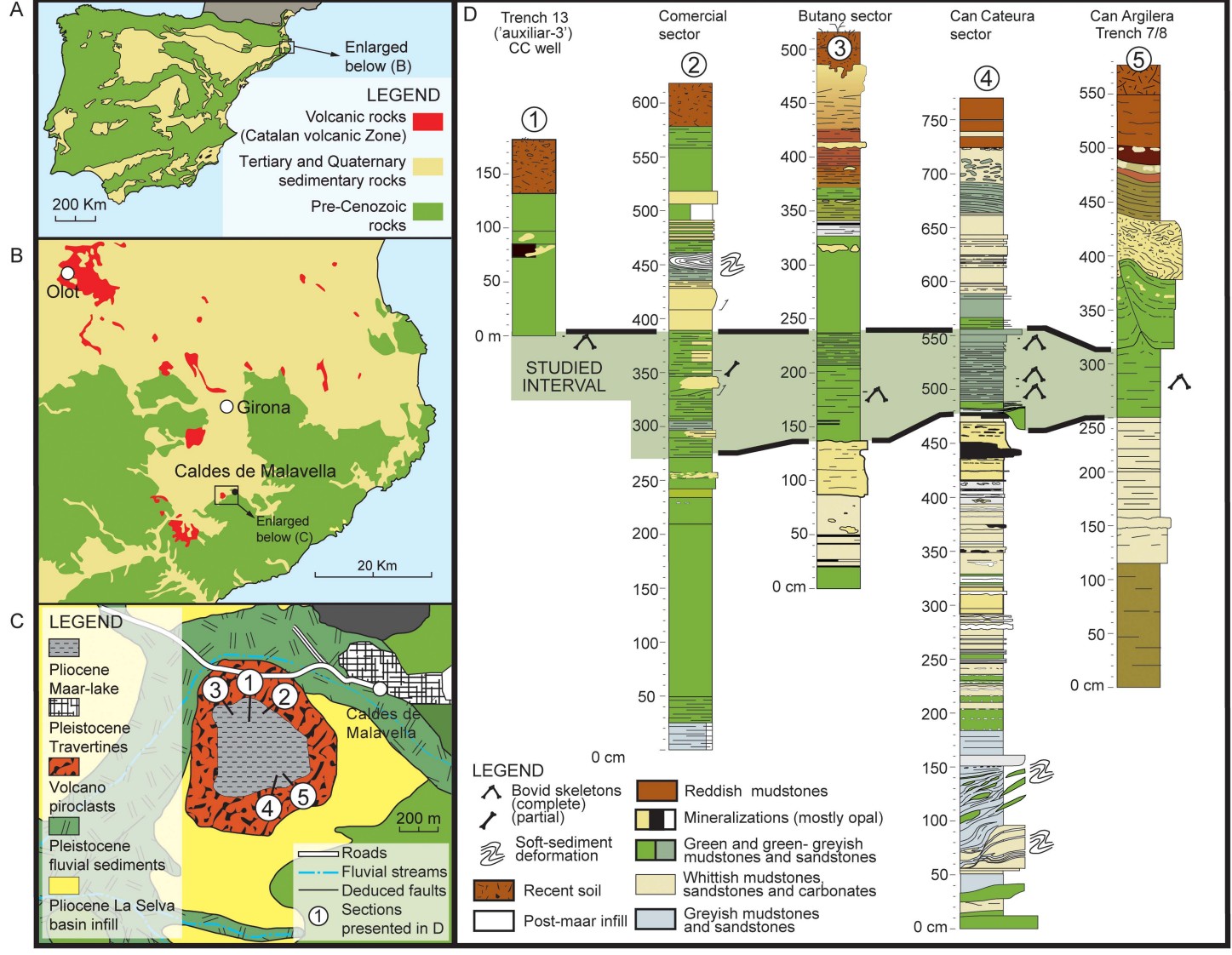

**Fig 1. Geological and stratigraphical setting of Camp dels Ninots (CN).** A–B, geological map and geographical localization of CN in the context of the North-Eastern Iberian Peninsula (A) and Caldes de Malavella (B, C); D, lithostratigraphical context of the studied Bovidae within the 4 sectors of CN locality.

## Materials and methods

Sample—The bovid record from Camp dels Ninots (CN) analyzed in this study is temporarily housed at the Institut Català de Paleoecologia Humana i Evolució Social (IPHES-CERCA, Tarragona, Spain). The sample includes 14 separate individuals in partial or complete anatomical connection. The complete list of specimens examined is provided in S1 Table. The measurements of the specimens are provided in Tables 1, 2, S2–S30.

## Morphometric data collection

Fossil measurements were taken using a digital calliper and recorded to the nearest 0.1 mm. CN specimens were tomographically scanned using a Siemens Somatom at the Hospital de Sant Pau I Santa Tecla (Tarragona, Spain).

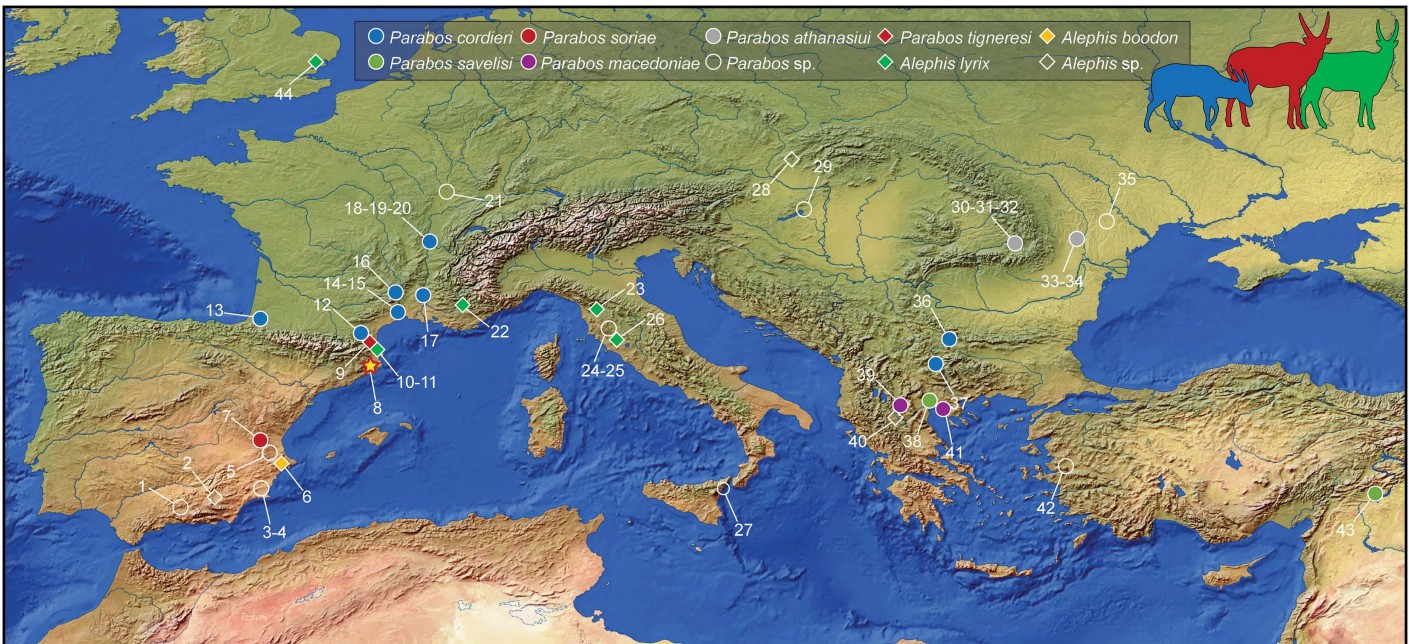

**Fig 2. Localities where *Parabos* and *Alephis* are recorded.** 1, Arenas del Rey; 2, Baza-1; 3, Puerto de la Cadena; 4, Librilla; 5, Alcalá del Júcar; 6, Alcoy-Mina; 7, Venta del Moro; 8; Camp dels Ninots; 9, Baho; 10, Perpignan; 11, Villeneuve-de-la-Raho and Serrat d'en Vaquer; 12, Ille-sur-Têt; 13, Saint Palais; 14, Sables de Montpellier; 15, Celleneuve; 16, Pont-de-Gail (Saint-Clément); 17, Saint-Laurent-des-Arbres; 18, Jassans-Riottier; 19, Tunnel de Caluire (Collonge); 20, Trèvoux; 21, Autrey-lès-Gray; 22, Puimoisson; 23, Montecarlo; 24; Casino basin; 25, Val di Pugna; 26, Velona; 27, Gravitelli; 28, Ivanovce; 29, Kisláng; 30, Capeni; 31, Varghis; 32, Iaras-1; 33, Beresti; 34, Malusteni; 35, South Bessarabia 36, Sofia basin; 37, Stamer; 38, Gephyra; 39, Ptolemais basin; 40, Milia; 41, Megalo Emvolon ("Falaise de Karabouroun"); 42, Esme-Manissa; 43, Erikdere; 44, Red Crag. Map made with Natural Earth.

Measurements on 3D digital models were obtained using MeshLab [25]. Statistical analyses were performed in PAST v.5.0.2 [26].

## Anatomical and metric standards

The anatomical terminology and linear measurements follow Cherin et al. [27] and Sorbelli et al. [28], with minor modifications (see Fig 4). Comparative data were either collected directly or sourced from the literature (full references in figures and table captions).

## Morphometric analyses

The bovid from Camp dels Ninots was compared to several extant and fossil Bovidae from Eurasia and Africa by means of the following elements:

1. Horncore size and proportion: assessed by plotting the anteroposterior diameter against the transverse diameters, both taken at the base (above the pedicle) and with the boxplot of the transverse-to-anteroposterior diameters ratio *100.

2. Lower dental row size and proportion: evaluated using molar length versus the premolar length biplot, both measured at the base and the boxplot of the ratio molar/premolar lengths *100.

3. Supraoccipital size and proportion: assessed by plotting the transverse diameter versus the anteroposterior diameter, both taken at the narrowest points, and by means of boxplot of the ratio between the two measures *100.

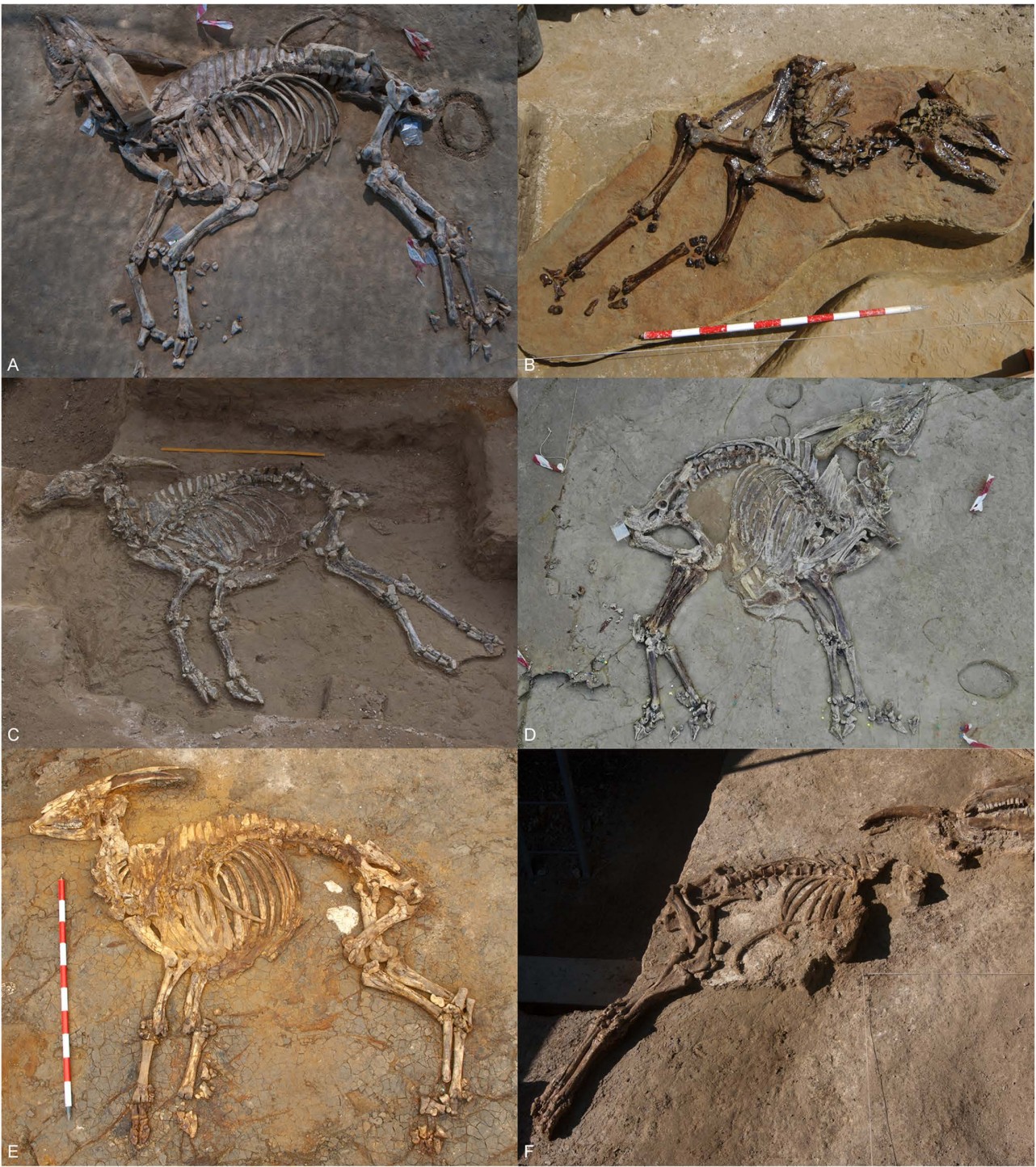

**Fig 3. Articulated skeletons of *Parabos tigneresi* from Camp dels Ninots.** A, IPHES.CN'11-B4; B, IPHES.CN'04-B1; C, IPHES.CN'05-B2; D, IPHES.CN'19-B15; E, IPHES.CN'17-B14; F, IPHES.CN'06-B3.

**Table 1. Cranial measurements (in mm) of *Parabos tigneresi* from Camp dels Ninots. Estimated measurements are in italics.**

| Measures | IPHES. CN'04- B1-1,4,5,6,10 | IPHES. CN'05- B2-1,4,5 | IPHES. CN'06- B3-1 | IPHES. CN'11- B4-1 | IPHES. CN'12- B10-1 | IPHES. CN'17- B14-1 | IPHES. CN'19- B15-1 |
|---|---|---|---|---|---|---|---|
| 1. Chord length following the maximum curve at the anterior keel (L/R) | >240.0/- | 370.0/395.0 | -/370.0 | 320.0/306.0 | 405.0/- | 410.0/420.0 | >280.0/320.0 |
| 2. Chord length following the maximum curve at the posteromedial keel (L/R) | >210.0/- | 330.0/345.0 | -/340.0 | 294.0/290.0 | 370.0/- | 350.0/380.0 | >230.0/290.0 |
| 3. Horn length from the base to the tip along the anterior keel (L/R) | >200.0/- | 366.0/380.0 | -/348.0 | 313.0/315.0 | 370.0/- | 397.0/- | >275.0/309.0 |
| 4. Transversal diameter of the horn base (L/R) | 52.0/- | 61.9/*58.0* | – | 56.0/59.0 | – | – | – |
| 5. Anteroposterior diameter of the horn base (L/R) | 64.7/- | 72.3/75.0 | – | 62.5/63.0 | 73.0/- | – | *67.5/67.9* |
| 6. Width of the supraoccipital (minimum posterior distance of temporal fossae) | – | – | – | 67.2 | – | – | – |
| 7. Length of the supraoccipital (Akrokranion–Posterior end of parietals) | – | – | – | 49.6 | – | – | – |
| 8. Greatest width of the occipital condyles | 95.1 | – | – | 84.5 | – | – | – |
| 9. Height of the foramen magnum (Basion–Opisthion) | – | – | – | 38.7 | – | – | – |
| 10. Width of the foramen magnum | – | – | – | 30.8 | – | – | – |
| 11. Greatest mastoid width (Otion–Otion) | – | – | – | 165.0 | – | – | – |
| 12. Greatest width at the base of paraoccipital processes | – | – | – | 133.0 | – | – | – |
| 13. Distance between the horncore base and the orbit | – | – | 54.4 | 68.0/>66.0 | – | 80.6 | – |
| 14. Basioccipital width at the caudal constriction | 38.2 | – | – | 51.6 | – | – | – |
| 15. Basioccipital width at the posterior tuberosities | 43.9 | *52.0* | – | 54.1 | – | – | – |
| 16. Basioccipital width at the anterior tuberosities | – | *33.6* | – | 34.5 | – | – | – |
| 17. Total skull length (Akrokranion–Prosthion) | – | – | – | >440.0 | – | 480.0 | 465.0 |
| 18. Distance between the Akrokranion and the maxillary tuberosity | – | – | – | – | – | 265.0 | – |
| 19. Distance between the Akrokranion and the oral margin of the orbit | – | – | 188.0 | 207.0 | – | 175.0 | 207.0 |
| 20. Neurocranium length (Nasion–Basion) | – | – | – | – | – | 285.0 | – |
| 21. Distance between the aboral border of the occipital condyle and the Entorbitale | – | – | 205.0 | – | – | 174.0 | – |
| 22. Nasal length | – | – | – | 207.3 | – | 207.0 | – |
| 23. Maximum nasal width | – | – | – | 43.0 | – | – | – |
| 24. Minimum lateral facial length (Prosthion–Entorbitale) | – | – | – | – | – | 275.0 | 265.0 |
| 25. Splanchnocranium length (Prosthion–Nasion) | – | – | – | 285.0 | – | 270.0 | – |
| 26. Maximum lateral facial length (Prosthion–Ectorbitale) | – | – | – | – | – | 324.0 | 313.0 |
| 27. Infraorbital foramen/Prosthion length | – | – | – | – | – | 146.0 | – |
| 28. Premaxillary length | 137.0 | – | – | 132.6 | – | 132.0 | 125.8 |
| 29. Maximum inner length of the orbit | – | – | 50.0 | – | – | 54.9 | 48.7 |
| 30. Maximum inner height of the orbit | – | – | 56.0 | – | – | 64.2 | 50.6 |

**Table 2. Tooth measurements (in mm) of *Parabos tigneresi* from Camp dels Ninots. Estimated measurements are in italics.**

| Measures | IPHES. CN'04- B1-9–19 | IPHES. CN'05- B2-1,2,3,10,11 | IPHES. CN'06- B3-1 | IPHES. CN'11- B4-1–9 | IPHES. CN'12- B6-1 | IPHES. CN'12- B9-1 | IPHES. CN'12- B10-1 | IPHES. CN'12- B13-2–16 | IPHES. CN'17- B14-1 | IPHES. CN'19- B15-1 |
|---|---|---|---|---|---|---|---|---|---|---|
| 1. Alveolar length of the upper teeth row (L/R) | 132.4 | – | 146.1 | 136.0/141.0 | – | – | – | – | 148.1/142.8 | 139.4/138.1 |
| 2. Alveolar length of the upper premolar row (L/R) | 55.7 | – | 64.4 | 57.0/58.0 | – | – | – | – | 60.7/60.2 | 60.3/60.1 |
| 3. Alveolar length of the upper molar row (L/R) | 80.4 | – | 87.7 | 86.6/88.6 | – | – | 90.3/- | – | 85.5/83.0 | 80.4/79.7 |
| 4. Length of P2 (L/R) | 19.3/19.2 | 18.9/- | -/19.6 | 18.7/- | 19.2/- | -/18.5 | 20.4/- | 21.1/19.7 | 22.4/20.9 | 17.9/19.2 |
| 5. Width of P2 (L/R) | 13.7/15.7 | 16.6 | -/>15.0 | 16.5/- | 17.2/- | -/16.7 | – | 16.8/17.8 | – | – |
| 6. Length of P3 (L/R) | 19.0/20.1 | 17.8/18.1 | -/18.6 | 18.8/19.3 | 20.1/19.5 | 18.4/18.2 | – | 19.3/20.4 | 19.5/20.3 | 16.6/19.4 |
| 7. Width of P3 (L/R) | 17.5/19.2 | 20.7/19.6 | – | 20.7/24.0 | -/13.3 | 14.0/- | – | 24.2/24.6 | – | – |
| 8. Length of P4 (L/R) | 16.8/18.8 | 17.7/18.0 | -/17.9 | 18.1/20.3 | 17.3/- | 17.3/- | – | 20.3/19.5 | 17.9/19.8 | 18.4/17.9 |
| 9. Width of P4 (L/R) | 22.2/21.3 | 22.6/22.6 | -/>17.0 | 22.6/- | 21.1/- | 21.3/- | – | 26.5/26.5 | – | – |
| 10. Length of M1 (L/R) | 22.2/- | 23.5/- | -/24.2 | 22.9/22.7 | -/22.5 | 23.3/- | 24.2/- | -/23.3 | 22.2/22.5 | 21.0/22.5 |
| 11. Width of M1 (L/R) | 25.6/- | 24.7/- | – | -/26.0 | -/22.5 | 22.3/- | – | -/28.8 | – | – |
| 12. Length of M2 (L/R) | 27.8/22.0 | 27.6/28.6 | -/28.0 | 27.7/31.1 | -/*29.0* | *28.0*/- | 30.5/- | -/28.0 | 27.2/27.0 | 24.5/26.7 |
| 13. Width of M2 (L/R) | 27.5/25.0 | 26.2/26.4 | – | -/27.6 | – | 29.0/- | – | -/28.8 | – | – |
| 14. Length of M3 (L/R) | -/30.6 | 29.4/30.5 | -/34.2 | 34.9/34.1 | -/28.6 | 29.0/- | 31.3/- | -/31.4 | 31.0/31.3 | 29.1/29.1 |
| 15. Width of M3 (L/R) | -/25.8 | 25.0/23.5 | -/>25.0 | – | -/>20.0 | 25.0 | – | -/31.4 | – | – |
| 16. Alveolar length of the lower teeth row (L/R) | – | 147.4/>150.0 | -/153.3 | 151.5/149.7 | – | – | 165.0/- | – | 155.0/152.3 | 145.8/155.4 |
| 17. Alveolar length of the lower premolar row (L/R) | – | 56.3/>65.0 | -/60.4 | 59.0/58.1 | – | – | 63.1/- | – | 55.9/54.1 | 54.2/54.4 |
| 18. Alveolar length of the lower molar row (L/R) | 90.8 | 89.5/88.6 | -/94.8 | 96.1/95.5 | – | – | 99.9/- | – | 96.1/93.4 | 89.3/90.4 |
| 19. Length of p2 (L/R) | – | 15.0/14.9 | -/14.4 | 15.6/*15.0* | – | -/21.0 | 17.7/- | 16.8/17.2 | 15.2/15.1 | 13.9/13.3 |
| 20. Width of p2 (L/R) | – | 8.8/8.9 | -/9.2 | 9.0/9.2 | – | -/16.0 | – | 11.0/11.4 | – | – |
| 21. Length of p3 (L/R) | -/18.6 | 19.8/19.7 | -/20.7 | 19.9/20.2 | – | 20.8/- | 20.0/- | 20.5/20.1 | 20.9/21.2 | 16.1/17.6 |
| 22. Width of p3 (L/R) | -/11.4 | 11.6/12.0 | -/12.2 | 12.6/12.2 | – | 12.2/- | – | 12.7/12.9 | – | – |
| 23. Length of p4 (L/R) | -/21.3 | 20.4/20.7 | -/20.4 | 21.8/21.3 | – | 23.8/- | 24.0/- | -/*21.2* | 21.4/21.8 | 18.2/20.4 |
| 24. Width of p4 (L/R) | -/13.1 | 13.3/13.3 | – | 14.3/13.2 | – | 12.7/- | | 15.1/14.1 | – | – |
| 25. Length of m1 (L/R) | -/22.4 | 21.2/21.5 | -/24.9 | 24.2/24.6 | – | 28.7/- | 23.8/- | -/25.1 | 23.4/25.5 | 22.6/23.2 |
| 26. Width of m1 (L/R) | -/15.2 | 15.1/14.7 | – | 15.3/16.2 | – | 21.7/- | – | -/17.6 | – | – |
| 27. Length of m2 (L/R) | -/24.5 | 26.0/25.4 | -/26.0 | 27.1/28.0 | – | – | 29.3/- | 25.9/27.6 | 27.8/28.7 | 25.9/26.3 |
| 28. Width of m2 (L/R) | -/18.0 | 16.4/17.8 | – | – | – | – | – | 19.2/19.3 | – | – |
| 29. Length of m3 (L/R) | -/38.7 | 39.2/38.8 | -/44.1 | 43.1/43.6 | – | -/37.6 | 40.0/- | 42.0/41.7 | 41.2/43.1 | 38.2/40.3 |
| 30. Width of m3 (L/R) | -/17.3 | 18.2/17.6 | – | – | – | -/16.6 | – | 19.3/20.3 | – | – |
| 31. Length of i1 | 14.5 | – | -/12.2 | – | – | – | 15.4/15 | – | 13.8/11.8 | – |
| 32. Width of i1 | 9.4 | – | -/7.6 | – | – | – | 10.0/9.9 | – | – | – |
| 33. Length of i2 | 12.3 | – | -/8.9 | – | – | – | 13/12.3 | – | -/11.6 | – |
| 34. Width of i2 | 7.7 | – | -/6.9 | – | – | – | 9.0/8.3 | – | – | – |
| 35. Length of i3 | 8.1/9.1 | – | -/9.0 | – | – | – | 10.5/11.4 | – | 7.7/7.2 | – |
| 36. Width of i3 | 7.0/7.7 | – | -/7.1 | – | – | – | 8.2/8.0 | – | – | – |
| 37. Height of the mandible at m3 | -/62.5 | 62.5/- | – | – | – | – | – | – | 60.6/*59.2* | 60.0/65.3 |
| 38. Height of the mandible at m1 | -/40.7 | 47.3/47.1 | – | – | – | – | – | – | 41.6/*49.4* | 43.8/44.2 |
| 39. Length of the mandibular corpus | – | – | – | – | – | – | – | – | >340.0 | 382.0 |

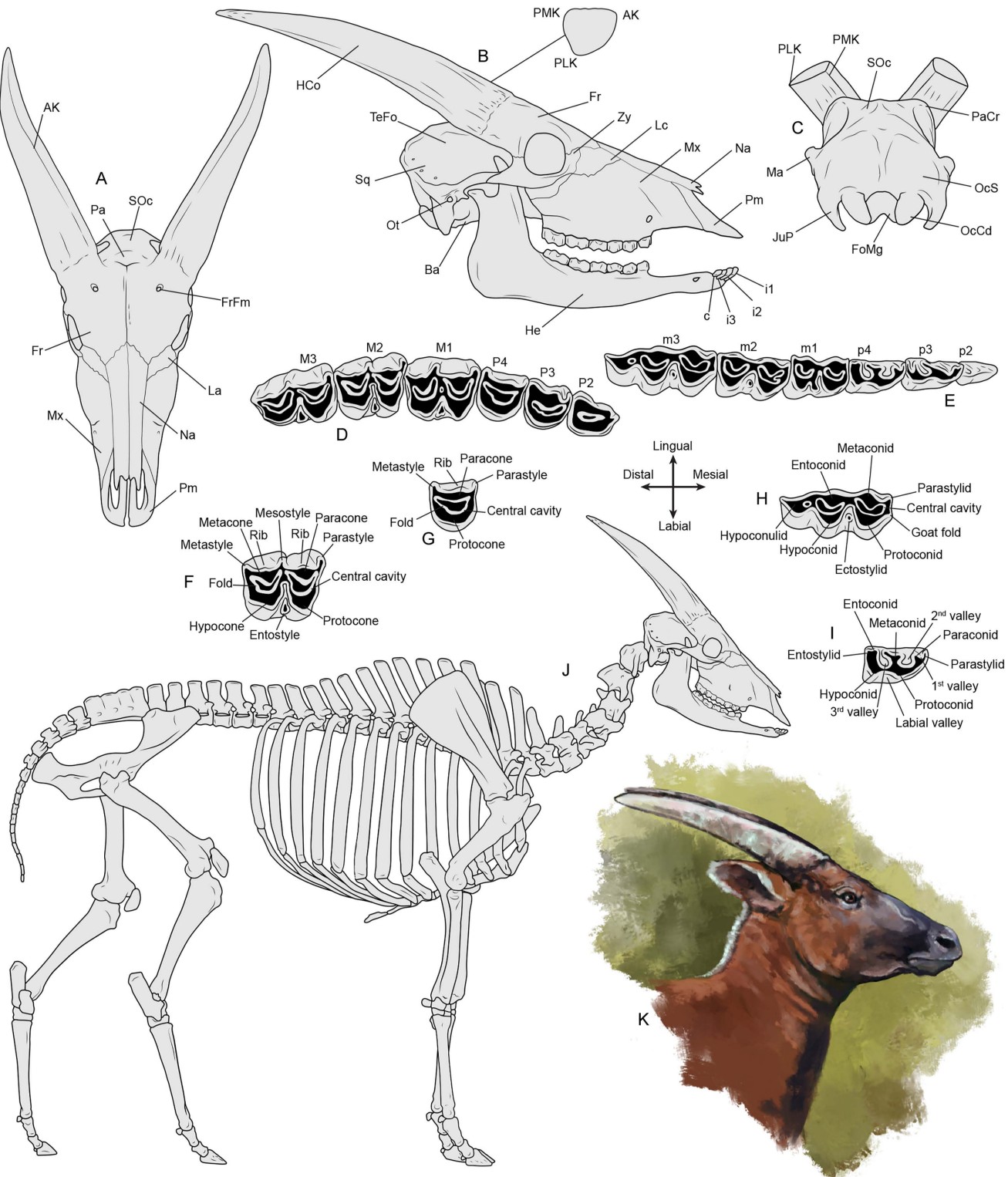

**Fig 4. Osteology and in-life reconstruction of *Parabos tigneresi* based on the specimens from Camp dels Ninots.** Cranium (A–C), dentition (D–I), skeleton (J) and life restoration of an adult male with colours based on the extant bongo (*Tragelaphus euryceros*) (K), with the position of the most important anatomical elements mentioned in the text. Drawings not to scale. Legend: AK, anterior keel; Ba, basioccipital; c, canine; FoMg, foramen

magnum; Fr, frontals; FrFm, frontal foramina; HCo, horncore; He, hemimandible; i1, first incisor; i2, second incisor; i3, third incisor; JuP, jugal process; Lc, lacrimal; M1, first upper molar; m1, first lower molar; M2, second upper molar; m2, second lower molar; M3, third upper molar; m3, third lower molar; Ma, mastoid; Mx, maxilla; Na, nasals; OcS, occipital squama; Ot, otion; P2, second upper premolar; p2, second lower premolar; P3, third upper premolar; p3, third lower premolar; P4, fourth upper premolar; p4, fourth lower premolar; Pa, parietal; PaCr, parietal crest; PLK, posterolateral keel; Pm, premaxilla; PMK, posteromedial keel; Soc, supraoccipital; Sq, squamosal; TeFo, temporal fossa; Zy, zygomatic. Artwork by LS.

4. Basioccipital tuberosities size and proportion: analysed via a bivariate scatter plot comparing the posterior tuberosities transverse diameter (=width) against the anterior tuberosity width and with the boxplot of these two *100.

5. Teeth (P4, M1, M3, p3, p4, m3) size and proportion: Analysed by plotting tooth length against width, both taken at the base.

6. Limb bones proportions: examined using log-transformed stylopodium length (humerus and femur) versus autopodium (radius/tibia) and metapodial (metacarpal and metatarsal) lengths. Data points and scatter clouds were generated based on values from individual specimens, with each point representing the two corresponding bone lengths of a single limb. This approach was not applicable to *Parabos cordieri*, *Alephis lyrix*, *Leptobos etruscus*, and *Bison* spp., as it was not possible to assign specific bones to individual specimens. In these cases, the points are represented by the minimum, maximum and mean values for each element.

## Hypsodonty index

The hypsodonty degree was calculated following Fortelius et al. [29] in which the height of the tooth taken at its highest point is divided by its basal length. The sample includes only unworn or early wear stage teeth.

## Body mass estimation

Body mass was estimated using equations from Scott [30] and Janis [31], based on average extant species masses and 11 dental and postcranial variables (S31 Table). Body mass was estimated for nine of the most complete CN individuals and for other taxa from direct observation or using measurements from the literature (S32 Table). Each individual's body mass was calculated as the average of the mean values obtained separately from all available variables, including both left and right elements where applicable.

## Taxonomic considerations

The fossil relatives of extant boselaphines (i.e., *Boselaphus* and *Tetracerus*) are largely unknown and most, if not the totality, of the Miocene boselaphine-like taxa such as *Miotragocerus*, *Tragoportax* and allies (e.g., *Austroportax*, *Strepsiportax*, *Protragocerus*, *Helicoportax,* among others), which have been historically referred to Boselaphini, might be part of a distinct clade. Already in the 50s several authors proposed to unify these forms under the name of Tragocerina/Tragocerini, which, however, cannot be considered valid due to nomenclatural issues [32, 33, 34). Bibi et al. [5] proposed the term Tragoportacini to define the *Miotragocerus-Tragoportax* complex. This nomenclature had received alternate use in literature being formally accepted [35–37] or not [38–40]. Although, at the current state of the art, the clade is not well defined and lacks a proper diagnosis we do recognize that the *Miotragocerus-Tragoportax* complex differs substantially from Boselaphini enough to be considered a separate group. Therefore, in this paper, we adopt the nomenclature proposed by Bibi et al. [5] due to its relatively wide use in literature and to avoid informal and verbose definition such as *Miotragocerus-Tragoportax* complex. Other taxonomical nomenclature that requires specification are hereby listed: nomenclature referred to *Bison* and *Leptobos* follows Azzarà et al. [41] and Sorbelli et al. [28], nomenclature referred to *Miotragocerus* and *Tragoportax* follows Kostopoulos [22,42], nomenclature referred to *Ugandax* follows Gentry [43], nomenclature referred to *Proamphibos* follows Nishioka et al. [44].

**Institutional abbreviations**

IPHES, Institut Català de Paleoecologia Humana i Evolució Social, Tarragona (Spain); LGPUT, Museum of Geology, Palaeontology and Palaeoanthropology of the Aristotle University of Thessaloniki (Greece); AMNH, American Museum of Natural History, New York (USA); RIN, Rajabhat Institute of Nakhon Ratchasima, Nakhon Ratchasima (Thailand); UFR FSL, Faculté des Sciences, Université Claude Bernard Lyon 1, Lyon (France); MNHNP, Museum national d'Histoire naturele, Paris (France); NMBS, Naturhistorisches Museum Basel, Basel (Switzerland); IGF, Sezione di Paleontologia del Museo di Storia Naturale, Università di Firenze, Florence (Italy); IGPM, Institute für Geologie und Paläontologie Universität Münster, Münster (Germany); GSI, Geological Survey of India, Calcutta (India); MFN, Museum für Naturkunde, Berlin (Germany); ICP IPS, Institut Català de Paleontologia Miquel Crusafont, Sabadell (Spain).

## Results

### Systematic palaeontology

Unranked hierarchy

Artiodactyla Owen 1848

Ruminantia Linnaeus 1758

Bovidae Gray, 1821

Bovinae Gray, 1821

*Parabos* Arambourg and Piveteau, 1929a [16]

Selected synonymy list

*Antilope* de Christol, 1832 [45]

*Antilope* Gervais, 1848–1852 [46]

in pars. *Antilope* Gervais, 1867-1869 [47]

in pars. *Palaeoryx* Depéret, 1885 [48]

*Parabos* nov. gen, Arambourg and Piveteau 1929 [16]

in pars. *Parabos*, Pilgrim, 1939 [18]

in pars. *Parabos*, Merla 1949 [49]

in pars. *Parabos*, Samson and Radulescu, 1963 [50]

in pars. *Parabos*, Gromolard, 1980 [15]

in pars. *Parabos*, Gromolard, 1981 [20]

in pars. *Parabos*, Gromolard and Guérin, 1980 [21]

in pars. *Parabos*, Morales, 1984 [51]

1991: *Parabos*, Michaux et al., 1991 [52]

2006: *Parabos*, Montoya et al., 2006 [53]

2009: *Parabos*, Bibi, 2009 [24]

2017: *Parabos*, Crégut-Bonnoure and Tsoukala 2017 [54]

2022: *Parabos*, Kostopoulos, 2022 [22]

### Diagnosis

Medium to large sized Bovinae diagnosed by the following combination of characters: horns inserted above the posterior edge of the orbits, well-separated from each other, emerging parallel or slightly diverging, straight or medially curved, with weak heteronymous torsion. Horns with very weak or no mediolateral compression and faint or no surface grooving. Basal cross-section triangular, with three keels, or rounded, without evident keels. Keels, when present, with various degrees of development but anterior keel always stronger than the other two. Anterior keel never stepped. Frontal sinuses extended into the pedicle and slightly into the horn core bases. Pedicle sinus not strutted. Neurocranium shortened with parietal crests elevated above the parietals, supraoccipital elongated and occipital squama relatively wide. Supraoccipital flexed toward the occipital so the angle created by the supraoccipital and the parietal surfaces is obtuse. Skull roof behind the horncores depressed with smooth or weakly sculpted surface. Supraorbital foramina located at the base of the pedicles, flush on the surface without a surrounding fossa. Ethmoidal fenestra and lachrymal fossa absent at least in one species but unknown in the others. Basioccipital triangular with posterior tuberosities almost double the width of the anterior tuberosities. Premaxillae in contact with the nasals. Teeth large and mesodont with wrinkled buccal surface and scarce or no cement. Presence of incipient basal pillars on molars. Premolar series length reduced relatively to the molars (premolar length ca. 60% of the molar length). Fourth lower premolar with prominent metaconid, expanded both lingually and mesiodistally, often in contact with the paraconid and closing the anterior lingual valley. Goat fold often presents in lower molars, especially in the third molar. Limb bones elongated but showing a slight reduction in the length of the metapodials relative to the stylopodium (emended from [15]).

### Differential diagnosis

*Parabos* differs from *Alephis* in the straighter, less mediolaterally compressed and less twisted horncores, the more developed posterior keels of the horns so their base has a clearer triangular cross-section (this last character is not valid for *Parabos savelisi*), the larger supraoccipital and stronger parietal crests. It differs from crown Bovini in the same characters mentioned for *Alephis* as well as in a longer neurocranium with horncores positioned more frontward, supraorbital foramina placed more posteriorly, not located within elongated grooves, and less hypsodont dentition with smaller basal pillars. It differs from *Miotragocerus*, *Tragoportax* and allies (i.e., Tragoportacini *sensu* [5]) in larger size, horncores lacking strong mediolateral compression, weaker frontal keel lacking steps, shorter and wider neurocranium, smaller supraoccipital, weaker temporal crests, lack of sculpting on the parietals and posterior frontals, more reduced premolars relative to the molars, strengthening of the molar pillars and ribs. It differs from extant Boselaphini in the longer and more posteriorly oriented horns, lack of sculpting on the parietals and posterior frontals, shorter and taller neurocranium, and supraoccipital more strongly flexed against the skull roof.

### Type species

*Parabos cordieri* (de Christol, 1832 [45]), as designated by Gromolard (1980 [15]).
   **Included species.** *Parabos soriae* Morales 1984 [51], *Parabos savelisi* Crégut-Bonnoure and Tsoukala 2017 [54], *Parabos tigneresi* (Michaux et al., 1991 [52]).
   **Stratigraphic and Geographic distribution.** Early Pliocene (MN13–MN16) of Europe.
   Species: *Parabos tigneresi* (Michaux et al., 1991 [52] comb. nov.)
   Selected synonymy list

*Alephis tigneresi* Michaux et al., 1991 [52]

*Alephis tigneresi* Montoya et al., 2006 [53]

*Alephis tigneresi* Bibi 2009 [24]

*Alephis tigneresi* Fejfar et al., 2012 [55]

*Alephis tigneresi* Crégut-Bonnoure and Tsoukala 2017 [54]

*Alephis tigneresi* Kostopoulos, 2022 [22]

## Diagnosis

The following combination of characters distinguishes this species from all known species of *Parabos* and *Alephis*. Horns weakly diverging, weakly curving posteriorly and medially, with faint heteronymous torsion; more so than in any other *Parabos* species but less than in *Alephis*. Horn core basal cross-section sub-triangular, slightly compressed mediolaterally, with three keels evident. Prominent anteromedial keel and weaker posterolateral and posteromedial keels. Medial and posterior surfaces of the horn flattened. Medial surface covered by shallow grooves (absent in *P. cordieri* and *P. savelisi* and deeper in *A. lyrix*). Frontals and pedicles pneumatized with sinuses reaching the base of the horns. Braincase relatively short and high with relatively small occipital squama. Large supraoccipital, shorter than in any other *Parabos* species but longer than in *Alephis*. Ethmoidal fenestra absent. Dentition very large. Pillars and ribs relatively well-developed, similar to the condition shown in *Alephis lyrix*. Lower p4 as in *P. cordieri* and Tragoportacini in which the metaconid is inflated, almost reaching the paraconid and often closing the anterior valley. Cement present but thin. Size larger than any other *Parabos* species and comparable with large-sized extant Bovini (ca. 500 kg). Limb bones more massive than in Boselaphini, Tragoportacini and any other *Parabos* species but slenderer than crown Bovini (emended from [52]).

## Type locality

Baho Section (Perpignan Basin, Roussillon, France), Pliocene.
Stratigraphic and Geographic distribution—Early Pliocene (MN14) of western Europe (France, Spain).

## Referred specimens

The Camp dels Ninots bovid sample studied in this work includes 1,779 specimens for a total of a minimum of 14 individuals. The sample is figured in Figs 3; 5–16. Measurements are provided in Tables 1, 2, S2–S30. The complete list of the specimens divided by individuals is provided in S1 Table.

## Description

**Cranium.** The complete horns of 11 individuals are preserved (Figs 5–8), although the lateral compression severely deformed their bases except for the specimens IPHES.CN'04-B1-1, IPHES.CN'05-B2-1 and IPHES.CN'11-B4-1 which do not show signs of strong deformation (Figs 5; 8). The horns are long, robust, and generally large. The size and the morphology of the horncores is roughly homogeneous within the sample with the noticeable specimens IPHES. CN'04-B1-1 and IPHES.CN'19-B15-1 which feature the smallest and most straight horncores (Figs 6A–B; 8I–M; Table 1). The cores arise just behind the posterior edge of the orbit. They emerge slightly diverging, then curve posteriorly and medially at the tip. There is a very faint heteronymous torsion (i.e., the right horn has an anticlockwise torsion from the base up) in the distalmost portion of the core (Figs 5A; 8E, F). The section is sub-triangular, mediolaterally compressed, with flattened medial (subparallel to the sagittal axis) and posterolateral surfaces and convex anterolateral surface (Fig 7). The section at the tip is subcircular. The medial and anterolateral surfaces are about of the same size, both larger than

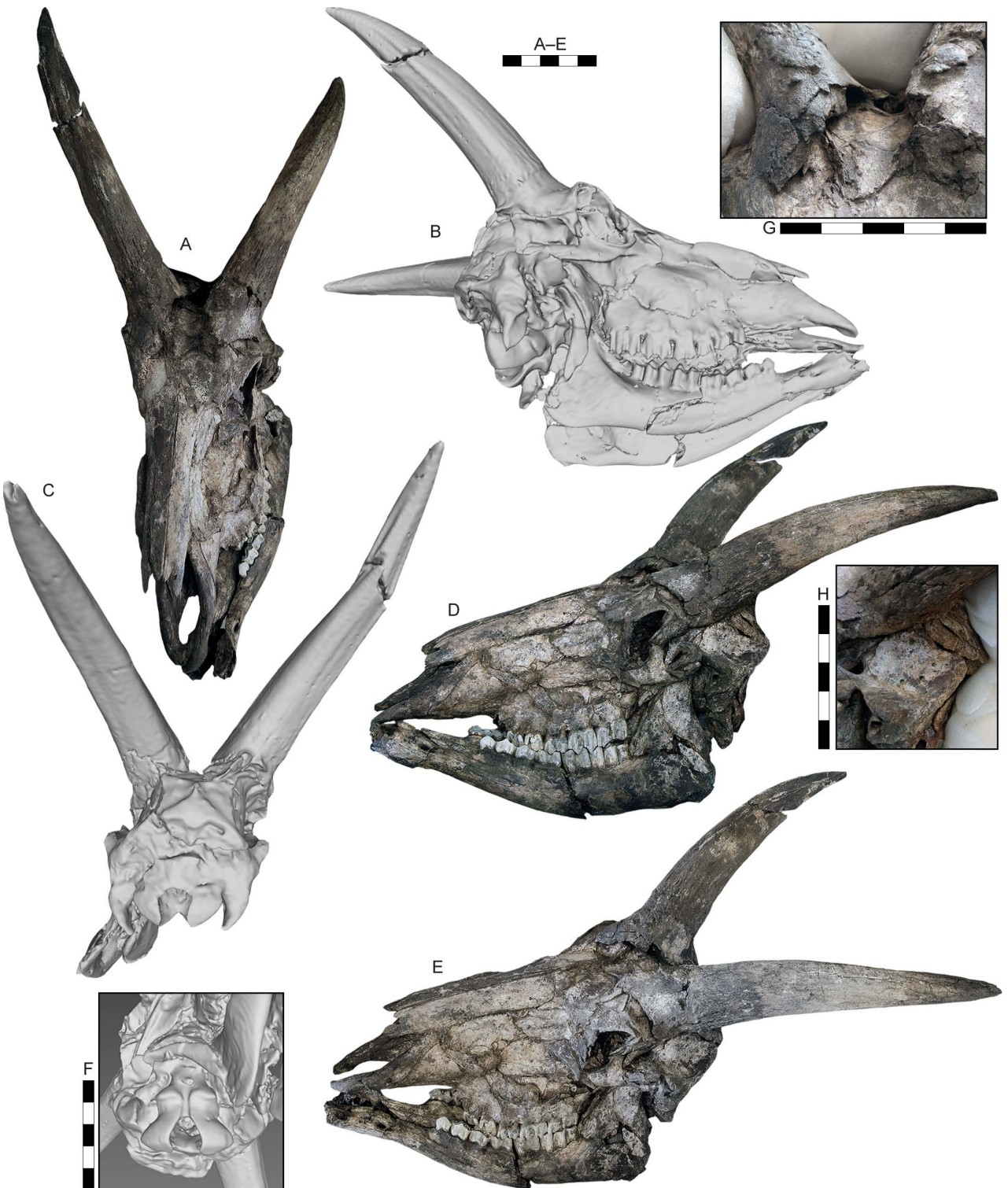

**Fig 5. Skull of of _Parabos tigneresi_.** IPHES.CN'11-B4-1 in anterodorsal (A), left lateral (B), posterior (C), left lateral (D), and dorsolateral left (E) views and close-ups on the basioccipital (F), horncore bases and frontals (G) and temporal fossa (H). Scale bar: 100 mm.

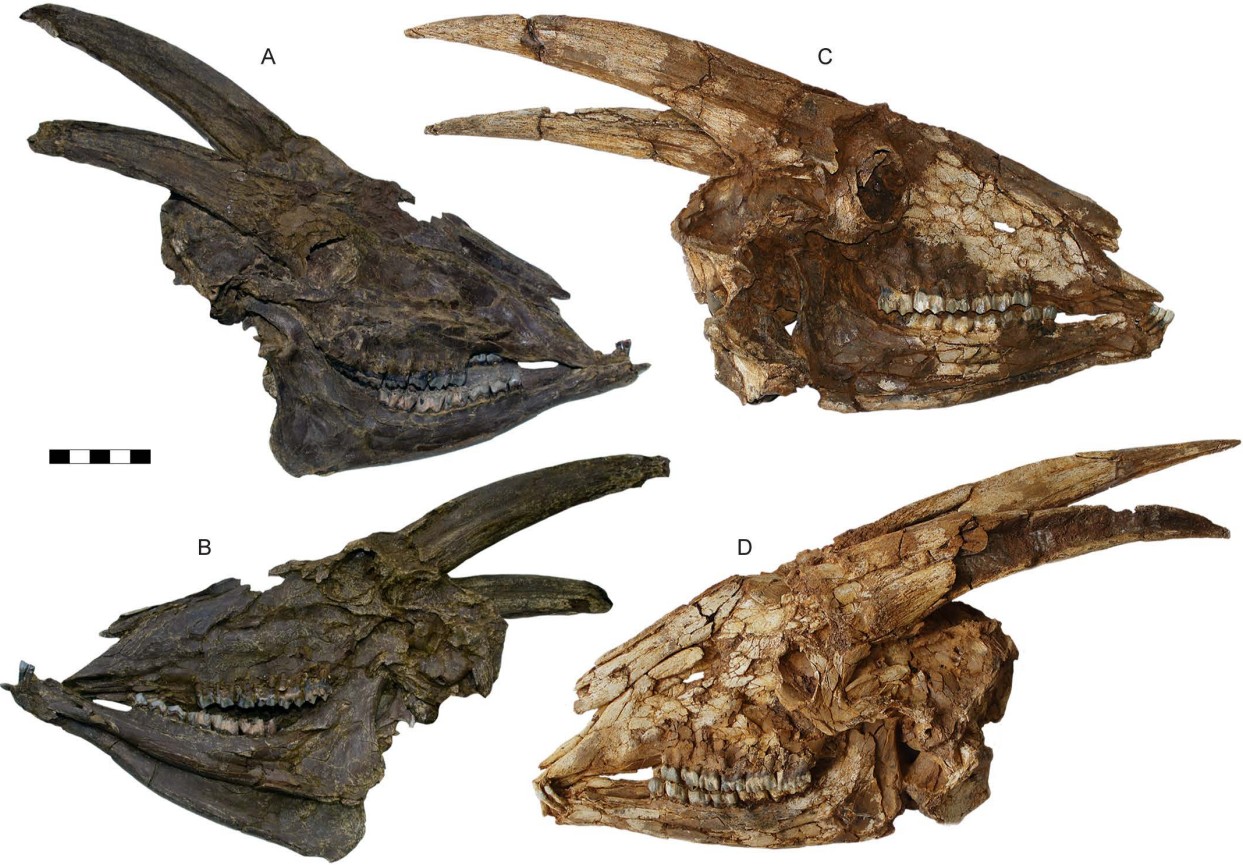

**Fig 6. Skulls of *Parabos tigneresi*.** A–B, IPHES.CN'19-B15-1 in right (A) and left lateral (B) views; C–D, skull IPHES.CN'17-B14-1 in right (C) and left lateral (D) views. Scale bar: 100 mm.

the posterolateral one. There are three keels which run along the core for its entire length and disappear at the tip. The anterior keel is the strongest, represented by a sharp edge whereas the posteromedial and posterolateral keels are relatively blunt. All the keels are less evident at the basalmost portion of the horn (i.e., at the contact with the pedicle). The lateral outline of the anterior keel is gently curving backward without evident stepped/abrupt transitions (Figs 5–8). Several shallow grooves are present on the medial and posterolateral surfaces (Fig 6A, C). A deeper groove is present on the posterolateral face, laterally bordering the posteromedial keel (Fig 8J). This furrow is more or less developed depending on the specimen (e.g., more in IPHES.CN'04-B1-1 and less in IPHES.CN'05-B2-1). The pedicles are relatively long and pneumatized. The sinuses extend into the horns but do not continue over the first 1/5 of the core length (Fig 8N–Q). The basal surface of the horn is covered by a rugose surface and irregular bone-growths, most likely due to the advanced ontogenetic stage of most of the individuals (e.g., IPHES.CN'11-B4–1) (Fig 5G).

The neurocraniums are all severely compressed although, in the specimens IPHES.CN'11-B4-1, IPHES.CN'17-B14-1 and IPHES.CN'19-B15-1, complete (Figs 5; 6). The post-cornual and occipital area are damaged and/or deformed in all the individuals. Despite that, the skull IPHES.CN'11-B4-1 still partially retains its original morphology. Most of the following description is based on this specimen (Fig 5). The braincase is relatively short. The occipital squama is low and wide with trapezoidal shape (i.e., wide base, narrow top, Fig 5C). The post-cornual portion of the skull is relatively short, featuring a protruding supraoccipital constriction (=chignon sensu [56]; = intertemporal bridge sensu [57]). This element is squared,

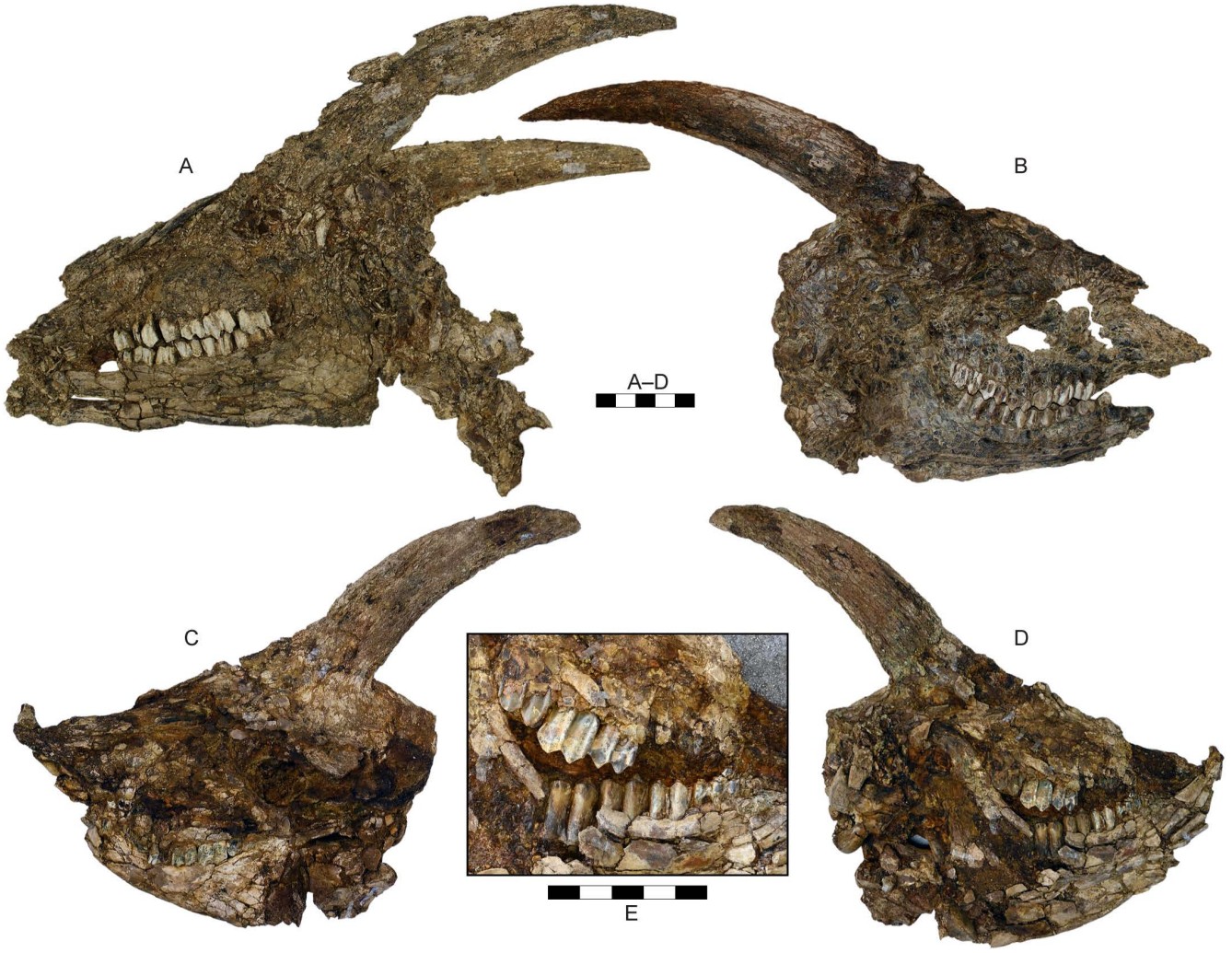

**Fig 7. Skulls of *Parabos tigneresi*.** A, skull IPHES.CN'12-B10-1 in left lateral view; B, skull IPHES.CN'06-B3-1 in right lateral view; C–D, IPHES.CN'12-B7-1 in left (C) and right (D) lateral view. Scale bar: 100 mm.

relatively short and wide (i.e., transversal diameter>anteroposterior diameter; Table 1). It lies partially deflected toward the occipital plane, so the angle formed by the two planes is slightly obtuse (Figs 5D; 6C). The frontoparietal surface it is not sculptured although a very faint rugosity is present in the posterior portion of the parietals. The temporal fossae are shallow (i.e., the parietal crests do not overhang the fossae) and high, marked by multiple large foramina running along the contact between the squamosal and the nuchal crest (Fig 5H). Both parietal and nuchal crests are pronounced and converge dorsodistally, creating the above-mentioned constriction at the supraoccipital level. The parietal crests (=upper temporal crests) are elevated above the parietals. The mastoid area is enlarged, greatly expanded laterally and posteriorly. The basioccipital is triangular, characterized by small, elongated anterior tuberosities and large, bulging posterior tuberosities. There is a narrow but well-defined sagittal groove separating the left and right posterior tuberosities, starting at the ventral border of the foramen magnum and ending at the anterior tuberosities. The basioccipital is flexed forming an acute angle with the occipital plane (ca. 50–60°). The occipital crest that runs dorsoventrally in the middle of the squama is weak (Fig 5C). The occipital condyles are large but laterally compressed. Lateral to the condyles, separated by deep

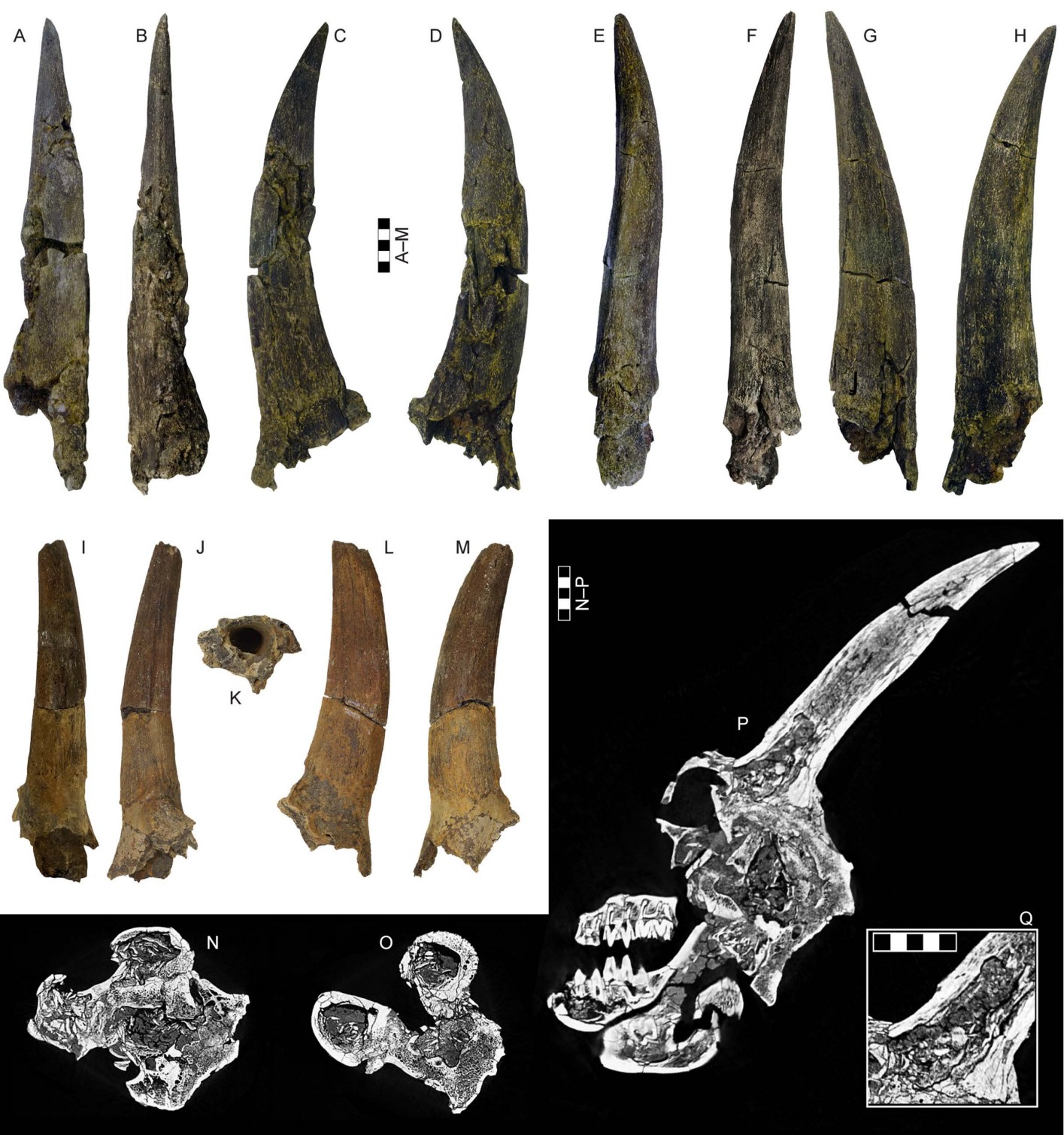

**Fig 8. Horncores and neurocranium of *Parabos tigneresi*.** A–D, right horncore IPHES.CN'05-B2-4 in anterior (A), posterior (B), medial (C) and lateral (D) views; E–H, left horncore IPHES.CN'05-B2-5 in anterior (E), posterior (F), medial (G) and lateral (H) views; I–M, left horncore IPHES.CN'04-B1-4 in anterior (I), posterior (J), proximal (K), medial (L), and lateral (M) views; N–Q, CT images IPHES.CN'11-B4-1 with transverse cross-sections at the pedicles (N) and horncore bases (O), and sagittal cross-section through the right horncore (P) and close-up of the horncore sinus in the same view (Q). Scale bars: 50 mm.

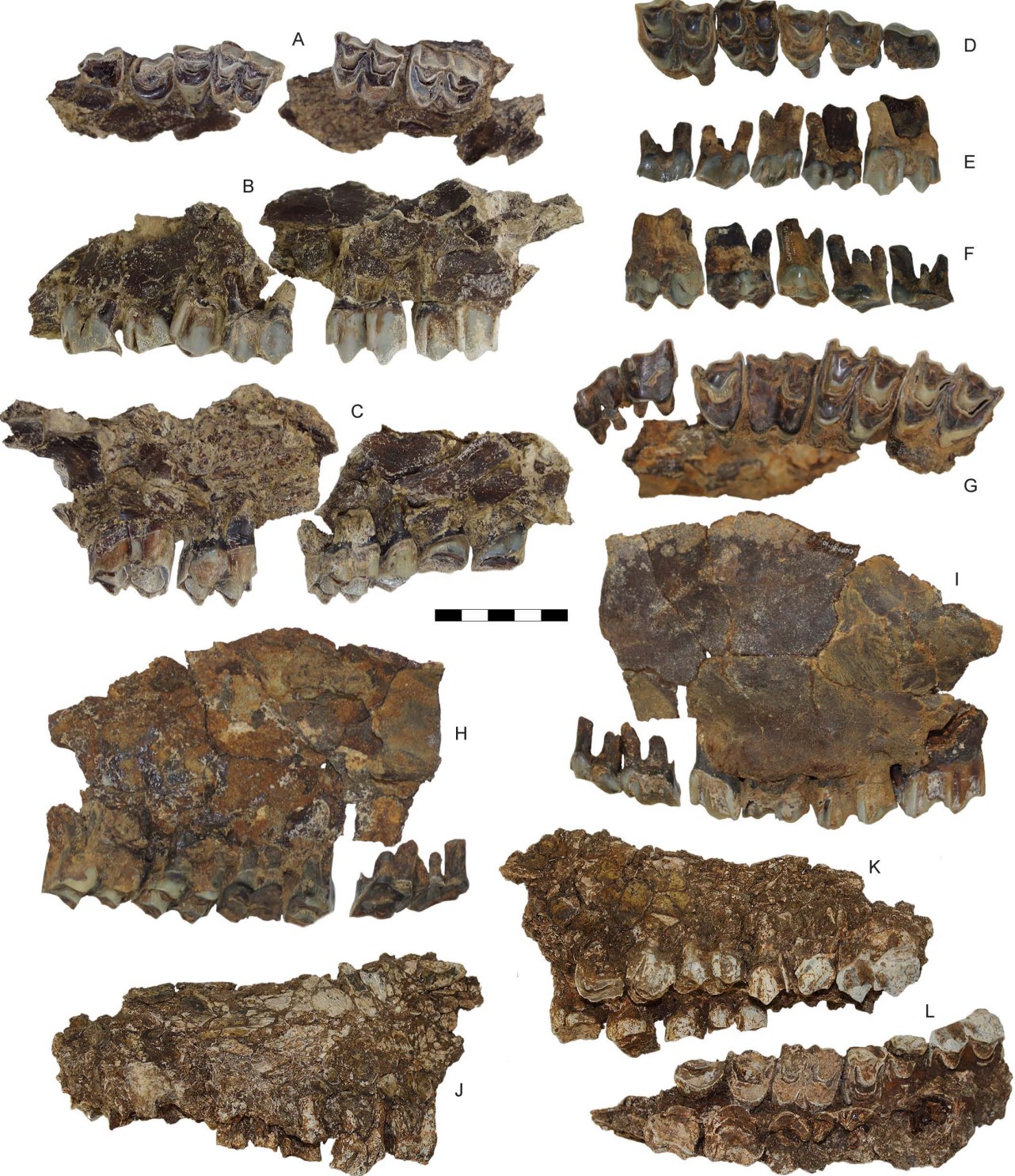

**Fig 9. Upper dentition of *Parabos tigneresi*.** A–C, left maxilla with P2-M3 IPHES.CN'05-B2-1 in occlusal (A), labial (B) and lingual (C) views; D–F, right P2-M3 IPHES.CN'04-B1-14–17,19, in occlusal (D), labial (E) and lingual (F) views; G–I, left maxilla with P2-M3 IPHES.CN'04-B1-10 in in occlusal (G), lingual (H) and labial (I) views; J–L, left and right maxillae with left P3-M3, R P2-M2 CN'12-B9-1 in left (J), right lateral (K) and occlusal (L) views. Scale bar: 50 mm.

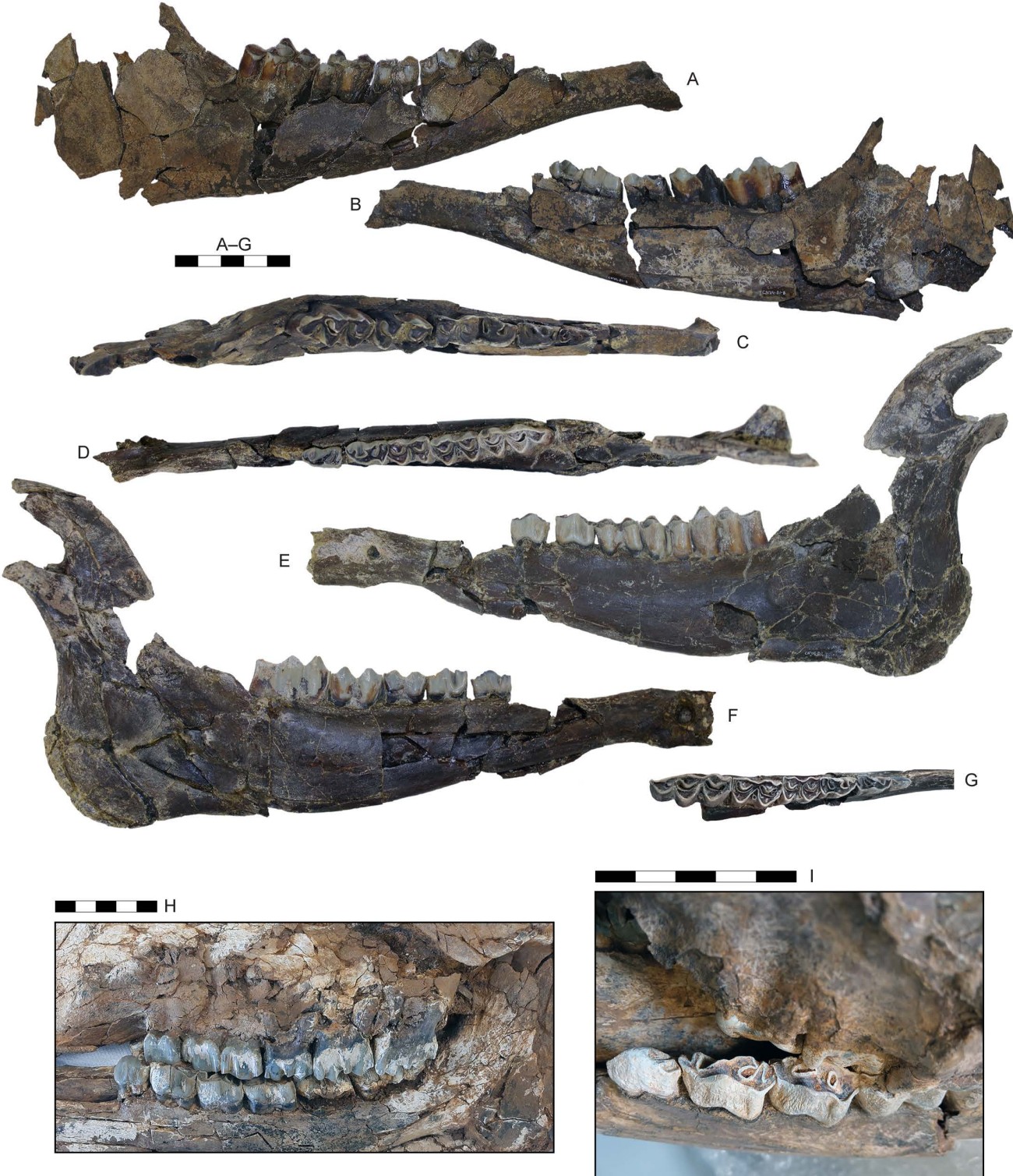

**Fig 10. Lower dentition of *Parabos tigneresi*.** A–C, right hemimandible with p3-m3, IPHES.CN'04-B1-8 in lingual (A), labial (B) and occlusal (C) views; D–F, left hemimandible with p3-m3, IPHES.CN'05-B2-3, in occlusal (D), lingual (E) and labial (F) views; G, right hemimandible with p2-m3, IPHES.CN'05-B2-2, in occlusal view; H, upper and lower left tooth rows of IPHES.CN'17-B14-1 in labial view; I, p2–p4 of IPHES.CN'11-B4-1 in occlusolabial view. Scale bars: 50 mm.

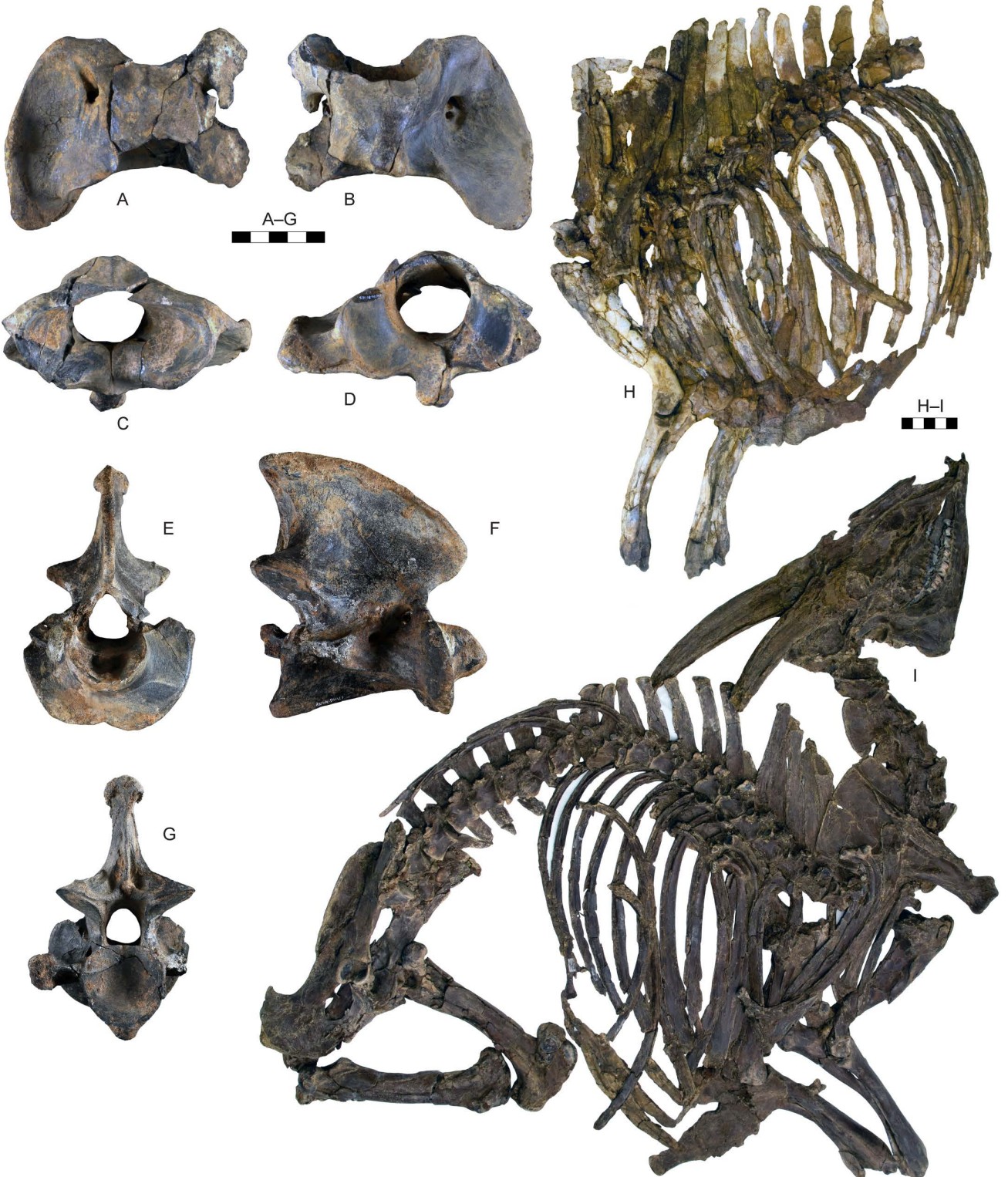

**Fig 11. Vertebrae and partial skeletons of *Parabos tigneresi*.** A–D, atlas IPHES.CN'04-B1-125 in dorsal (A), ventral (B), anterior (C) and posterior (D) views; axis IPHES.CN'04-B1-127 in anterior (E), lateral (F) and posterior (G) views; H, partial axial (rib cage, trunk vertebrae) and appendicular (scapula, humerus, radius) skeletal elements of CN'17-B14 in left lateral view; I, partial axial (skull, vertebral column, rib cage) and appendicular (scapula, humerus, radius and ulna, pelvis and femur) skeletal elements of CN'19-B15 in right lateral view. Scale bars: 50 mm (A–G), 100 mm (H–I).

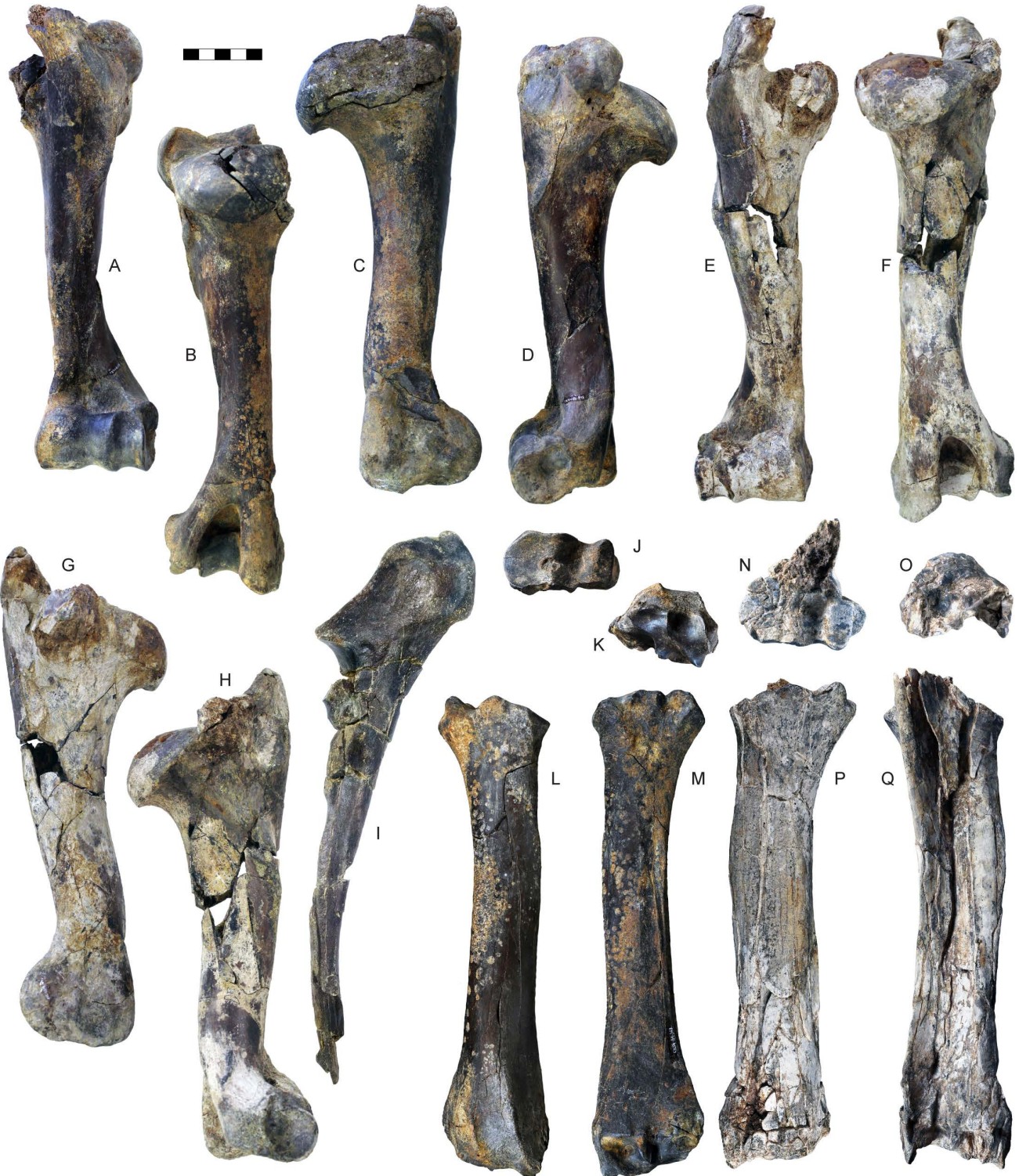

**Fig 12. Humerus, radius and ulna of *Parabos tigneresi*.** A–D, left humerus IPHES.CN'04-B1-345 in anterior (A), posterior (B), medial (C) and lateral (D) views; E–H, right humerus IPHES.CN'11-B4-300 in anterior (E), posterior (F), medial (G) and lateral (H) views; I, left ulna IPHES.CN'05-B2-337 in lateral view; J–M, right radius IPHES.CN'04-B1-303 in proximal (J), distal (K), anterior (L) and posterior (M) views; N–Q, left radius IPHES.CN'11-B4-303 in proximal (N), distal (O), anterior (P) and posterior (Q) views. Scale bars: 50 mm.

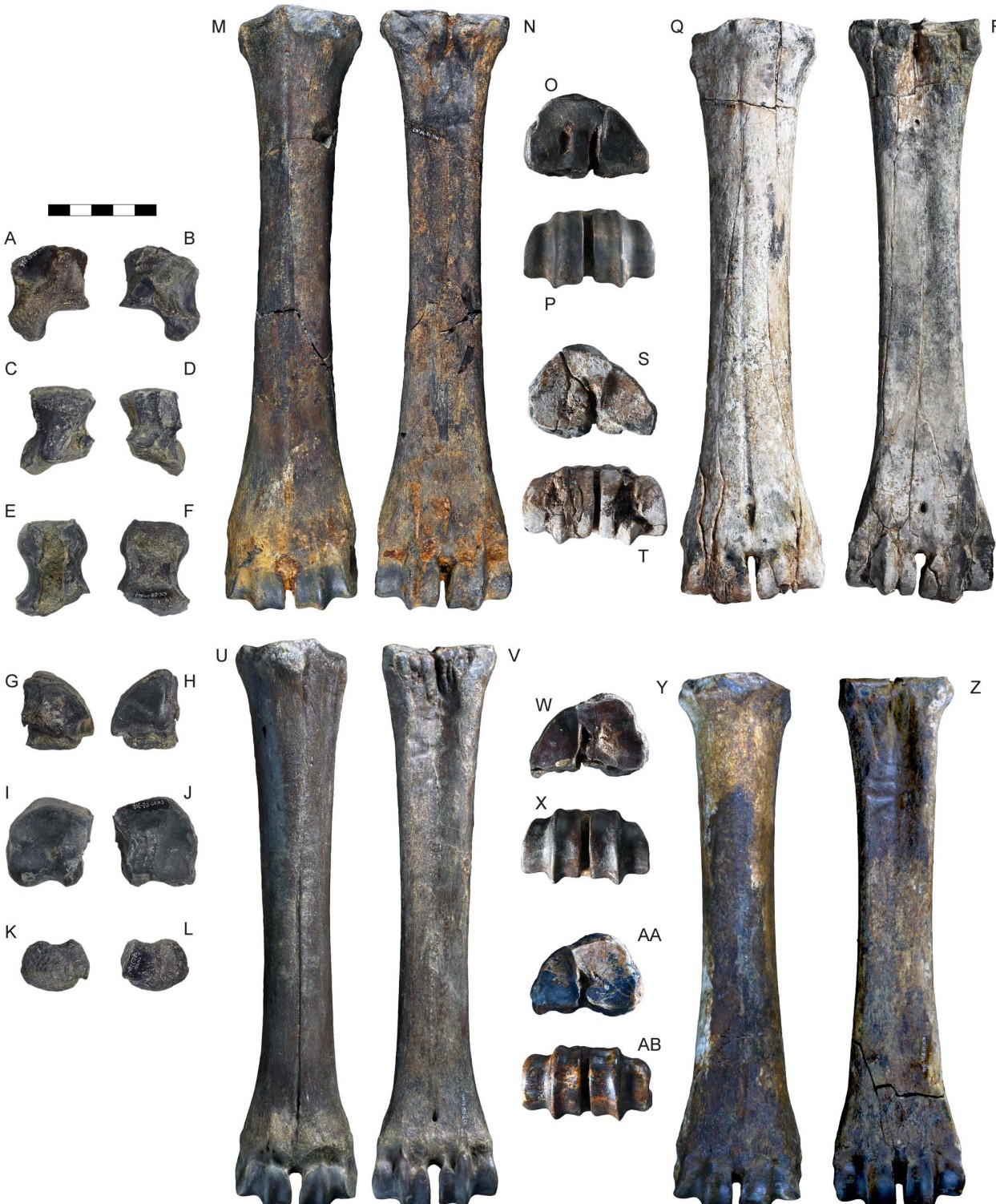

**Fig 13. Carpal and metacarpal bones of *Parabos tigneresi*.** A–B, right pyramidal IPHES.CN'05-B2-326 in lateral (A) and medial (B) views; C–D, right semilunar IPHES.CN'05-B2-428 in proximal (C) and distal (D) views; E–F, right scaphoid IPHES.CN'05-B2-313 in lateral (E) and medial (F) views; G–H, right unciform IPHES.CN'05-B2-311 in proximal (G) and distal (H) views; I–J, right magnum-trapezoid IPHES.CN'05-B2-312 in proximal (I) and distal (J) views; K–L, right pisiform IPHES.CN'05-B2-325 in lateral (K) and medial (L) views; M–P, right metacarpal IPHES.CN'04-B1-344 in anterior (M), posterior

(N), proximal (O) and distal (P) views; Q–T, right metacarpal IPHES.CN'11-B4-306 in anterior (Q), posterior (R), proximal (S) and distal (T) views; U–X, left metacarpal IPHES.CN'19-B15-432 in anterior (U), posterior (V), proximal (W) and distal (X) views; Y–AB, left metacarpal IPHES.CN'17-B14-316 in anterior (Y), posterior (Z), proximal (AA) and distal (AB) views. Scale bar: 50 mm.

but narrow condylar fossae, are the long paraoccipital processes (=jugal processes). These elements follow the lateral borders of the condyles, curving internally in their distal portion, exceeding the ventral limit of the condyles.

The frontals are relatively small and narrow, posteriorly elongated so they extend far back the horncore bases. The frontal foramina are small and located at the anterior edge of the pedicle, at the level of the posterior rim of the orbit. The strong deformation of the skulls does not allow to assess the presence of pits for the accommodation of the foramina but they seem to be absent (Fig 5A, G). The interfrontal suture is raised in a faint crest (Fig 5G). The orbits are circular, only slightly projecting and not tubular. Their anterior margin reaches the M3 (Figs 5; 6). The orbit rim is quite thin and sharp. The posterior orbital bar (frontal process of the zygomatic bone and zygomatic process of the frontal bone) is slim. The ethmoidal fenestra is absent. The lacrimal fossae seem to be not present although the strong deformation of the skull does not allow to assess properly the condition of this character which, if present, should be extremely reduced (Fig 5D, E). The lacrimal bone is large and squared with a weak indentation on the maxillary. The nasals are long ending posteriorly to the anterior margin of the orbits, indenting on the frontals with an acute angle (Fig 5A, E). The broadest portion of the nasals is at the level of the anterior lacrimomaxillary contact. The nasals anterior end is bifid with large lateral flanges which exceed the medial ones (Fig 5A). The maxilla is relatively high. The teeth row is located well below (i.e., ventrally to) the maxilla and premaxilla (Fig 5D). The infraorbital foramina are relatively small and located at the level of the P2. The premaxilla is long and thick. It does not taper distally and reaches the nasals but it stops before the level of P2 (Fig 5E, D).

The mandible is slender and ventrally convex. The ramus is at 90° with respect to the corpus (Figs 5; 6; 10). The posteroventral edge of the ramus is protruding distally and ventrally. The coronoid process is quite elongated and projects backwards, ending posterior to the condyle (Figs 10E, F). The notch between the coronoid process and condyle is very shallow.

**Dentition.** The teeth are large, characterized by incipient basal pillars and relatively prominent ribs, but possess a low crown. Most of the specimens, with the exception of IPHES.CN'12-B7-1, are fully adult or senile, making it difficult to precisely assess their true degree of hypsodonty. However, specimens IPHES.CN'05-B2 and IPHES.CN'11-B4 display a moderate wear stage (Figs 5; 9A–C; 10D–F) and specimen IPHES.CN'12-B7-1 represents a young adult, exhibiting the first two molars at an early stage of wear, with the fourth deciduous premolar still in position (Fig 7C–E). The crown height-to-length ratio of the second molars (both upper and lower) in the older two specimens is approximately 0.7–0.8. In contrast, the same ratio calculated for the upper molars in the young adult specimen is around 1.0–1.1.

The upper tooth row characterized by large premolars. Despite that, the premolar row length does not exceed that of the molar row (Figs 5; 6; Table 2). The teeth are quadrangular in occlusal shape with a wide base and strong pillars. There is a small quantity of cement on the labial surface of the molars and the buccal walls are wrinkled.

The P2 is large, slightly longer than P3 (Table 2). In occlusal view, P2 and P3 are highly asymmetrical with both parastyle and paracone shifted mesially (Fig 9A, D, G). In P2 these elements are so displaced that the parastyle is almost lying on the mesial border of the tooth. Both premolars feature a distally pinched occlusal outline especially in the P3. The paracone fold—the groove separating the paracone from the parastyle—is generally deep. The third premolar has a stronger and distally curving parastyle. The metastyle projects distally, separated from the paracone by a wide and shallow valley. The metacone is not recognizable, being fused with the metastyle. Protocone and hypocone are completely fused in P2 and weakly separated in the P3 (Fig 9A, D, G). The enamel central cavity of both premolars is simple (i.e., folds absent or weak when present). The P4 has roughly the same size of P3 albeit slightly smaller (Table 2); has a squared shape with symmetrical occlusal outline (Fig 9A, D, G). The paracone generally occupies the central part of the tooth or is mesially shifted (e.g., in IPHES.CN'06-B1-1) and it is distally and buccally bordered by two shallow grooves. When the paracone is

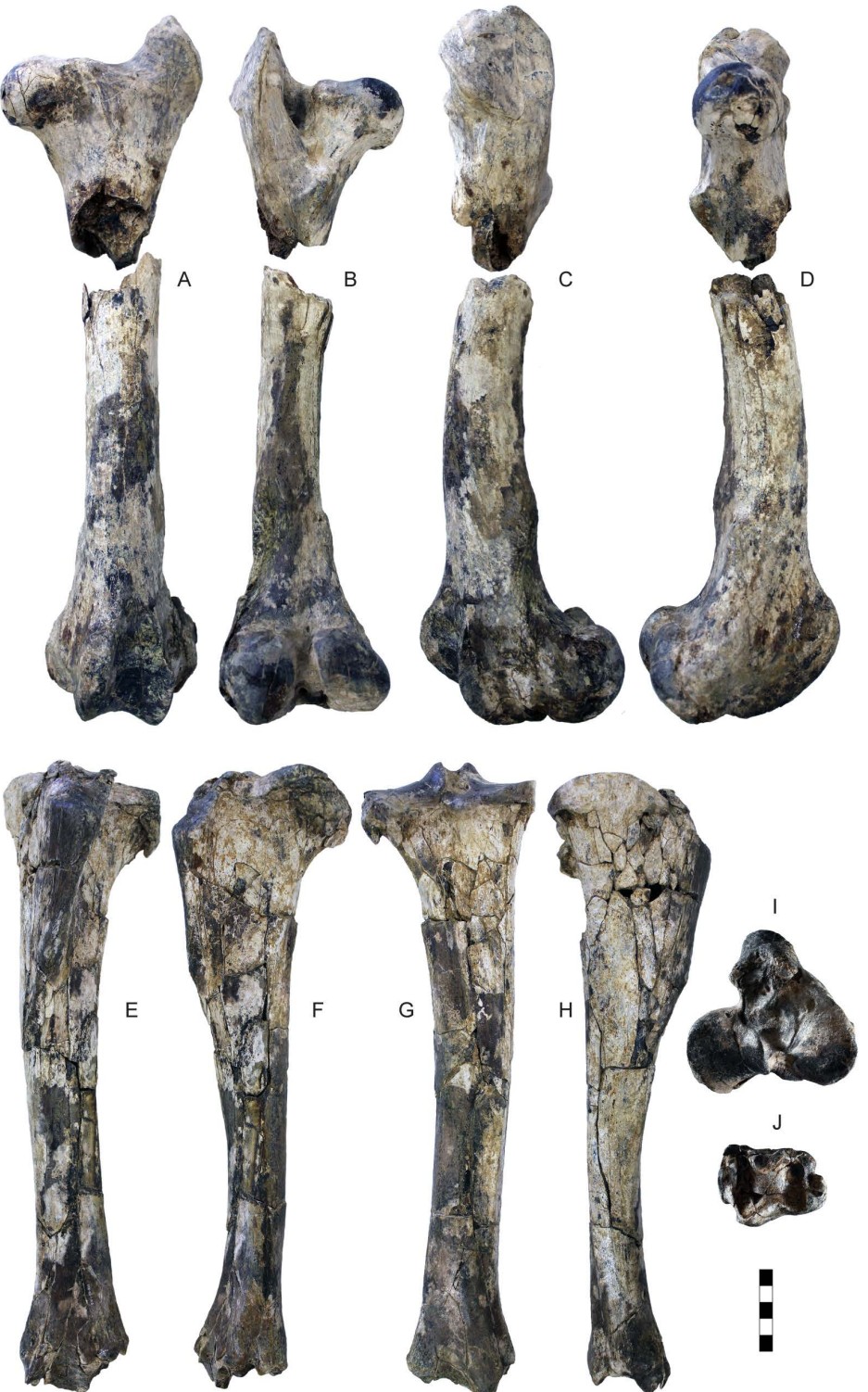

**Fig 14. Femur and tibia of *Parabos tigneresi*.** A–D, left femur IPHES.CN'11-B4-401 in anterior (A), posterior (B), lateral (C) and medial (D) views; E–J, left tibia IPHES.CN'11-B4-403 in anterior (E), lateral (F), posterior (G), medial (H), proximal (I) and distal (J) views. Scale bar: 50 mm.

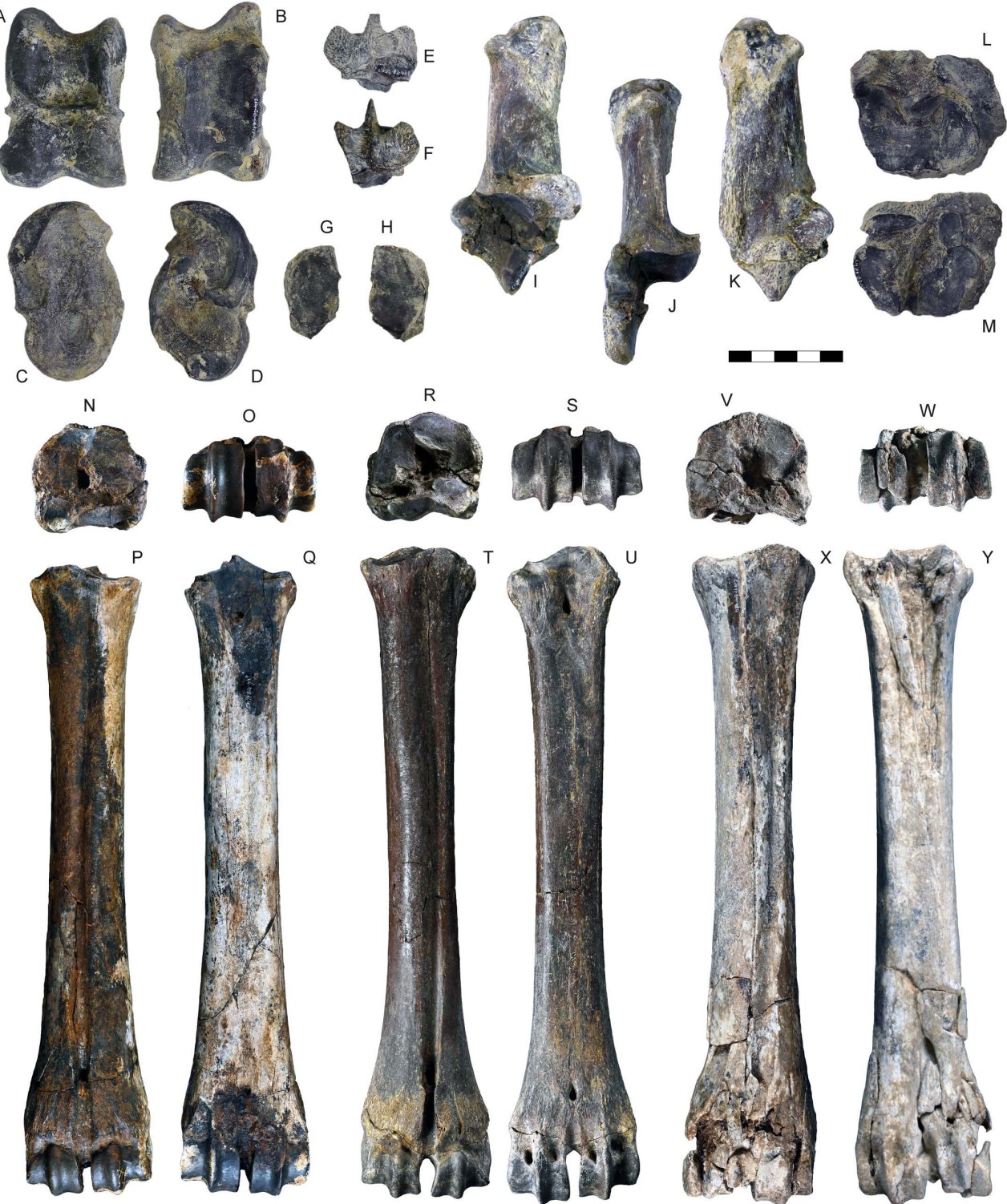

**Fig 15. Tarsal and metatarsal bones of *Parabos tigneresi*.** A–D, right astragalus IPHES.CN'05-B2-402 in anterior (A), posterior (B), medial (C) and lateral (D) views; E–F, right malleolus IPHES.CN'05-B2-409 in lateral (E) and medial (F) views; G–H right cuneiform IPHES.CN'05-B2-410 in proximal (G) and distal (H) views; I–K, right calcaneum IPHES.CN'19-B15-308 in medial (I), anterior (J) and lateral (K) views; L–M, left cubonavicular IPHES.CN'05-B2-400 in proximal (L) and distal (M) views; N–Q, right metatarsal IPHES.CN'17-B14-447 in proximal (N), distal (O), anterior (P) and

posterior (Q) views; R–S, left metatarsal IPHES.CN'19-B15-300 in proximal (R), distal (S), anterior (T) and posterior (U) views; V–Y, left metatarsal IPHES.CN'11-B4-404 in proximal (V), distal (W), anterior (X) and posterior (Y) views. Scale bar: 50 mm.

located more mesially the mesial groove is deeper than the distal one. The paracone rib is weak. The parastyle and metastyle are both well-developed and projecting buccally. The protocone lingual wall is rounded although slightly pinched. The outline of the central cavity is U-shaped, featuring small folds at the mesiolingual and distolingual borders (Fig 9A, D, G).

The upper molars show a low degree of hypsodonty with strong styles and almost parallel margins, slightly diverging occlusally (Figs 7E; 9B). The protocone is generally mesiodistally compressed and lingually shifted. Parastyle and mesostyle are both well-developed although not strongly pinched (i.e., weak constriction). In some cases, the mesostyle can be mesially projecting, especially in M1 (e.g., in IPHES.CN'06-B1-1; Fig 9D). The metastyle is the least developed of all the labial columns with the exception of M3 in which this style is large and distally projecting (Fig 9A, D, G). The paracone and metacone ribs are relatively weak. The entostyle lies between the protocone and hypocone, attached for most of its length to the lobes (Fig 9A, D). It is well-developed and mesiodistally compressed, with an irregular but simple outline. Cementum is scarce but deep in places, especially between the entostyle and the lingual wall of the two lobes (Fig 9C, H). The outline of the central enamel cavities has an irregular crescent shape, sometimes characterized by faint folds, mainly concentrated in the distal wall (Fig 9A, D, G). Between the two cavities a small, circular, enamel islet is often found (see IPHES.CN'05-B2; Fig 9A).

The lower teeth are characterized by large premolars and mesiodistaly elongated molars (Fig 10; Table 2). Cementum is mostly absent, the enamel is wrinkled and the basal pillars are well-developed. In most of the specimens the teeth are in an advanced state of wear.

The incisor morphology is similar for i1–i3. The crown of the tooth has a shovel-like shape. The size of the teeth and the width of the occlusal surface decrease moving toward the distal incisors with the i1 being the largest and the i3 the smallest (Fig 6C; Table 2). There is a small canine which strongly resembles the i3 in both size and morphology. The p2 is reduced (shorter than p3) but still relatively large (Table 2). The p3 and p4 are extremely similar with the exception of few elements (Fig 10C, D, G, I). The parastylid is sharp and projects mesiolingually. The paraconid lies perpendicular to the mesiodistal axis of the tooth. These two stylids are divided by a shallow and short furrow that ends well before the cervix of the tooth (1st valley, according to [58]; Fig 10D, I). This groove is normally deeper in p4 (Fig 10D, G). The large median valley which separates the paraconid from the metaconid (2nd valley, sensu [58]; anterior valley, sensu [59]), is deep with a U-shape in p4 and quite shallow with a V-shape in p3, being partially obliterated by the incipient protoconid. In few instances (e.g., specimen IPHES.CN'11-B4-1, Fig 10I) the p4 2nd valley is almost closed by the mesial protrusion of the metaconid. The metaconid of p3 is a large but sharp pillar developing distally, reaching the mesial border of the entoconid (Fig 10C, D, G, I). The p4 metaconid projects lingually, perpendicular to the main axis of the tooth. It is larger than in p3 and it expands mesiodistally (Fig 10C, D, G, I). In the less worn p4s the metaconid is smaller and more developed distally (Fig 10D). As wear increases, the metaconid expands mesiodistally, inflating and occupying most of the central part of the tooth. In both p3 and p4, the posterior valley (3rd valley sensu [58]; back valley sensu [59]) is quite deep but ends well before the bottom of the crown. At mid-late wear of stage this valley is often closed by the distal border of the metaconid contacting with the entoconid and become an islet (i.e., IPHES.CN'11-B4-1, IPHES.CN'12-B13-1, Fig 10I). The entoconid and entostylid are almost completely fused, separated by a very faint groove which tends to disappear at late wear stage (still faintly visible in IPHES.CN'05-B2-1, Fig 10F). There is a wide but shallow labial valley produced by the labial projection of the protoconid and hypoconid (Fig 10F, I).

The molars are mesiodistally elongated with relatively strong ribs (Fig 10A, B, D, E, H; Table 2). The protoconid of m2 and m3 is mesiodistally compressed whereas the hypoconid is more elongated (Fig 10C, F). The parastylid projects mesiolingually (especially in m2 and m3). The metastylid is absent (Fig 10D, G). On the mesiolabial side of m3 there is a

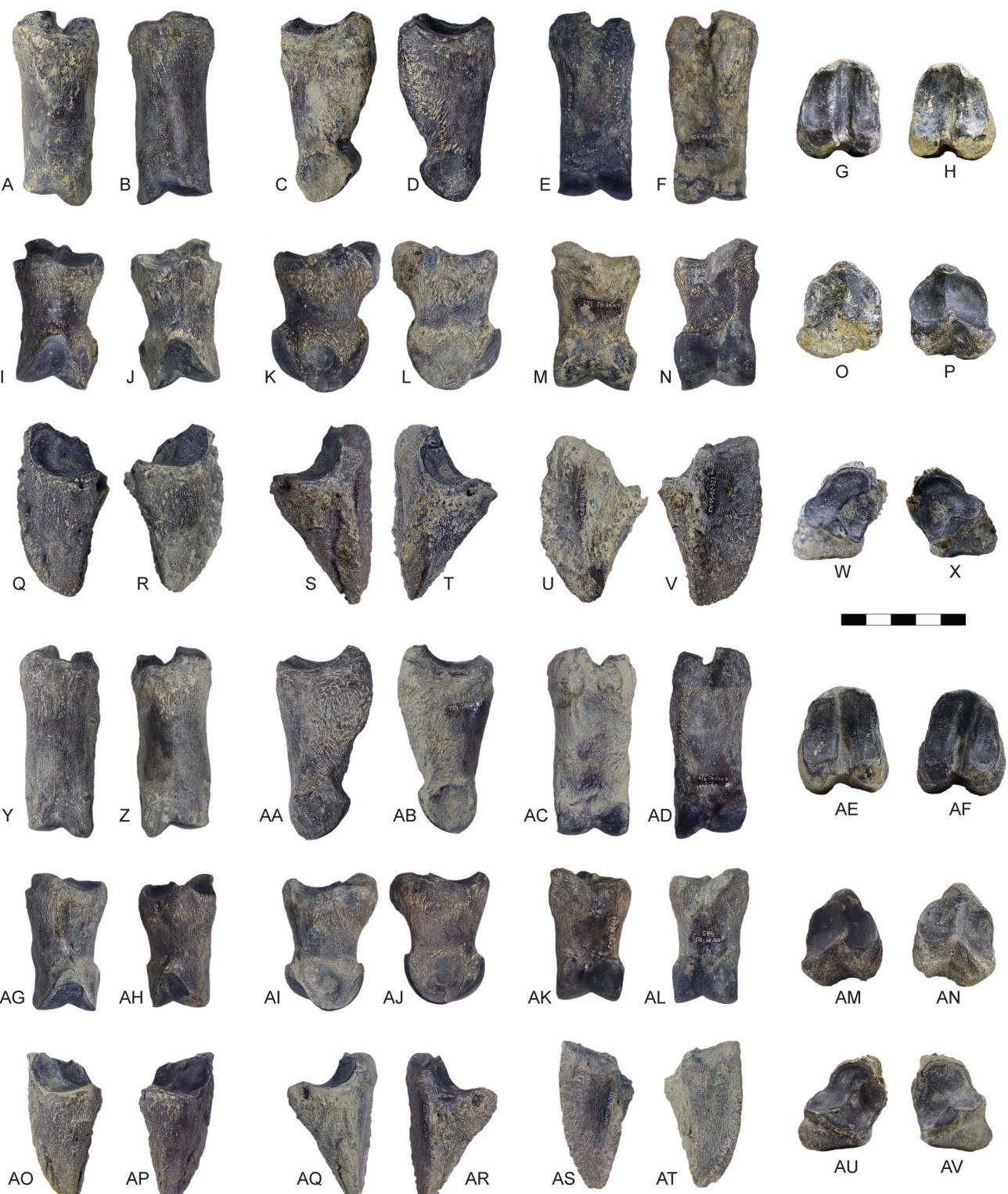

**Fig 16. Phalanges of *Parabos tigneresi*.** A, C, F, H, right proximal phalanx of the right forelimb IPHES.CN'05-B2-320 in anterior (A), abaxial (B), posterior (C) and proximal (D) views; B, D, E, G, left proximal phalanx of the right forelimb IPHES.CN'05-B2-333 in anterior (B), abaxial (D), posterior (E) and proximal (G) views; I, K, N, P, right intermediate phalanx of the right forelimb IPHES.CN'05-B2-332 in anterior (I), abaxial (K), posterior (N) and

proximal (P) views; J, L, M, O, left intermediate phalanx of the right forelimb IPHES.CN'05-B2-323 in anterior (J), abaxial (L), posterior (M) and proximal (O) views; Q, S, V, X, right distal phalanx of the right forelimb IPHES.CN'05-B2-322 in anterior (Q), abaxial (S), posterior (V) and proximal (X) views; R, T, U, W, left distal phalanx of the right forelimb IPHES.CN'05-B2-425 in anterior (R), abaxial (T), posterior (U) and proximal (W) views; Y, AA, AD, AF, right proximal phalanx of the right hindlimb IPHES.CN'05-B2-329 in anterior (Y), abaxial (AA), posterior (AD) and proximal (AF) views; Z, AB, AC, AE, left proximal phalanx of the right hindlimb IPHES.CN'05-B2-432 in anterior (Z), abaxial (AB), posterior (AC) and proximal (AE) views; AG, AI, AL, AN, right intermediate phalanx of the right hindlimb IPHES.CN'05-B2-435 in anterior (AG), abaxial (AI), posterior (AL) and proximal (AN) views; AH, AJ, AK, AM, left intermediate phalanx of the right hindlimb IPHES.CN'05-B2-429 in anterior (AG), abaxial (AI), posterior (AL) and proximal (AN) views; AO, AQ, AT, AV, right distal phalanx of the right hindlimb IPHES.CN'05-B2-426 in anterior (AO), abaxial (AQ), posterior (AT) and proximal (AV) views; AP, AR, AS, AU, left distal phalanx of the right hindlimb IPHES.CN'05-B2-427 in anterior (AP), abaxial (AR), posterior (AS) and proximal (AU) views. Scale bar: 50 mm.

faint goat fold at the tip of the crown (e.g., IPHES.CN'05-B2-1, Fig 10G). The ectostylid is well-developed but narrow and located in the mid-point between the two lobes. In occlusal view, the ectostylid of m2 and m3 has an ovoidal outline and is isolated from the main body of the tooth whereas in m1 has a more complex outline and lies attached to the hypoconid (Fig 10C, D, G). The labial margin of the lobes is V-shaped in early wear stages and U-shaped in later wear stages (Fig 10C, D, G). The hypoconulid lies on the same plane of the first two lobes although labially inclined. This latter element is connected to the hypoconid through a wide bridge. In few cases there is a small enamel islet on this bridge, right distally to the hypoconid's central cavity (Fig 10D, G). The cavities have relatively simple, half-moon shaped outlines. The m2s might present a small fold at the mesial wall of the protoconid central cavity (Fig 10G). The distal stylid protrudes linguodistally (Fig 10C, D, G).

**Postcranium.** Overall, the body is large and bulky with long limbs (Fig 11). The number of vertebrae (not considering the caudal) is 30, including 7 cervical, 13 thoracic, 6 lumbar and 4 sacral (S2–S6 Tables. The thoracic vertebrae are characterized by tall neural spines, especially in the first half of the thorax (i.e., n° II–VIII; S5 Table).

The humerus, radius, and metacarpal have similar lengths although there is a slight reduction of the distal elements (radius/humerus lengths ca. 1.0, metacarpus/humerus lengths ca. 0.8–0.9; S8; S9; S17 Tables). The humerus is relatively stout, with massive epiphyses and large insertions for the muscles (Fig 12A–H; S8 Table). The shaft is strongly convex in its medial side and presents a well-developed deltoid tuberosity. The trochlear crest is shifted toward the lateral part of the articulation so the medial hemitrochlea is almost double the size of the lateral one. The radius is large. It possesses a relatively thin diaphysis in the most gracile specimens and more robust shaft in the largest individuals (e.g., IPHES. CN'04-B1-303 and IPHES.CN'11-B4-303; Fig 12J–Q; S9 Table). The lateral tuberosity of the proximal epiphysis is well-developed. The angle formed by the proximal surface of this tuberosity and the lateral edge of the epiphysis is wider than 90°. The ulna's oleocranon is long and wide, bended on the ulna's diaphysis (Fig 12I; S10 Table). The metacarpal is hourglass shaped with large epiphyses and long but robust diaphysis (Fig 13M–Z; S17 Table). The proximal epiphysis is deep although the transversal diameter is larger than the anteroposterior one. In the proximal articulation, the medial facet is almost double the size of the lateral. The diaphysis cross section is half-moon shaped, anteroposteriorly compressed, with flattened posterior side. Its robusticity changes accordingly to the individual (e.g., slenderer in IPHES.CN'19-B15-432 and stouter in IPHES.CN'11-B4-306). The diaphysis enlarges right above the contact with the distal epiphysis, so the transversal diameter measured above the articulation is roughly the same of the diameter measured at the articulation level. The anterior sulcus is long but narrow and shallow. The trochlear ridges are relatively weak and the articulation surface is low. The ridges are subparallel, slightly diverging distally.

The hindlimbs show similar proportions to the anterior ones (tibia/femur lengths ca. 1.0; metatarsus/femur lengths ca. 0.8) (Figs 14; 14; S19; S21; S27 Tables). The femur is long and characterized by large epiphyses (Fig 14A–D; S19 Table). The great trochanter is large and very high. The medial trochlear ridge of the distal epiphysis is much larger than the lateral one, featuring also a blunter crest. Both condyles are massive, expanding mediolaterally thus partially closing the intercondylar fossa. The medial condyle is rounded whereas the lateral one is slightly compressed in mediolateral direction and tilted proximomedially to distolaterally. The tibia is relatively stout, roughly of the same length of the femur

(Fig 14E–J; S21 Table). The proximal epiphysis is mediolaterally expanded; the lateral condyle is anteroposteriorly compressed and has a concave posterior edge. The intercondylar eminence rises well above the proximal articular surface. The tibial tuberosity on the anterior portion of the proximal epiphysis is pronounced. On the distal epiphysis the medial and lateral articular grooves are parallel to the sagittal axis of the tibia. The astragalus is large and stout (Fig 15A–D; S23 Table). The intertrochlear notch is relatively shallow but deeper than the one at the distal articulation. The medial distal trochlea projects more medially than the lateral one on its distalmost portion. The calcaneus is relatively elongated due to its well-developed tuber calcanei (Fig 15I–K; S24 Table). This element has parallel anterior and posterior margins. The tuberosity (=tuber calcis) has a deep indentation on its anterior margin. The cubonavicular (=central tarsal bone) is squared and robust (Fig 15L, M; S25 Table). On the proximal surface, the articulation facet for the accommodation of the astragalus presents a wide synovial fossa that clearly divide the facet in two. The metatarsal is hourglass shaped with long and straight diaphysis and large epiphyses (Fig 15N–Y; S27 Table). The proximal epiphysis is composed by two major articulation facets, roughly of the same shape and size, on the anterior half, and two smaller on the posterior border. The posteromedial facet is subrounded and located at the posterior corner of the epiphysis, whereas the posterolateral facet has sigmoidal shape and its major axis develops perpendicularly to the sagittal plane of the surface. The cross section of the diaphysis is quadrangular marked by a deep, long sulcus on the anterior border which narrows its distal portion but reaches the intertrochlear incisure. The distal diaphysis inflates at the contact with the epiphysis. The distal articulation surface is reduced and the trochlear ridges are blunt and subparallel. The phalanges are large and stout (Fig 16; S28–30 Tables). The forelimb elements are more massive (i.e., shorter and stouter) than the hindlimb ones, especially in the intermediate and distal phalanges. The proximal phalanx is the longest one and presents a very thick proximal half (Fig 16A–H, Y–AF). The proximal articulation is divided into two facets by a deep and wide groove. The axial (=interdigital) surface of the shaft is rugose. The distal articulation is visible in anterior (=dorsal) view. The intermediate phalanx is shorter, presenting a well-developed distal articulation surface which extends onto the anterior and posterior (=ventral=palmar) surfaces (Fig 16I–P, AG–AN). Posteriorly to the proximal articulation there is a well-developed plateau which is more proximally protruding in its lateral side. The posterior surface is crossed by a wide sagittal groove. The outline of the abaxial distal articulation is rounded whereas the axial outline is more triangular with the apex pointing posterodistally. The distal phalanx has a markedly concave axial edge (Fig 16Q–X, AO–AV). The proximal articulation is inclined so the apex (i.e., anterior point) of the articulation is located axially. The articulation is elevated above the ground due the presence of a bone growth below it (i.e., posteriorly). There is no process at the anterior margin of the articulation. Lumbar vertebrae, scapula, carpals, pelvis, patella and other tarsals are typical for Bovidae, not presenting any peculiarity worth noticing (Figs 11; 13; 15; S6; S7; S11–S16; S17; S20; S26 Tables).

## Comparisons

The bovid from Camp dels Ninots belongs to the Bovinae subfamily given its quadrangular braincase (i.e., 'box-shaped' neurocranium), low facial-cranial flexion, keeled horn cores lacking transverse ridges, and simple cranial sutures [22,24]. Extant Bovinae is composed of the tribes Tragelaphini, Boselaphini and Bovini in addition to the saola (*Pseudoryx nghtinhensis*), which has been placed in its own tribe (Pseudorygini) or within Bovini [e.g., 60, 61]. Most Miocene Bovinae have traditionally been referred to the Boselaphini, even though workers have long suspected these to include the ancestors of Bovini (see [5] and references therein). Therefore, it has been proposed that species potentially on the line to Bovini be recognized as stem bovins, those on the line to Boselaphini as stem boselaphins, and numerous species of *Tragoportax, Miotragocerus,* and related boselaphine-like bovids treated as a separate tribe, Tragoportacini (e.g., [5], see Materials and Methods). Among Neogene Bovinae, the Camp dels Ninots bovid shows cranial and dental characters similar to both Tragoportacini and early (stem) Bovini from the Miocene and Pliocene of Eurasia and Africa. Consequently, our comparisons focus primarily on these groups and extant tribes of Bovidae (Figs 17–24). Due to the deformation of the cranial elements, complete metrics could not be obtained; thus, morphological and morphometric comparisons were largely confined to the most intact and undeformed elements, including horns, dentition, and occipital and basioccipital bones.

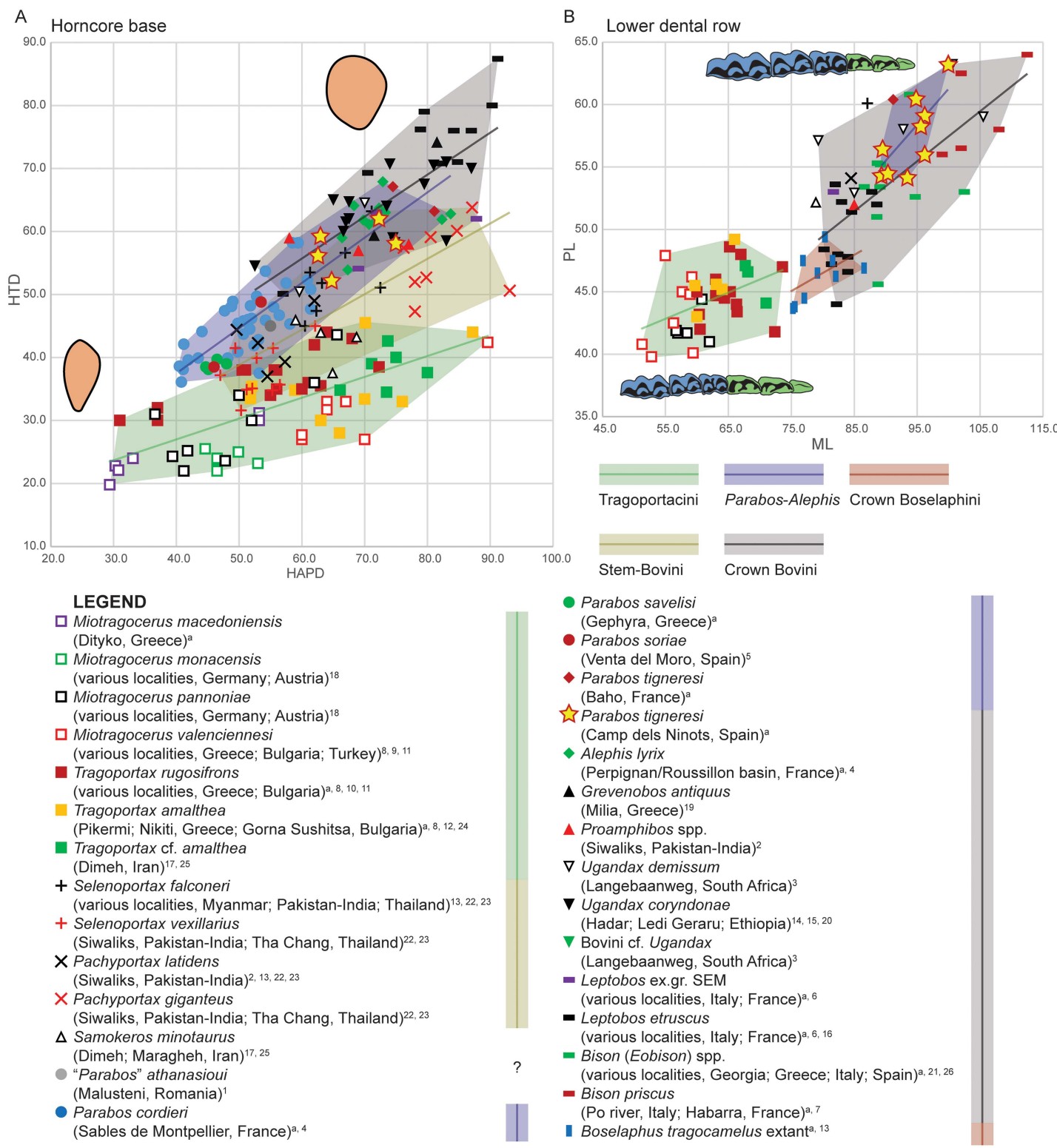

**Fig 17. Metric comparison of cranial and dental elements in *Parabos tigneresi* and other bovids from the Eurasian Neogene and Quaternary.**
A, basal horn core transversal and anteroposterior diameters; B, lower molar and premolar row lengths. HAPD, anteroposterior diameter of the horncore; HTD, transverse diameter of the horncore; ML, lower molar row length; PML, lower premolar row length. Superscript denotes data source: a, this study;

1, [62]; 2, [18]; 3, [63]; 4, [20]; 5, [51]; 6, [57]; 7, [64]; 8, [65]; 9, [42]; 10, [66]; 11, [67]; 12, [68]; 13, [69]; 14, [70]; 15, [71]; 16, [72]; 17, [73]; 18, [38]; 19, [74]; 20, [75]; 21, [76]; 22, [44]; 23, [77]; 24, [39]; 25, [36]; 26, [28]. The full dataset is provided in S33 and S34 Tables.

The horns of the CN bovid slightly bend posteromedially with a faint heteronymous torsion and are characterized by one prominent frontal keel and two weaker posterior keels and a triangular cross-section (Figs 23P; 24G). These features are shared with the Baho *Alephis tigneresi*, distinguishing both the Spanish and French samples from the younger *Alephis lyrix*, which, exhibits longer horn cores with more pronounced heteronymous torsion and medial curvature of the cores (Figs 23S; 24J; Table 3). In addition, *A. lyrix* differs from the CN bovid in having a wider divergence angle of the horns, deeper grooves on the medial surface, and an ovoidal cross-section with a less prominent posterolateral keel (Figs 23S; 24J). The horns of *A. tigneresi* and *Parabos cordieri* show strong similarities. Nevertheless, *P. cordieri* horns differ from CN and Baho samples in the following aspects: smaller absolute size, less divergent horn bases, weaker medial bending and torsion, more prominent lateral keel and absence of surface grooves (Figs 23M; 24F; Table 3). *Parabos savelisi* from Gephyra (Greece), differs from the bovid from CN in the same characters as *P. cordieri* (Figs 23J; 24E; Table 3). Additionally, *P. savelisi* is marked by a strong reduction of the three keels, resulting in a sub-ovoidal cross-section (Fig 24E, [54]), distinguished from the triangular cross-sections of *P. cordieri* and *A. tigneresi*. Due to the extremely limited material available for *Parabos soriae*, *Parabos macedoniae*, and *Parabos athanasioui*, detailed cranial comparisons are not possible. However, the horncores of both *P. soriae* and *P. athanasioui* are notably smaller than those of the CN bovid (Fig 17A). A commentary on the diagnostic traits and taxonomic status of these species is provided in the discussion (see next chapter). The horns of the studied sample differ from the typical Tragoportacini morphology which presents, apart from the overall smaller size, a stronger mediolateral compression and, often (as it is the case of *Miotragocerus*), only two well-defined keels (Figs 17A; 18C; 23A, D; 24A, B). In addition, some Tragoportacini show a stepped anterior margin on the anterior keel which is completely absent in the sample from CN. The Late Miocene *Samokeros minotaurus* from Greece and Iran differs from CN specimens in smaller horns, with oval basal cross-section, mediolaterally flattened distal cross-section, stronger medial curvature, and lacking keels (Figs. 17A; 24D). The bovins *Grevenobos antiquus* (Late Pliocene, Balkans) and *Leptobos* spp. (Plio-Pleistocene, Eurasia), although having similar basal horn size, show some derived Bovini features, namely marked grooves, deep pneumatization that extends into the horn core and wide horn divergence, which allow us to separate them from the CN sample (Figs 23Y; 24H, M). The horns of putative stem Bovini/early Bovini such as *Selenoportax*, *Pachyportax*, and *Proamphibos* (South Asia) can be differentiated from both *A. tigneresi* and the CN bovids by their greater mediolateral compression and more developed anterior keel (*Selenoportax* spp. and *Pachyportax* spp.), stronger torsion (*Selenoportax* spp. and *Pachyportax* spp.), and more pronounced medial curvature (*Selenoportax vexillarius*, *Pachyportax giganteus*, and *Proamphibos* spp.) (Figs 17A; 18C; 23G; 24I, L). Lastly, species of *Ugandax* (Pliocene Africa) differ from CN bovid in keels less evident, more pronounced medial curvature, deeper surface grooving, and greater divergence, all indicating that this genus belongs in the Bovini (Fig 23V). Horn core dimensions in CN specimens are similar to those in *Parabos*, *Alephis*, *Grevenobos*, *Ugandax*, *Proamphibos* and *Leptobos* (Fig 17A). These forms all have horn bases lacking significant mediolateral compression, unlike the moderate compression in *Pachyportax* and *Selenoportax* and the extreme compression in *Miotragocerus* and *Tragoportax* (Fig 18C). In particular, the studied sample fall within the cluster represented by the large-sized bovids with weakly compressed horns such as *A. lyrix*, *A. tigneresi*, and *Parabos* spp. (i.e., *Parabos-Alephis* cluster), also partially overlapping with the crown bovini group formed by *Leptobos*, *Proamphibos* and *Ugandax* spp (Figs 17A; 18C).

The morphology of the braincase in the CN bovid aligns it with the earliest stem Bovini. While the mastoid region is larger than that of Miocene Tragoportacini, the braincase relative length is shorter. Similar proportions can be observed in more evolved forms such as *Parabos*, *Alephis*, *Selenoportax* and *Ugandax*. The two convergent temporal crests are elevated above the parietals and the supraoccipital constriction, despite being well-developed is not as long and narrow as it is in *Miotragocerus* or *Tragoportax* (Fig 23B, E, Q). The CN bovid supraoccipital is similar to that in *Selenoportax*,

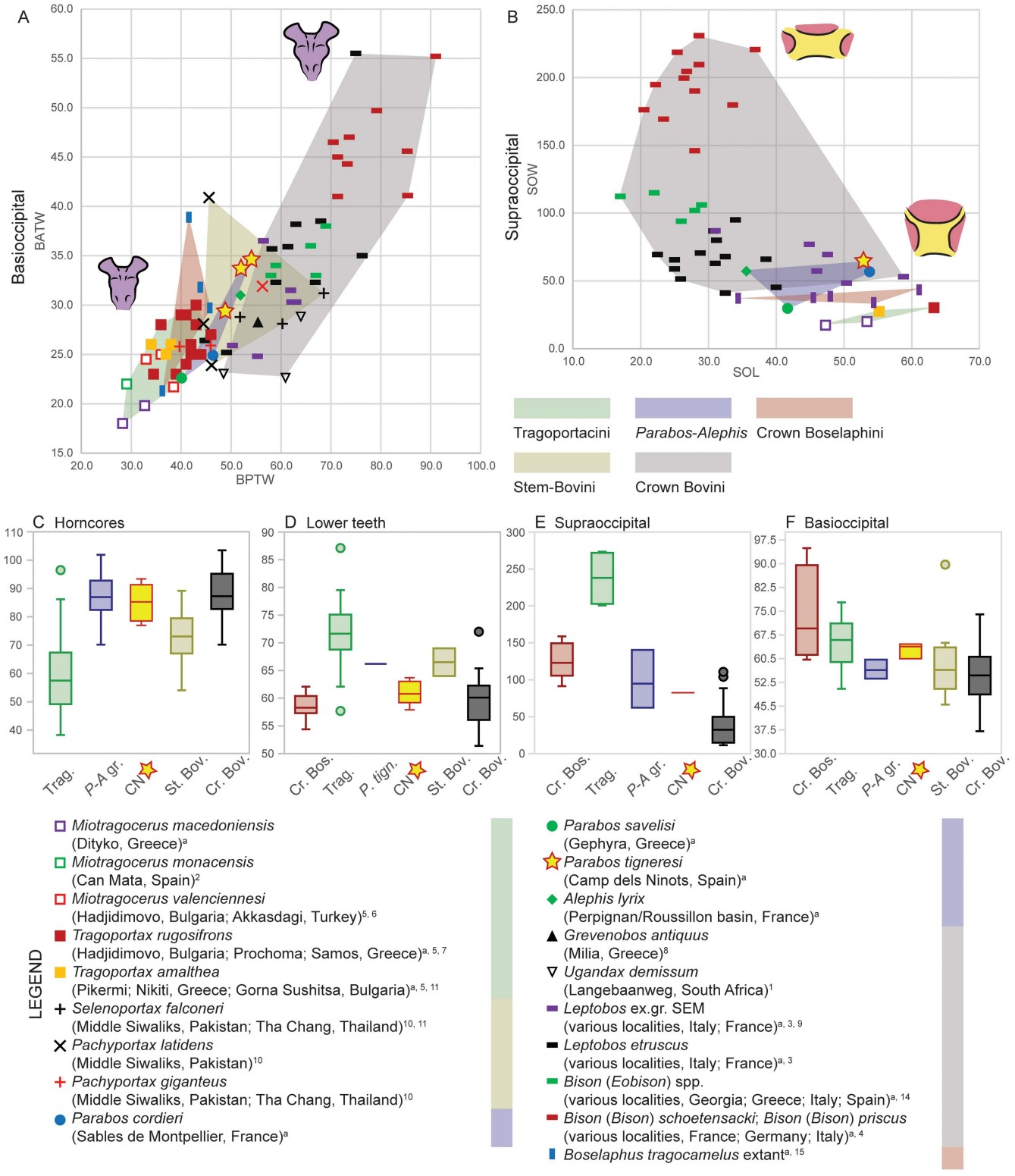

**Fig 18. Metric comparison of cranial and dental elements in *Parabos tigneresi* and other bovids from the Eurasian Neogene and Quaternary.** A, basioccipital tuberosities transversal diameters; B, supraoccipital length and width; C, boxplot of the horncore base compression index: D, boxplot of the lower premolar index; E, boxplot of the supraoccipital compression index; F, boxplot of the basioccipital tuberosities index. Abbreviations: BATW,

basioccipital width at the anterior tuberosities; BPTW, basioccipital width at the posterior tuberosities; SOL, supraoccipital length; SOW, supraoccipital width. Superscript denotes data source: a, this study; 1, [63]; 2, [78]; 3, [57]; 4, [64]; 5, [65]; 6, [42]; 7, [67]; 8, [74]; 9, [27]; 10, [44]; 11, [77]); 12, [39]; 13, [36]; 14, [28]. The full dataset is provided in S35 and S36 Tables.

*Parabos*, *Alephis lyrix* and the earliest forms of *Leptobos* ex gr. SEM (i.e., *L. stenometopon*, *L. elatus*, *L. merlai*) but longer and narrower than in later *Leptobos* ex gr. EV (i.e., *L. etruscus*, *L. vallisarni*) (Fig 23H, K, N, Q, T, Z). The supraoccipital in IPHES.CN'11-B4-1 falls within the variation of *Leptobos* ex gr. SEM and next to the cluster formed by *A. lyrix*, *P. cordieri* and *P. savelisi*, which have a narrow but relatively short supraoccipital (Fig 18B). Tragoportacini are characterized by an extremely elongated and narrow supraoccipital. On the other side of the spectrum are derived Bovini such as *Bison*, which has a very anteroposteriorly compressed and wide supraoccipital. This element does not form a part of the skull roof and migrates on the occipital plane and became indistinguishable from the nuchal crest. This character is clearly visible in Fig 18E where the CN bovid and the *Parabos-Alephis* group have intermediate values of compression index between the crown Bovini and Tragoportacini. The rugosities covering the cranial roof, typical of Boselaphini and most *Miotragocerus* and *Tragoportax* species seems to be extremely reduced in the CN bovid. The basioccipital of the CN bovid is characterized by a triangular shape, with narrow, elongated anterior tuberosities and wide, bulging posterior tuberosities separated by a sagittal groove (Figs 5F; 23R). Bovini are distinguished from Tragoportacini by greater width across the posterior tuberosities (i.e., greater size), with the CN specimens intermediate between the two tribes, partially overlapping with latter group and the early Bovini (Fig 18A). The index given by the ratio between the anterior and posterior tuberosities indicates strong homogeneity within the comparative sample (Fig 18F). This result suggests that the relative widths of the anterior and posterior tuberosities do not distinctly differentiate the two tribes, although the most basal tragoportacines (e.g., *Miotragocerus*) tend to have more squared and smaller basioccipital. The sagittal groove of the basioccipital is quite narrow but marked by steep and clear walls, strongly resembling the condition in some specimens of *Tragoportax* and *Selenoportax* and differing from most of the other bovids (Fig 23I, R).

The dentition of the CN specimens is large (Table 2) with teeth larger than *Tragoportax* and *Miotragocerus* and in the size range of bovines such as *Selenoportax*, *Pachyportax*, *Parabos*, *Alephis*, and *Leptobos* (Figs 17B, 18D, 19, 20; Table 4). The premolar row is reduced compared to the molar row, a typical trait in Bovini and differing from the long premolars in Tragoportacini (Figs 16B; 18D). Fig 18D shows the CN specimens are of similar premolar reduction as crown Bovini and extant *Boselaphus* whereas the only specimen of *Alephis tigneresi* from Baho and early Bovini possess slightly longer premolar row and Tragoportacini has the highest values for this element.

The overall tooth morphology of the CN bovid resembles that of Late Miocene Asian forms such as *Selenoportax* and *Pachyportax*, as well as Pliocene European genera including *Parabos* and *Alephis*. Features such as sparse cementum, relatively weak styles/stylids, small goat folds on m3, a mesiodistally expanded metastyle on p4, and rugose enamel distinguish the CN sample from more advanced Bovini, such as *Leptobos*, *Ugandax*, and *Proamphibos* [63,69,79, among others]. Conversely, the presence of incipient labial and lingual basal pillars, although primitive, sets the CN bovid apart from the typical Tragoportacini of the Eurasian Miocene in which this feature is less developed. In both upper and lower molars, the CN remains have stronger ribs, larger styles and more complex internal enamel islets than in *Miotragocerus* and *Tragoportax*. Additionally, traces of cementum and the labially shifted hypoconulid on m3 distinguish the CN remains from *Miotragocerus*, *Tragoportax* and *Parabos cordieri*. Despite these differences, the studied sample are associated to the aforementioned genera by a p4 with a mesiodistally expanded metaconid and consequently reduced 2nd valley. In younger individuals, the metaconid is well-separated from the paraconid and projects distally, creating a wide 2nd valley. In more worn p4s, the metaconid expands mesially and gradually acquires the typical Tragoportacini morphology. The specimens characterized by very late wear stages show a narrowing of the 2nd valley, caused by the nearing of the paraconid and metaconid. This character is shared with *P. cordieri* but not with *A. lyrix*, in which the metaconid does not expand mesially and the 2nd valley rests open with wear. The CN bovid, however, is similar to *A. lyrix* in the closure of the 3rd valley in mid wear which, on the

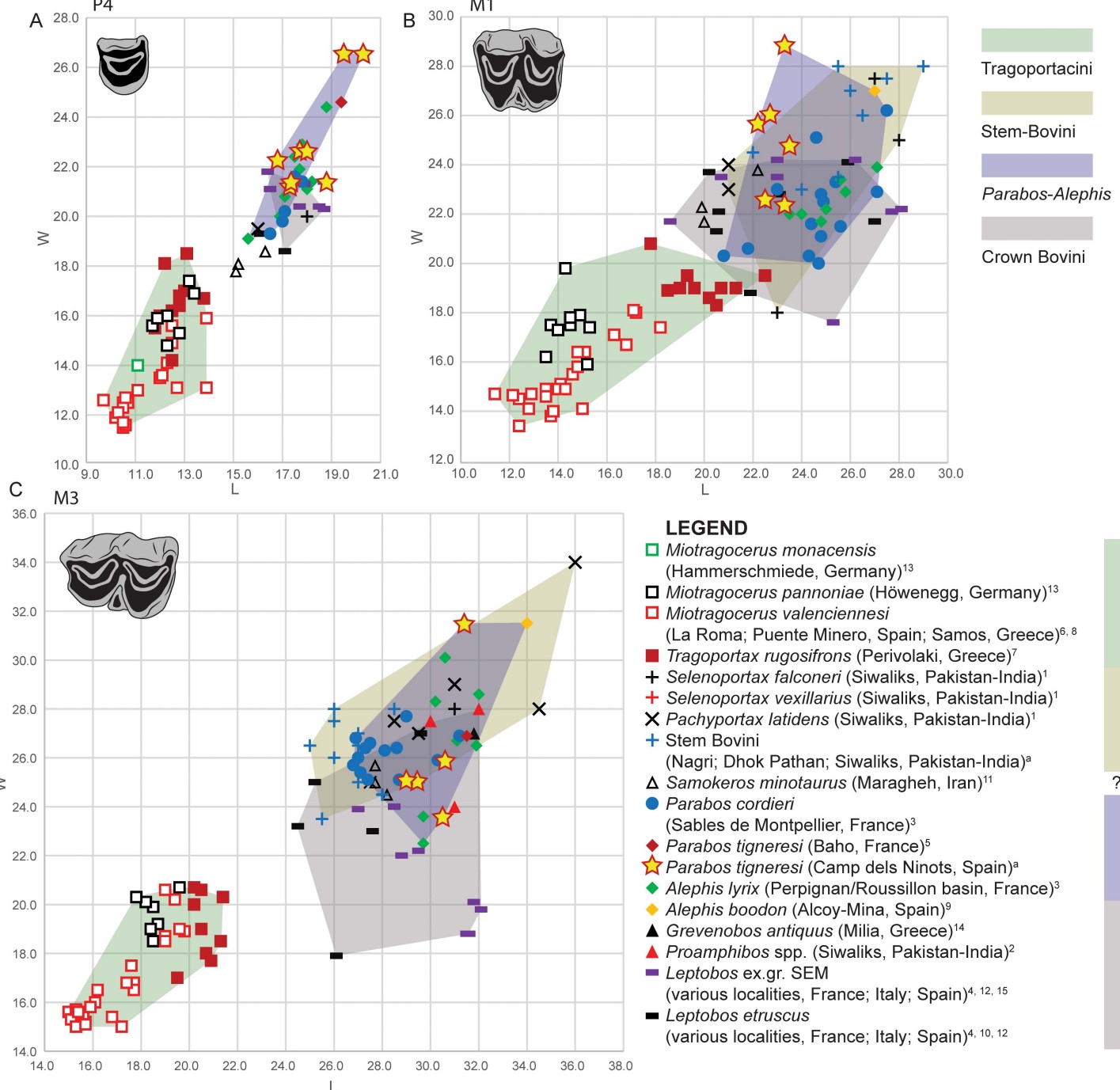

**Fig 19. Metric comparison of dental elements in *Parabos tigneresi* and other bovids from the Eurasian Neogene and Quaternary.** A, length vs width of P4; B, length vs width of M1; C, length vs width of M3.Superscript denotes data source: a, this study; 1, [79]; 2, [18]; 3, [20]; 4, [57]; 5, [52]; 6, [80]; 7, [66]; 8, [67]; 9, [53]; 10, [81]; 11, [73]; 12, [82]; 13, [38]; 14, [74]; 15, [27]. The full dataset is provided in S37 Table.

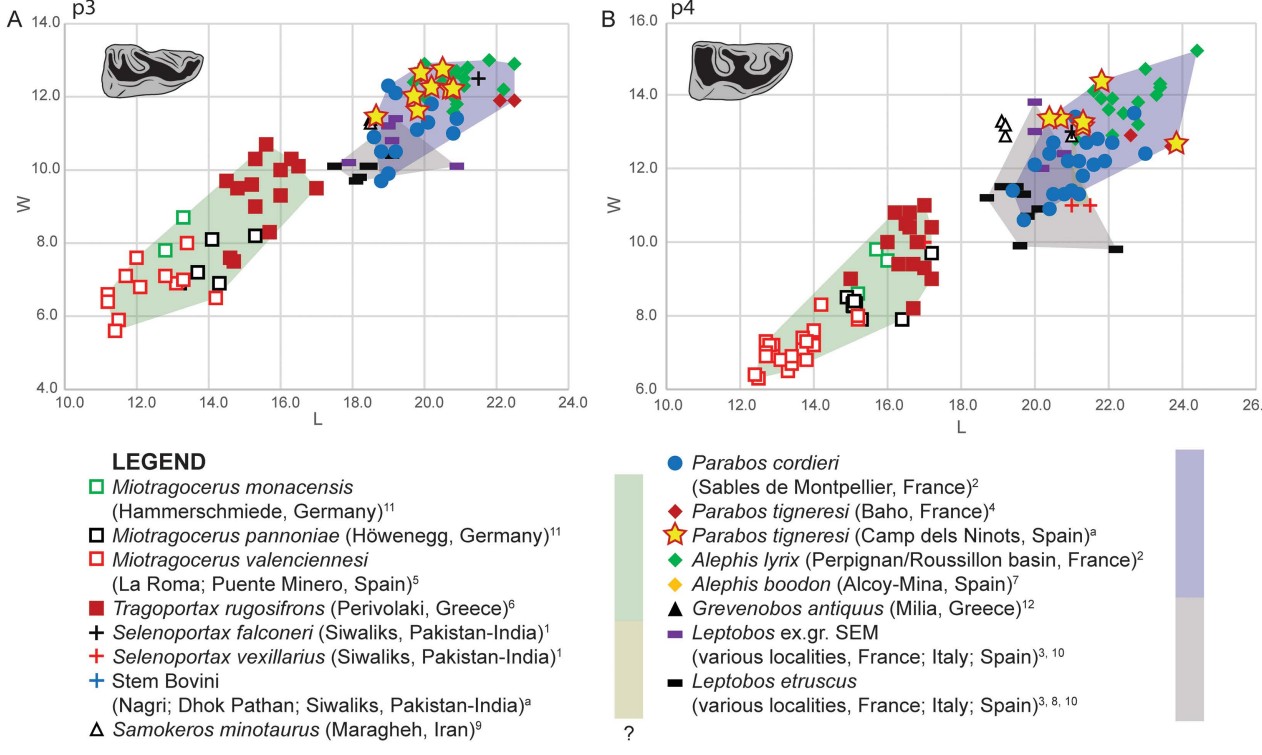

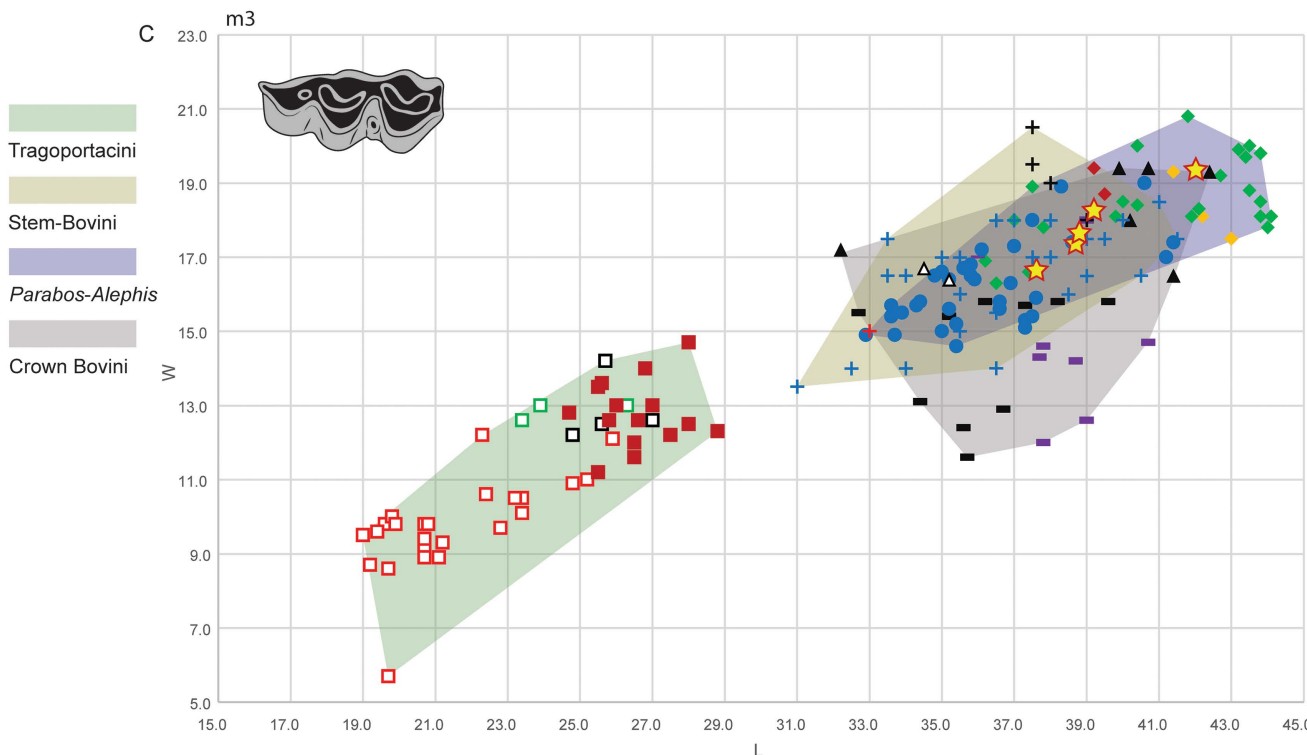

**Fig 20. Metric comparison of dental elements in *Parabos tigneresi* and other bovids from the Eurasian Neogene and Quaternary.** A, length vs width of p3; B, length vs width of p4; C, length vs width of m3. Superscript denotes data source: a, this study; 1 [79]; 2, [20]; 3, [57]; 4, [52]; 5, [80]; 6, [66]; 7, [53]; 8, [81]; 9, [73]; 10, [82]; 11, [38]; 12, [74]. The full dataset is provided in S38 Table.

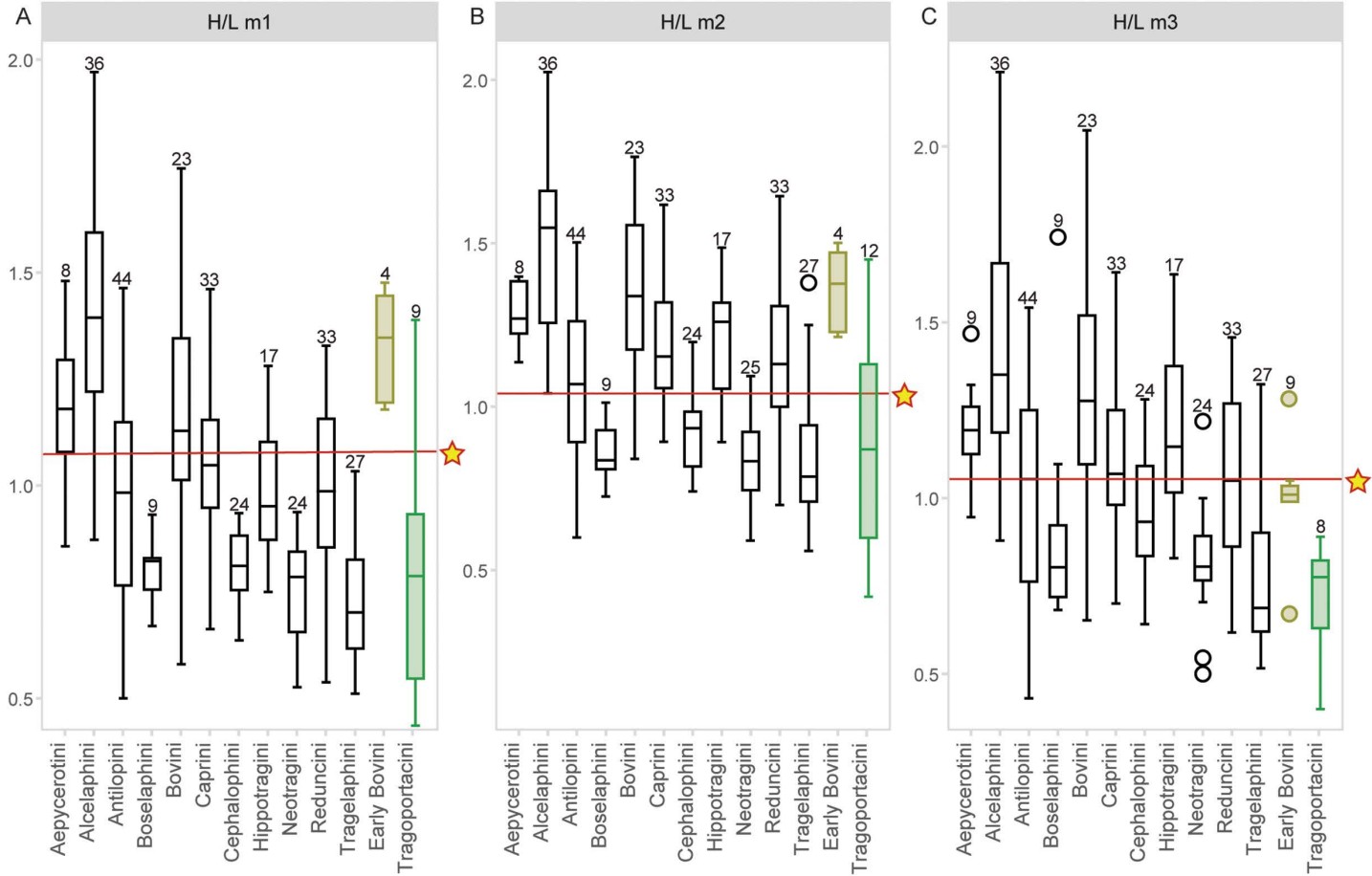

**Fig 21. Boxplot of the hypsodonty index in lower molars of *Parabos tigneresi* specimen IPHES. CN'12-B7-1 and other bovids tribes.** A, m1; B, m2; C, m3. Hypsodonty index is unworn crown height divided by crown length. Star and red line indicate values for **IPHES. CN'12-B7-1.** Coloured columns represent fossil Bovidae. Number of specimens shown above each sample. Data taken from [83] and references therein.

contrary, is always open in *P. cordieri* [20]. The hypsodonty index calculated on the least worn lower molars (specimen IPHES.CN'12-B7-1) shows that the CN bovid presents a mesodont condition, fitting between the more hypsodont tribes (e.g., Aepycerotini, Alcelaphini, Bovini) and the brachiodont Boselaphini, Neotragini and Tragelaphini (Fig 21). Interestingly, the CN individual, when compared with the two fossil groups analysed, namely Late Miocene stem Bovini from Asia and Tragoportacini (i.e., *Tragoportax* and *Miotragocerus* from Pikermi), shows values that are between the two groups in m1 and m2. On the other hand, the m3 value is slightly above the results obtained for the *Selenoportax-Pachyportax* complex (Fig 21C). In summary, the dentition of the CN bovid is more primitive than advanced Bovini and more derived than Tragoportacini, and most similar to stem or early bovins like *Parabos*, *Alephis*, *Pachyportax* and *Selenoportax*.

The limb bones of the CN bovid have similar size and proportions to *Alephis* and *Leptobos*, being larger than both Tragoportacini and *P. cordieri* (Fig 22). The metapodials are relatively elongated and lack the reduction typical of fossil and extant Bovini but do not show the extreme elongation typical of Antilopini (Fig 22B, D). On the contrary, relative metacarpal and metatarsal lengths are similar to those in Tragelaphini, Boselaphini, and Hippotragini, suggesting a plesiomorphic bovid condition. The relative lengths of the radius and tibia largely confirm this, though here the differences among bovid tribes are not so evident (Fig 22A, C).

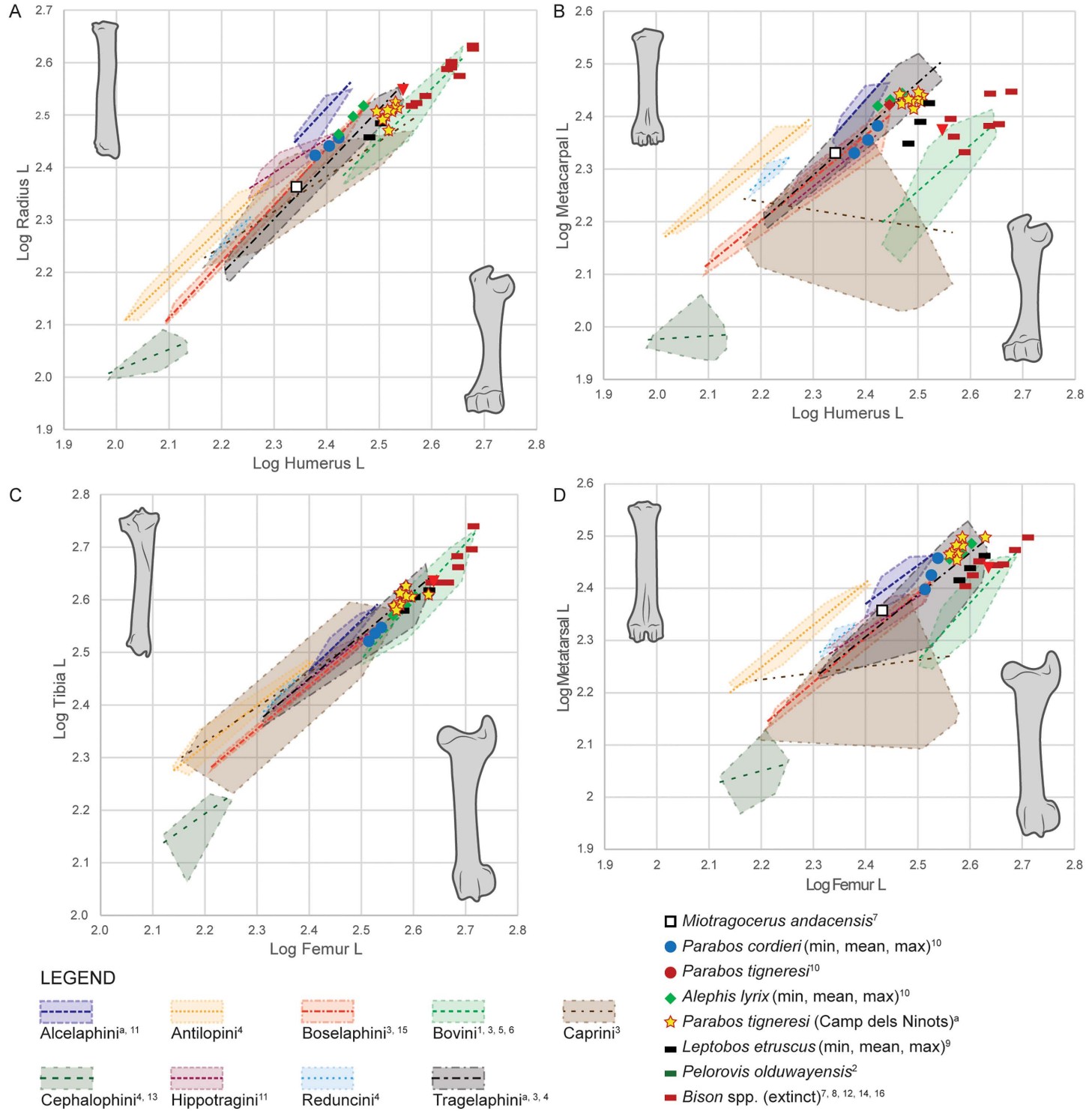

**Fig 22. Metric comparison of front and hind limb element length in *Parabos tigneresi* from Camp dels Ninots and other bovids from the Eurasian Neogene and Quaternary.** A, humerus vs radius lengths; B, humerus vs metacarpal lengths; C, femur vs tibia lengths; D, femur vs metatarsal lengths. Axes in log10. Superscript denotes data source: a, this study; 1, [84]; 2, [85]; 3, [86]; 4, [87]; 5, [88]; 6, [89]; 7, [90]; 8, [91]; 9, [56]; 10, [52]; 11, [92]; 12, [93]; 13, [94]; 14, [95]; 15, [96]; 16, [28].

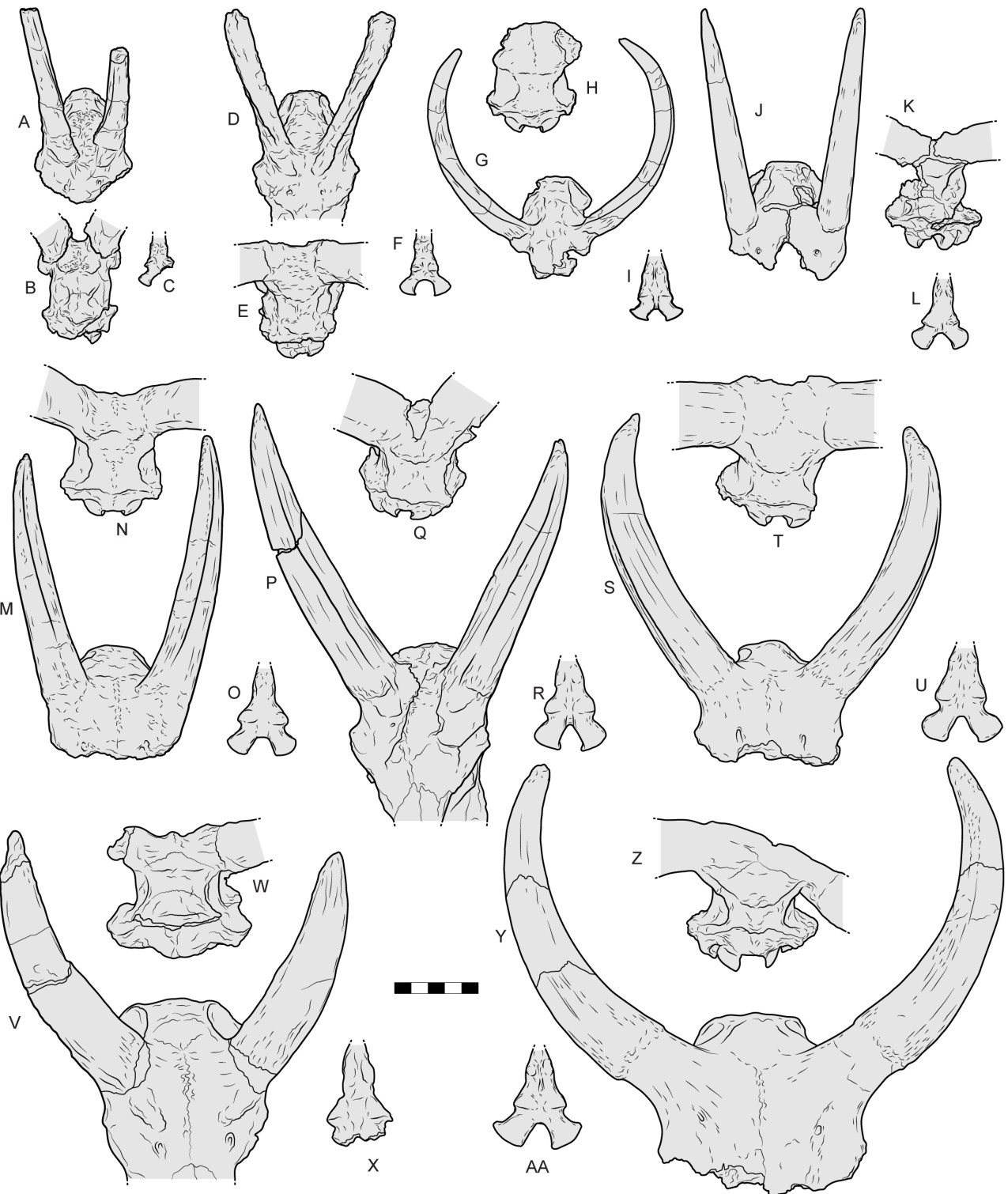

**Fig 23. Crania of various Eurasian Miocene, Pliocene and Pleistocene bovids compared with *Parabos tigneresi.*** A–C, *Miotragocerus mace-doniensis* (LGPUT DKO-53) from Ditiko (Greece) in dorsal (A) and posterodorsal (B) views, basioccipital (C); D–F, *Tragoportax amalthea* (LGPUT NIK-1688) from Nikiti (Greece) in dorsal (D) and posterodorsal (E) views, basioccipital (F); G–I, *Selenoportax vexillarius* (AMNH 19748, holotype and RIN 316) from Hasnot (Pakistan) and Tha Chang Sandpits (RIN 316, Thailand) in frontal (G, AMNH 19748) and posterodorsal (H, RIN 316) views,

basioccipital (I, RIN 316); J–L, *Parabos savelisi* (LGPUT GAS9, holotype) from Gephyra (Greece) in dorsal (J) and posterodorsal (K) views, basioccipital (L); M–O, *Parabos cordieri* (UFR FSL 40 024) from Sables de Montpellier (France) in dorsal (M) and posterodorsal (N) views, basioccipital (O); P–R, *Parabos tigneresi* (IPHES.CN'17-B14-1) from Camp dels Ninots (Spain) in dorsal (P) and posterodorsal (Q) views, basioccipital (R); S–U, *Alephis lyrix* (MNHNP, Figured in in Depéret, 1890, pl. V, fig. 4 (8), holotype) from Serrat d'en Vaquer in the Perpignan/Roussillon basin (France) in dorsal (S) and posterodorsal (T) views, basioccipital (U); V–X *Ugandax coryndonae* (AL 194−1, holotype) from Hadar formation (Ethiopia) in dorsal (V) and postero-dorsal (W) views, basioccipital (X); (Y–Y, *Leptobos etruscus* (NMBS Se792 and IGF 612, lectotype) from Senéze (France) and Upper Valdarno (Italy) in dorsal (Y, NMBS Se792) and posterodorsal (Z, IGF 612) views, basioccipital (AA, IGF612). Scale bar: 100 mm.

The CN bovid exhibits a distinctive combination of Tragoportacini and Bovini traits, similar to the Pliocene genera *Parabos* and *Alephis*. As extensively discussed above, the large size, the reduction of premolars in relation to the molars, the higher hypsodonty, the shortening of the postcornual section of the skull and the morphology of the horn-cores (e.g., presence of three well-developed keels without stepped anterior keel, weak transversal compression), differentiate the studied sample from *Tragoportax*, *Miotragocerus* and most of their Miocene tragoportacin allies [24,65,97]. At the same time the CN bovid has less derived dentition and neurocranium than *Alephis lyrix*, *Grevenobos*, *Ugandax*, *Proamphibos* and *Leptobos*, among others. The comparison with *Parabos* species reveals notable similarities which are, however, contrasted by some features derived in the direction of *Alephis*, namely the larger size, the incipient terminal horn torsion, the more divergent and slightly medially bending cores, the faint grooving of the medial surface, the larger and more massive dentition, among others. These characters ally the Camp dels Ninots with the species *Alephis tigneresi* from the Pliocene French locality of Baho confirming the preliminary attribution provided by Gómez de Soler et al. [9]. The record from CN therefore allows us to better define the morphological characters of this otherwise poorly known species. This study evidences several characters, shared with the genus *Parabos*, which clearly segregate *A. tigneresi* from *Alephis* and its type species *A. lyrix*. These characters include the weak medial bending nor significant twisting of the horn, the presence of three evident keels (i.e., clear triangular cross section of the core), the lack of deep grooves on the horncores' surface, the anteroposteriorly elongated supraoccipital and the strong parietal crests with depressed frontotemporal surface.

Given the multiple differences existing between *A. lyrix* and *A. tigneresi* and the similarities found between *Parabos* and *A. tigneresi* we suggest the relocation of *A. tigneresi*, hereinafter named *Parabos tigneresi*, from *Alephis* to *Parabos*.

## Discussion

### Reappraised taxonomy of *Parabos* and *Alephis*

Arambourg and Piveteau [16] erected the genus *Parabos* to accommodate the abundant remains of the bovine-like antelopes from the Pliocene localities of Sables de Montpellier (France) and Alcoy (Spain) which were firstly described as *Antilope cordieri* [45] and *Antilope boodon* [98] respectively. Both reallocated to *Palaeoryx* by Depéret [48], who also referred the Pliocene remains from the Roussillon/Perpignan Basin sites (France) to *Palaeoryx boodon*. Following this, Arambourg and Piveteau [16] erected the genus *Parabos* to accommodate both these species. Finally, Gromolard [15] reassigned the Roussilon *P. boodon* (but not the Alcoy sample) to a new genus and species, *Alephis lyrix*. Due to their similar morphology and geochronological distribution, *Parabos* and *Alephis* are often considered strictly, if not directly, related [15, 19, 24, 53, 55, 63, among others]. Further species have also since been assigned to these genera. Currently, *Parabos* includes the type species *Parabos cordieri* (de Christol, 1832) [45], *Parabos savelisi* Crégut-Bonnoure and Tsoukala 2017 [54], *Parabos athanasiui* Simionescu 1922 [99], *Parabos macedoniae* Arambourg and Piveteau 1929 [17] and *Parabos soriae* Morales 1984 [51]. *Alephis* includes the type species *Alephis lyrix* Gromolard 1980 [15] and *Alephis tigneresi* Michaux et al. 1991 [52]. '*Antilope*'? *boodon* Gervais, 1853 [98] which has been referred alternatively to *Parabos* [e.g., 15, 21, 51] or *Alephis* [e.g., 24, 53, 74] (Fig 2). In this study, we argue for the reassignment of *Alephis tigneresi* to *Parabos*. To help make this clearer, we next review the record of these genera.

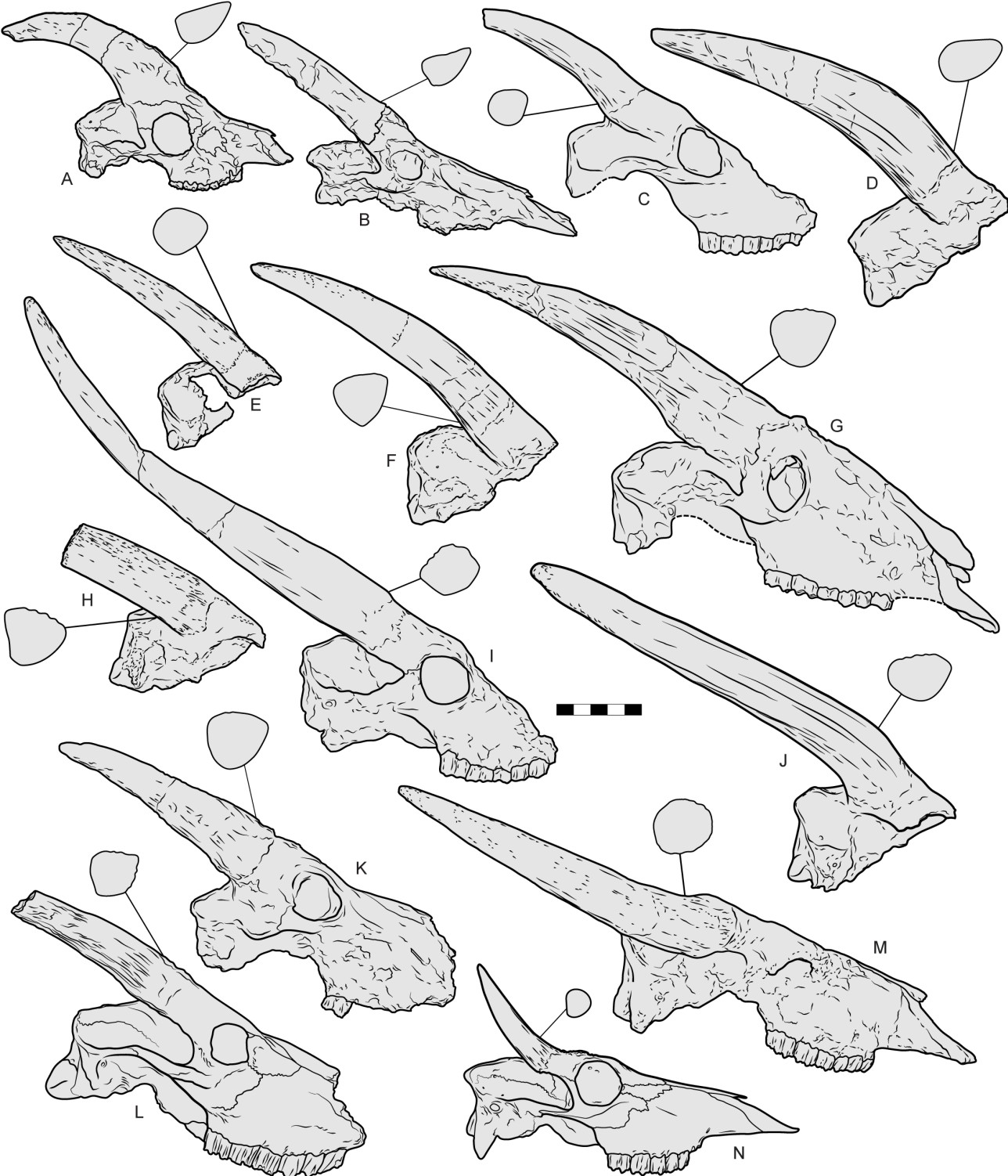

**Fig 24. Right lateral views of crania and basal horn core cross-sections of various bovids from the Eurasian Miocene, Pliocene and Pleistocene.** A, *Miotragocerus valenciennesi* (ICP IPS 5169) from Piera (Spain); B, *Tragoportax amalthea* (LGPUT NIK-1688) from Nikiti (Greece); C, "*Parabos*" *athanasiui* (No ID, mirrored) from Malusteni (Romania); D, *Samokoerus minotaurus* (IGPM PIM 99, holotype) from Samos (Greece); E, *Parabos savelisi* (LGPUT GAS 9, holotype, mirrored) from Gephyra (Greece); F, *Parabos cordieri* (UFR FSL 40 024) from Sables de Montpellier (France); G, *Parabos*

*tigneresi* (IPHES.CN'17-B14-1) from Camp dels Ninots (Spain); H, *Grevenobos antiquus* (LGPUT MIL 401a, holotype) from Milia (Greece); I, *Seleno-portax falconeri* (PRY 204) from central Myanmar; J, *Alephis lyrix* (MNHNP, Figured in in Depéret, 1890, pl. V, fig. 4 (8), holotype) from from Serrat d'en Vaquer in the Perpignan/Roussillon basin (France); K, *Ugandax coryndonae* (AL 194−1, holotype) from the Hadar formation (Ethiopia); L, *Proamphibos lachrymans* (GSI No. B. 561) from Jammu (India); M, *Leptobos etruscus* (IGF 612, lectotype) from Upper Valdarno (Italy); N, *Boselaphus tragocamelus* (MFN 108046) from South Asia. Scale bar: 100 mm.

---

*Parabos cordieri* occurs in many MN14 localities across Western Europe (Fig 2) [21]. The main traits that characterize this form include straight/slightly diverging, untwisted horncores with triangular cross-section at the base, strong anterior keel, large frontal foramina placed anteriorly to the horn pedicles, weakly raised parietal crests, elongated supraoccipital and dentition with reduced basal pillars and hypsodonty [15,18,24]. The analyses performed in this study evidence that the morphology, size and proportions of *P. cordieri* horns teeth and neurocranium differentiate it from all the known Tragoportacini and crown Bovini (previous chapter and Figs 17–20; 23; 24). The teeth show primitive features such as small stylids and low crowns, however, their size is consistently larger than all *Miotragocerus* and *Tragoportax* comparative sample, allying them with early Bovini (e.g., *Proamphibos*, *Leptobos*; Figs 19; 20). The horns are larger and less laterally compressed than the ones observed for Tragoportacini. Despite that, the presence of three well-developed keels testifies a primitive morphology which clearly separates *P. cordieri* also from crown Bovini (Figs 17A; 18C; 24F). At the same time, the postcornual section of the skull (e.g., supraoccipital area) is less elongated than in *Miotragocerus-Tragoportax* complex but still does not feature the typical squared and anteroposterior compressed morphology of later bovines (Figs 18E; 23). This unique mixture of traits makes difficult the positioning of this species (and of the whole genus) within Bovinae. Despite most of the authors refer the clade to Boselaphini or Tragoportacini its position within this group is not fully supported (see also below).

*Parabos savelisi* was first described by Crégut-Bonnoure and Tsoukala [54] on the basis of two partial neurocraniums (one male and one female) with horncores and an isolated, fragmentary metatarsal from the Greek locality of Gephyra (MN16, Fig 2). This small species of *Parabos*, is characterized by horncores with an ovoid cross-section, lacking the prominent keels that are distinctive of both *P. cordieri* and *P. tigneresi* (Fig 24E, F, G). This species is also distinguished by the least divergent and straightest horns of the genus (Fig 23J). The supraoccipital is elongated and narrow and the parietal roof stretches posteriorly so the postcornual portion of the cranium is reminiscing of the largest Tragoportacini such as *Tragoportax* (Fig 23K). The frontal foramina are located posteriorly on the internal side of the pedicle and lack of clear grooves for their accommodation. The frontal sinuses stop right after the pedicles and extend into the horncores only in one specimen (GAS 23). All these characters indicate a plesiomorphic stage for this species of *Parabos*, linking it with Boselaphini/Tragoportacini. Trifonov et al. [100] referred a well-preserved cranium from Erikdere terrace V (Turkey, MN15–16) to *P.* cf. *savelisi*, although Kostopoulos [22] suggested that this specimen could be more closely related to *Tragoportax* than to *Parabos*. Based on the position and morphology of the horns—specifically, the mediolaterally compressed base and the faint step on the anterior keel—we are inclined to agree with the latter author, though a more comprehensive description of the fossil is needed for further clarification.

*Parabos soriae* from Venta del Moro (MN13, Spain) is a poorly understood species, known from a few isolated teeth, four incomplete horncores, and postcranial elements (Fig 2) [51]. This taxon appears to be smaller and less derived than the type species of the genus [51]. Morales [51], describing the species, noted the primitive aspects of the teeth which are more reminiscent of the Tragoportacini (Boselaphini in [51]) condition (i.e., weak hypsodonty, small ribs). The two most complete horns from Venta del Moro, fall within the variability of *Parabos* spp., next to the specimens referred to *P. cordieri* and *P. savelisi* but far from the gigantic *P. tigneresi* of Baho and Camp dels Ninots (Fig 17A). The triangular section of the horncores described by Morales [51] differs from the ovoid cores of *P. savelisi* but closely resembles those of *P. cordieri* and *P. tigneresi*. Their small size allied them better to the type species of the genus. Notwithstanding these similarities, the evident short and stocky appearance of *P. soriae* clearly differentiate it from the *Parabos* from Montpellier. At the present state of the art *P. soriae* represents the oldest and the most primitive member of the genus. The paucity of

**Table 3. Comparative cranial measurements (mm) of selected *Parabos* and *Alephis* samples. *Parabos cordieri* from the type locality of Sables de Montpellier, *Parabos savelisi* from the type locality of Gephyra, *Parabos tigneresi* from the type locality of Baho, *Alephis lyrix* from the type locality in Perpignan. Measures as in Table 1. Sources: a, this study, 1, [15]; 2, [52].**

| Species | *Parabos tigneresi* CN[a] | | | | *Parabos cordieri*[a, 1] | | | | *Parabos savelisi* [a] | | | | *Parabos tigneresi* [a, 2] | | | | *Alephis lyrix* [a, 1] | | | |
|---|---|---|---|---|---|---|---|---|---|---|---|---|---|---|---|---|---|---|---|---|
| Measures | Mean (N) | Min | Max | S.D. | Mean (N) | Min | Max | S.D. | Mean (N) | Min | Max | S.D. | Mean (N) | Min | Max | S.D. | Mean (N) | Min | Max | S.D. |
| 1. | 368.4 (9) | 320.0 | 420.0 | 43.30 | 318.3 (6) | 282.0 | 345.0 | 24.60 | 265.0 (2) | 263.1 | 266.9 | – | – | – | – | – | 488.8 (4) | 445.0 | 525.0 | – |
| 2. | 332.0 (9) | 290.0 | 380.0 | 34.06 | 297.2 (2) | 292.7 | 301.7 | – | 243.7 (2) | 240.7 | 246.6 | – | 425,0 | 420,0 | 430,0 | – | 434.5 (2) | 432.0 | 437.0 | – |
| 3. | 349.8 (8) | 313.0 | 397.0 | 33.90 | 318.0 (2) | 310.0 | 326.0 | – | 263.0 (2) | 261.0 | 265.0 | – | – | – | – | – | 460.5 (2) | 459.0 | 462.0 | – |
| 4. | 57.7(4) | 52.0 | 61.9 | | 47.0 (27) | 37.4 | 56.2 | 4.47 | 40.5 (4) | 38.8 | 41.9 | – | 67,0 | – | – | – | 61.8 (10) | 53.5 | 69.0 | 4.52 |
| 5. | 68.2 (8) | 62.5 | 73.0 | 4.75 | 53.7 (27) | 47.7 | 62.5 | 4.15 | 43.8 (4) | 42.3 | 45.1 | – | 80,0 | – | – | – | 73.4 (10) | 66.5 | 83.1 | 5.53 |
| 6. | 67.2 (1) | – | – | – | 54.8 (1) | – | – | – | 33.2 (2) | 28.9 | 37.5 | – | – | – | – | – | 56.8 (1) | – | – | – |
| 7. | 49.6 (1) | – | – | – | 53.1 (1) | – | – | – | 41.3 (1) | – | – | – | – | – | – | – | 40.6 (1) | – | – | – |
| 8. | 89.8 (2) | 84.5 | 95.1 | – | 81.6 (2) | 79.2 | 84.0 | – | 70.6 (1) | – | – | – | – | – | – | – | 94.0 (3) | 88.5 | 100.8 | – |
| 9. | 38.7 (1) | – | – | – | 29.4 (2) | 28.0 | 30.7 | – | 30.1 (1) | – | – | – | – | – | – | – | 35.2 (4) | 29.5 | 39.0 | – |
| 10. | 30.8 (1) | – | – | – | 31.1 (1) | – | – | – | 32.4 (1) | – | – | – | – | – | – | – | 35.2 (3) | 35.0 | 36.0 | – |
| 11. | 165.0 (1) | – | – | – | 136.3 (2) | 132.5 | 140.0 | – | 129.3 (1) | – | – | – | – | – | – | – | 165.5 (2) | 158.0 | 173.0 | – |
| 12. | 133.0 (1) | – | – | – | – | – | – | – | 111.0 (1) | – | – | – | – | – | – | – | – | – | – | – |
| 13. | 67.7 (3) | 54.4 | 80.6 | – | 52.6 (3) | 51.5 | 53.2 | – | – | – | – | – | – | – | – | – | 58.3 (3) | 55.0 | 61.0 | – |
| 14. | 44.9 (2) | 38.2 | 51.6 | – | – | – | – | – | 33.3 (1) | – | – | – | – | – | – | – | – | – | – | – |
| 15. | 50.0 (3) | 43.9 | 54.1 | – | 46.4 (3) | 24.8 | 25.0 | – | 41.4 (1) | – | – | – | – | – | – | – | 58.4 (4) | 53.6 | 65.0 | – |
| 16. | 34.1 (1) | 33.6 | 34.5 | – | 24.9 (2) | 24.8 | 25.0 | – | 15.4 (1) | – | – | – | – | – | – | – | 32.6 (4) | 28.5 | 35.0 | – |
| 17. | 472.5 (2) | 465.0 | 472.0 | – | 390.0 (1) | – | – | – | – | – | – | – | – | – | – | – | – | – | – | – |
| 18. | 265.0 (1) | – | – | – | – | – | – | – | – | – | – | – | – | – | – | – | – | – | – | – |
| 19. | 194.3 (4) | 188.0 | 207.0 | – | – | – | – | – | – | – | – | – | – | – | – | – | – | – | – | – |
| 20. | 285.0 (1) | – | – | – | – | – | – | – | – | – | – | – | – | – | – | – | – | – | – | – |
| 21. | 189.5 (2) | 174.0 | 205.0 | – | – | – | – | – | – | – | – | – | – | – | – | – | – | – | – | – |
| 22. | 207.2 (2) | 207.3 | 207.0 | – | – | – | – | – | – | – | – | – | – | – | – | – | – | – | – | – |
| 23. | 43.0 (1) | – | – | – | – | – | – | – | – | – | – | – | – | – | – | – | – | – | – | – |

*(Continued)*

Table 3. (Continued)

| Species | Parabos tigneresi CN[a] | | | | Parabos cordieri[a,1] | | | | Parabos savelisi[a] | | | | Parabos tigneresi[a,2] | | | | Alephis lyrix[a,1] | | | |
|---|---|---|---|---|---|---|---|---|---|---|---|---|---|---|---|---|---|---|---|---|
| Measures | Mean (N) | Min | Max | S.D. | Mean (N) | Min | Max | S.D. | Mean (N) | Min | Max | S.D. | Mean (N) | Min | Max | S.D. | Mean (N) | Min | Max | S.D. |
| 24. | 270.0 (2) | 265.0 | 275.0 | – | – | – | – | – | – | – | – | – | – | – | – | – | – | – | – | – |
| 25. | 277.5 (2) | 270.0 | 285.0 | – | – | – | – | – | – | – | – | – | – | – | – | – | – | – | – | – |
| 26. | 318.5 (2) | 313.0 | 324.0 | – | – | – | – | – | – | – | – | – | – | – | – | – | – | – | – | – |
| 27. | 146.0 (1) | – | – | – | – | – | – | – | – | – | – | – | – | – | – | – | – | – | – | – |
| 28. | 131.9 (3) | 125.8 | 137.0 | – | – | – | – | – | – | – | – | – | – | – | – | – | – | – | – | – |
| 29. | 51.2 (3) | 48.7 | 54.9 | – | 47.0 (1) | – | – | – | – | – | – | – | – | – | – | – | – | – | – | – |
| 30. | 56.9 (3) | 50.6 | 64.2 | – | – | – | – | – | – | – | – | – | – | – | – | – | – | – | – | – |

remains referred to this species, however, severely hampered its diagnosis and does not allow to infer more on its relationship with other members of the genus.

*Parabos athanasiui* is represented by an almost complete cranium, accompanied by a few postcranial bones from Malusteni (Romania, MN 15a), and sparse remains from other Romanian localities (Fig 2) [20,62,101–104]. Based on the illustrations provided by Simionescu [62], the Malusteni skull exhibits notable differences from the typical *Parabos* morphology. These include extremely reduced horn cores, their position relative to the orbit, their curvature, and the absence of a prominent keel (Fig 24C). Despite these distinctions, the dentition and horn size are comparable, though not entirely overlapping, with those of *Parabos* species (Fig 17A, B). This combination of conflicting features has been recognized by Radulescu et al. [103,104], who placed the generic attribution of "*Parabos*" *athanasiui* in quotation marks. Given its pronounced divergence from the observed morphology of *Parabos*, we recommend excluding this taxon from the genus *Parabos* until a thorough reassessment of the material assigned to this taxon is conducted.

*Parabos macedoniae*, firstly described from the Pliocene of Greece by Arambourg and Piveteau [17], is based on few teeth and postcranial bones of a large-sized bovid found in the "Falaise de Karabouroun" (= Megalo Emvolon; Greece, MN15; Fig 2). The lack of cranial material severely affected the definition of *P. macedoniae* which has been considered a dubious member of the genus [17,20,22,54]. Arambourg and Piveteau [17] provide a basic description of the teeth that appear to be relatively primitive and resembling to the ones of *P. cordieri* (e.g., rugose enamel, absence of cement, poorly developed ribs and pillars). The size and proportions of the available limb bones do not vary from the variability observed in other Pliocene large sized bovids such as *A. lyrix*, *P. tigneresi* and *G. antiquus* [17,20,74]. The extremely scanty material referred to this species does not allow to properly define any diagnostic features and hence its taxonomic status. The nomen dubium status for the taxon is hereby proposed. The Megalo Emvolon remains are here referred to Bovinae indet.

*Alephis lyrix* is a large bovid from the MN14–15 of Western Europe (Fig 2), characterized by several derived features which suggest its belonging to an early lineage of Bovini [15,19,22–24,53]. Its resemblance to the Asian genera *Pachyportax* and *Selenoportax* is evident in the strong anterior keel, and spiralization of the horns although the European genus displays more advanced traits, such as deeper grooves, weaker mediolateral compression of the cores, and a less depressed postcornual skull roof (Figs 17A; 23; 24) [15,24,44,77]. Apart from *Parabos*, the Late Miocene *Samokeros minotaurus* from Greece and Iran shares the closest morphological similarities with *A. lyrix*. Both taxa exhibit massive, medially curved horncores and a reduced, box-shaped neurocranium [97]. However, *Samokeros* differs from *Alephis*

**Table 4. Comparative dental measurements (mm) of selected *Parabos* and *Alephis* specimens. *Parabos cordieri* from the type locality of Sables de Montpellier, *Parabos tigneresi* from the type locality of Baho, *Alephis boodon* from the type locality of Alcoy, *Alephis lyrix* from the type locality in the Perpignan basin. Measures as in Table 2. Sources: a, this study; 1, [15]; 2, [52]; 3, [53].**

| Species | *Parabos tigneresi* CN[a] | | | | *Parabos cordieri*[a,1] | | | | *Parabos tigneresi*[a,2] | | | | *Parabos boodon*[a,3] | | | | *Alephis lyrix*[a,1] | | | |
|---|---|---|---|---|---|---|---|---|---|---|---|---|---|---|---|---|---|---|---|---|
| Measures | Mean (N) | Min | Max | S.D. | Mean (N) | Min | Max | S.D. | Mean (N) | Min | Max | S.D. | Mean (N) | Min | Max | S.D. | Mean (N) | Min | Max | S.D. |
| 1 | 140.5 (8) | 132.4 | 148.1 | 5.19 | 129.5 (2) | 129 | 130 | – | – | – | – | – | – | – | – | – | 136.1 (2) | 130 | 142.2 | – |
| 2 | 59.6 (8) | 55.7 | 64.4 | 2.67 | – | – | – | – | – | – | – | – | – | – | – | – | – | – | – | – |
| 3 | 84.7 (9) | 79.7 | 90.3 | 3.95 | – | – | – | – | 83.6 (1) | – | – | – | – | – | – | – | – | – | – | – |
| 4 | 19.6 (14) | 17.9 | 22.4 | 1.19 | 18.2 (4) | 16.8 | 19.2 | – | – | – | – | – | 22.0 (1) | – | – | – | 18.6 (8) | 16.5 | 19.6 | 1.02 |
| 5 | 16.4 (8) | 13.7 | 17.8 | 1.23 | 13.1 | 10.3 | 15.3 | 2.13 | – | – | – | – | 18.0 (1) | – | – | – | 16.5 (7) | 15.1 | 17.7 | 0.88 |
| 6 | 19.0 (17) | 16.6 | 20.4 | 1 | 18.3 (5) | 17.2 | 21 | 1.41 | – | – | – | – | – | – | – | – | 20.4 (9) | 19.6 | 21.3 | 1.25 |
| 7 | 19.8 (10) | 13.3 | 24.6 | 4 | 17.2 (7) | 16.4 | 18.1 | 0.61 | – | – | – | – | – | – | – | – | 19.8 (9) | 18 | 21.3 | 1.25 |
| 8 | 18.4 (14) | 16.8 | 20.3 | 1.14 | 17.8 (6) | 16.4 | 17.8 | 0.45 | 19.4 (1) | – | – | – | – | – | – | – | 17.6 (10) | 15.5 | 18.8 | 0.88 |
| 9 | 23.0 (8) | 21.1 | 26.5 | 2.23 | 20.6 (6) | 19.3 | 21.6 | 0.88 | 24.6 (1) | – | – | – | – | – | – | – | 21.4 (10) | 19 | 24.3 | 1.42 |
| 10 | 22.8 (13) | 21 | 24.2 | 0.87 | 22.5 (15) | 19 | 26.1 | 1.86 | – | – | – | – | 27.0 (1) | – | – | – | 23.1 (8) | 21.7 | 24.8 | 1.07 |
| 11 | 25.0 (6) | 22.3 | 28.8 | 2.43 | 24.7 (15) | 20.9 | 27.3 | 1.54 | – | – | – | – | 27.0 (1) | – | – | – | 24.6 (9) | 22.5 | 27.1 | 1.47 |
| 12 | 27.6 (15) | 22 | 31.1 | 2.18 | 26.6 (22) | 24.9 | 30.6 | 1.23 | – | – | – | – | – | – | – | – | 27.7 (16) | 119.5 | 30.2 | 2.54 |
| 13 | 26.9 (6) | 25 | 28.8 | 1.33 | 26.1 (20) | 24.8 | 28 | 0.97 | – | – | – | – | – | – | – | – | 26.7 (14) | 24 | 29,2 | 1.7 |
| 14 | 30.8 (15) | 27.8 | 34.9 | 2.16 | 28.0 (16) | 25.7 | 31.1 | 1.37 | 31.5 (1) | – | – | – | 34.0 (1) | – | – | – | 30.5 (15) | 26.5 | 34.5 | 2.16 |
| 15 | 26.4 (6) | 23.5 | 31.4 | 2.78 | 26.1 (16) | 23.8 | 27.5 | 0.9 | 26.9 (1) | – | – | – | 31.5 (1) | – | – | – | 26.3 (10) | 22.5 | 30.1 | 2.19 |
| 16 | 152.8 (9) | 145.8 | 165 | 5.59 | 139.1 (11) | 133.3 | 148.5 | 4.88 | 151.6 (1) | – | – | – | – | – | – | – | 153.2 (13) | 144.2 | 165.9 | 6.38 |
| 17 | 57.3 (9) | 54.1 | 63.1 | 3.12 | – | – | – | – | 60.4 (1) | – | – | – | – | – | – | – | – | – | – | – |
| 18 | 93.1 (11) | 88.6 | 99.9 | 3.65 | – | – | – | – | 91.2 (1) | – | – | – | – | – | – | – | – | – | – | – |
| 19 | 15.8 (13) | 13.3 | 21 | 2.01 | 15.6 (7) | 14.6 | 16.7 | 0.61 | 17.9 (2) | 17.9 | 17.9 | – | – | – | – | – | 15.4 (4) | 14.8 | 16.2 | – |
| 20 | 10.4 (8) | 8.8 | 16 | 2.46 | 8.9 (7) | 7.7 | 10.4 | 0.83 | 9.9 (2) | 9.7 | 10 | – | – | – | – | – | 8.6 (4) | 7.9 | 8.9 | – |
| 21 | 19.7 (14) | 16.1 | 21.2 | 1.41 | 19.5 (12) | 18.5 | 20.8 | 0.78 | 22.3 (2) | 22.1 | 22.5 | – | – | – | – | – | 20.9 (18) | 19.5 | 22.4 | 0.79 |
| 22 | 12.2 (9) | 11.4 | 12.9 | 0.49 | 11.1 (14) | 9.7 | 12.3 | 0.76 | 11.9 (2) | 11.9 | 11.9 | – | – | – | – | – | 12.3 (19) | 11 | 13 | 0.52 |
| 23 | 21.3 (13) | 18.2 | 24 | 1.48 | 20.9 (23) | 10.6 | 13.4 | 0.76 | 23.2 (2) | 22.6 | 23.7 | – | – | – | – | – | 22.5 (16) | 21 | 25 | 1.07 |

*(Continued)*

| Species | Parabos tigneresi CN[a] | | | | Parabos cordieri[a,1] | | | | Parabos tigneresi[a,2] | | | | Parabos boodon[a,3] | | | | Alephis lyrix[a,1] | | | |
|---|---|---|---|---|---|---|---|---|---|---|---|---|---|---|---|---|---|---|---|---|
| Measures | Mean (N) | Min | Max | S.D. | Mean (N) | Min | Max | S.D. | Mean (N) | Min | Max | S.D. | Mean (N) | Min | Max | S.D. | Mean (N) | Min | Max | S.D. |
| 24 | 13.6 (8) | 12.7 | 15.1 | 0.7 | 20.9 (23) | 18.9 | 22.7 | 0.93 | 12.8 (2) | 12.6 | 12.9 | – | – | – | – | – | 14.0 (13) | 12.7 | 15.1 | 0.68 |
| 25 | 23.9 (13) | 21.2 | 28.7 | 1.97 | 21.3 (30) | 17.8 | 25.3 | 1.49 | 22.5 (2) | 22.5 | 22.5 | – | – | – | – | – | 23.8 (21) | 19.7 | 28 | 2.1 |
| 26 | 16.5 (7) | 14.7 | 21.7 | 2.47 | 14.9 (34) | 13 | 17.1 | 0.95 | 18.5 (2) | 18.5 | 18.5 | – | – | – | – | – | 17.0 (25) | 14.7 | 21 | 1.39 |
| 27 | 26.7 (11) | 24.5 | 29.3 | 1.47 | 25.3 (28) | 21.9 | 28.3 | 1.32 | 29.0 (2) | 28.8 | 29.2 | – | 32.0 (1) | – | – | – | 27.9 (28) | 24.1 | 30.6 | 1.71 |
| 28 | 18.1 (5) | 16.4 | 19.3 | 1.19 | 16.9 (29) | 14.6 | 20 | 1.29 | 19.9 (1) | 19.5 | 20.2 | – | 18.0 (1) | – | – | – | 18.9 (27) | 17.2 | 21.8 | 1.23 |
| 29 | 40.8 (14) | 37.6 | 44.1 | 2.16 | 36.1 (34) | 32.7 | 41.5 | 1.92 | 39.4 (2) | 39.2 | 39.5 | – | 42.2 (3) | 41.4 | 43 | – | 40.6 (21) | 35.7 | 44 | 2.7 |
| 30 | 18.2 (6) | 16.6 | 20.3 | 1.37 | 16.1 (41) | 14.7 | 19 | 1.06 | 19.1 (2) | 18.7 | 19.4 | – | 18.3 (3) | 17.5 | 19.3 | – | 18.6 (24) | 16 | 20.7 | 1.18 |

for the keeless, distally flattened horncores, narrower angle of horn emergence, posteriorly placed, unsunken frontal foramina and more primitive dentition (Fig 24D, J) [36,73,97]. The Asian and African counterparts of *A. lyrix* could be considered *Proamphibos lachrymans* and *Ugandax demissum* respectively (Figs 23S–X; 24J–L). These three species are indeed sharing numerous Bovini apomorphies characterizing the size and the morphology of the horncores and dentition, although both the Asian and African taxa differ from *A. lyrix* by their slightly more advanced features [18,63]. Their presence during a timespan contemporaneous with or even preceding *A. lyrix* challenges any direct ancestor-descendant relationship [24,69]. Currently, the classification of *Alephis* within the Bovini is widely accepted. However, its relatively primitive morphology places it among the most basal forms within the tribe [2, 15, 23, 24, among others]. Its relationship with *Parabos* and *Samokeros* is still unclear although the affinity with the former genus seems stronger (see below).

*Alephis boodon* was first described by Gervais [98] as *Antilope? boodon*, based on scant dental and postcranial remains from Alcoy-Mina (MN14–15, Spain, Fig 2). The attribution to *Antilope*, although uncertain, was given due to the similarities found between the Spanish and the French remains ascribed to *Antilope cordieri* by de Christol [45]. Both taxa were subsequently moved in *Palaeoryx* [48] and then lumped in *Parabos* [16]. Gromolard [15], in her reappraisal of *Parabos* established a new genus (i.e., *Alephis*) and separated the Roussillon specimens from the type material of *P.? boodon*, creating a new taxon which became the type species of the genus, namely *Alephis lyrix*. The Iberian remains, however, were left in open nomenclature and largely forgotten by scholars, including Michaux et al. [52], who revisited the taxonomy of the *Parabos-Alephis* group. Montoya et al. [53] reassigned *Parabos? boodon* to *Alephis* on the basis of its similarities with *A. lyrix*. The authors however remark that the extremely scant type material hinder any tentative of a proper taxonomic attribution. We here confirm that the dentition of *A. boodon* are similar in morphology and size to *A. lyrix* as well as *P. tigneresi* and lack important distinctive features that could facilitate differentiation. In the absence of horn cores or other more diagnostic materials, we propose that *A. boodon* is a nomen dubium and refer the Alcoy Mina material to Bovinae indet. cf. *Parabos* vel *Alephis*.

*Alephis tigneresi* was erected by Michaux et al. [52] on the basis of few remains of a large bovid from Baho, a village in the Roussillon/Perpignan Basin (MN 14, France, Fig 2). The material, likely referrable to a single individual, includes two horncores, a fragment of the skull (i.e., portion of the right frontals, parietals and temporals), several teeth and postcranial bones. The remains of Camp dels Ninots, already attributed to this taxon by Gómez de Soler et al. [9] and

hereby confirmed, represents the richest and most complete record for this species and the whole genus. The analyses performed allowed us to recognize several plesiomorphic traits which ally *P. tigneresi* to *Parabos* (see previous chapter) and therefore compel the moving of the species from *Alephis* to the latter genus. At the same time, both the Iberian and French record, referred to *P. tigneresi* are characterized by intermediate morphology between *Parabos cordieri* –"primitive" condition– and *Alephis lyrix* –"derived" condition–(Figs 23P–U; 24G, J). The intermediate characters between *P. cordieri* and *A. lyrix* of *P. tigneresi* are hereby provided. *Parabos tigneresi* features posteriorly directed horncores with smaller diverging angle compared to *Alephis* but larger than the one shown by the subparallel cores of *P. cordieri*. The incipient distal torsion and the weak medial furrows of *P. tigneresi* cornual appendages separate it from both *A. lyrix* (strongly spiralized and grooved horns) and *P. cordieri* (straight, untwisted and smooth horns). Although the horn keels of *P. cordieri* resemble the ones of *P. tigneresi*, in the latter taxon the posterolateral keel is blunter. This character is amplified in *A. lyrix*, so much that this keel disappears. The development of the supraoccipital in *P. tigneresi* appears to be intermediate between the ones of *P. cordieri* (longer) and *A. lyrix* (shorter). Lastly *P. tigneresi* present a dentition similar to the one of *Alephis* (large teeth, strong basal pillars, presence of cement) which separates it from the more primitive *Parabos cordieri*. This evidence suggests that *P. tigneresi* may either represent an intermediate form between *P. cordieri* and *A. lyrix* as part of a monophyletic group or reflect independent convergences toward Bovini morphology. A phylogenetic analysis is needed to test whether *Alephis* could have originated from a derived *Parabos* similar to *P. tigneresi*.

## The position of *Parabos* and *Alephis* within Bovinae

It is generally accepted that Bovini evolved from Miocene 'Boselaphini' [5,18,23,79, among others]. Features such as large size, divergent horns, and high crowned teeth with large basal pillars, among others, discriminate taxa such as *Selenoportax* and *Pachyportax* from *Miotragocerus*, *Tragoportax* and allies, traditionally placed within 'Boselaphini', and place them on the line of Bovini [18,78,79]. Many of these characters are also present in *Parabos* and *Alephis*, which could represent either stem Bovini, or 'boselaphins' that independently converged on Bovini-like morphology.

Gromolard [15], presenting an emended diagnosis of *Parabos*, emphasized the primitive aspects of the genus and referred it to Boselaphini (Tragoportacini in this work). Over time, most of the scholars have supported this classification [e.g., 20,24,54]. Indeed, the type species *Parabos cordieri* differs notably from the contemporaneous African taxa which had already developed clear derived features in the direction of Bovini [24,69]. For example, *Ugandax* sp. from the Late Miocene of the Middle Awash (Ethiopia) and *Ugandax demissum* from the Early Pliocene of Langebaanweg (South Africa) show advanced dentition, stouter and more divergent horn cores, reduced neurocranium and a broader occipital surface, than *Parabos* [24,63,105]. At the same time, this work shows that *Parabos* does show characters that clearly distinguish it from fossil 'Boselaphini' like *Tragoportax* and *Miotragocerus*, and suggest, instead, affinity to (or convergence with) purported stem bovins from Miocene of Asia like *Selenoportax* and *Pachyportax* (Figs 17–24) These include the absence of sculpturing on the frontoparietal surface, the reduction of the premolar row relative to the molar row, the strong ribs and basal pillars on molars the incipient hypsodonty, the shortening of the neurocranium and the large size. Therefore, while *Parabos* might not belong in Bovini, it also does not seem to belong in Tragoportacini or 'Boselaphini', as already partially postulated by Montoya et al. [53]. In a phylogenetic analysis, Geraads [23] recovered *P. cordieri* either outside or at the very base of Bovini, confirming the similarities we and others have noted and suggesting, however, that *Parabos* might be part of 'Boselaphini' (here Tragoportacini); a view shared also by Gentry [2] and Bibi [24]. If this is true, these taxa would have independently and parallelly developed Bovini traits during the latest stages of Neogene. Nonetheless, some results in the phylogenetic analysis performed by Bibi [24] points toward the inclusion of both *Parabos* and *Alephis* in stem Bovini. Finally, Montoya and colleagues [53] suggest a more cautious approach evidencing that referring these genera to either one of these tribes is extremely difficult. To make matters more complicated, while many authors agree that *Alephis* is an early Bovini and *Parabos* is not [15,20,21,24], others have proposed that *Alephis* evolved from *Parabos* [53,63]. If the latter statement is true, the separation of the two genera in different tribes would be difficult to justify.

The new CN material shed light on the morphology of *Parabos tigneresi*, a species that until now was classified within the genus *Alephis* [52] suggesting a close relationship between the two genera. Indeed, although the stratigraphic ranges of *Parabos* and *Alephis* remains uncertain, this does not rule out an ancestor-descendant relationship. As demonstrated in this work, key characters shared by these two taxa and other Eurasian genera commonly referred to early Bovini/stem-Bovini (e.g., *Pachyportax*, *Selenoportax*, *Grevenobos*) address to a relationship between them and support their inclusion in Pan-Bovini [24,44,69,74,77]. This would imply the presence of a common ancestor which evolved in Eastern Palaearctic and, somewhat during Miocene, reached Europe, giving rise to the *Parabos-Alephis* lineage which, however, preserved relatively primitive features compared with the roughly coeval African early Bovini (i.e., *Ugandax*). In contrast, the highly primitive morphology of *P. soriae* and *P. savelisi* suggests a close relationship with the Tragoportacini, indicating that the genus may have originated from a boselaphin-like lineage at the Mio-Pliocene boundary and evolved convergently toward the Bovini trough the Ruscinian. The presence of multiple Bovini-like offshoots in Europe during the Late Miocene to Early Pliocene should not be ruled out (see also *Samokeros*). The widely recognized recurrence of homoplasies in Bovidae complicates this issue [2].

On the light of the current knowledge, *Parabos* and *Alephis*: might be either a lineage diverging from the stem group of Pan-Bovini (sensu [69]) or a late surviving lineage of Tragoportacini morphologically convergent with early Bovini (Fig 25). A third hypothesis sees *Parabos* and *Alephis* not part of the same lineage and independently nested within the two tribes. A comprehensive and updated phylogenetic analysis of Mio-Pliocene Bovinae is of the utmost importance to shed light on the first evolutionary steps of Bovini and the potential presence of ghost stem groups.

## Palaeoenvironment and palaeoecology of *Parabos* and *Alephis*

By the end of the Miocene, the stable dry conditions that had facilitated the expansion of the so-called Pikermian chronofauna across Eurasia and Africa were replaced by a more humid climate marked by regional differentiations, which culminated in widespread reforestation, particularly in Mediterranean Europe [106,107 among others]. At the onset of the Early Pliocene (ca. 5.33–3.6 Ma), the increased seasonality and humidity fragmented the vast open landscapes of the Western Palaearctic. This transition triggered the final decline of the Pikermian chronofauna and severed long-lasting ecological connections between Africa and Eurasia [106,107]. In western Mediterranean Europe, the Late Miocene bovid community, primarily composed of Tragoportacini and Antilopini, was already poorer than that of southeastern Europe, which, in contrast, was characterized by a high diversity and strong affinities with assemblages ranging from Anatolia to Afghanistan [22,108,109]. Possibly from the latest Miocene (MN13), but certainly by the Early Pliocene, *Parabos* and *Alephis* became the sole large-sized bovid in Mediterranean Europe, especially in its western regions (Fig 2).

The revised age of Camp dels Ninots at ca. 4.4 Ma places it within the Early Pliocene warm period. Fossil pollen indicates that the vegetation in the region of Camp dels Ninots included many extinct thermophilus and hygrophilous species, as well as temperate taxa with Mediterranean affinities and high-altitude conifers [10,110]. The immediate surroundings of the lake at Camp dels Ninots were dominated by a riparian forest, characterized by plant species adapted to wet conditions, whereas the regional vegetation around the site was composed of a mixture of subtropical and temperate elements, reflecting a transitional environment [10,110,111]. The CN paleoenvironment was relatively warm and humid, and characterized by small-scale cyclical changes between hygrophilous and xerophilous plants, correlated to precipitation oscillation [10,110].

Kingdon [112] suggested that Bovini ancestors were forest-dwelling browsers with simple and mesodont/brachydont dentition. Bibi [69] proposed that the transition towards more open habitats marked by seasonal drying during the Middle to Late Miocene led to a dietary adaptation that produced the first stem bovins. Teeth showing higher crowns, increases pillars and occlusal complexity appear as early as 8.9 Ma, in the Siwaliks (Pakistan, India) and in the Irrawaddy beds of central Myanmar [44,69]. The westward spreading of $C_4$ grass during the later Miocene led to the expansion of these early Bovini in the Middle East and Africa [69]. In contrast, the first bovin-like dentitions in Western Europe only appear

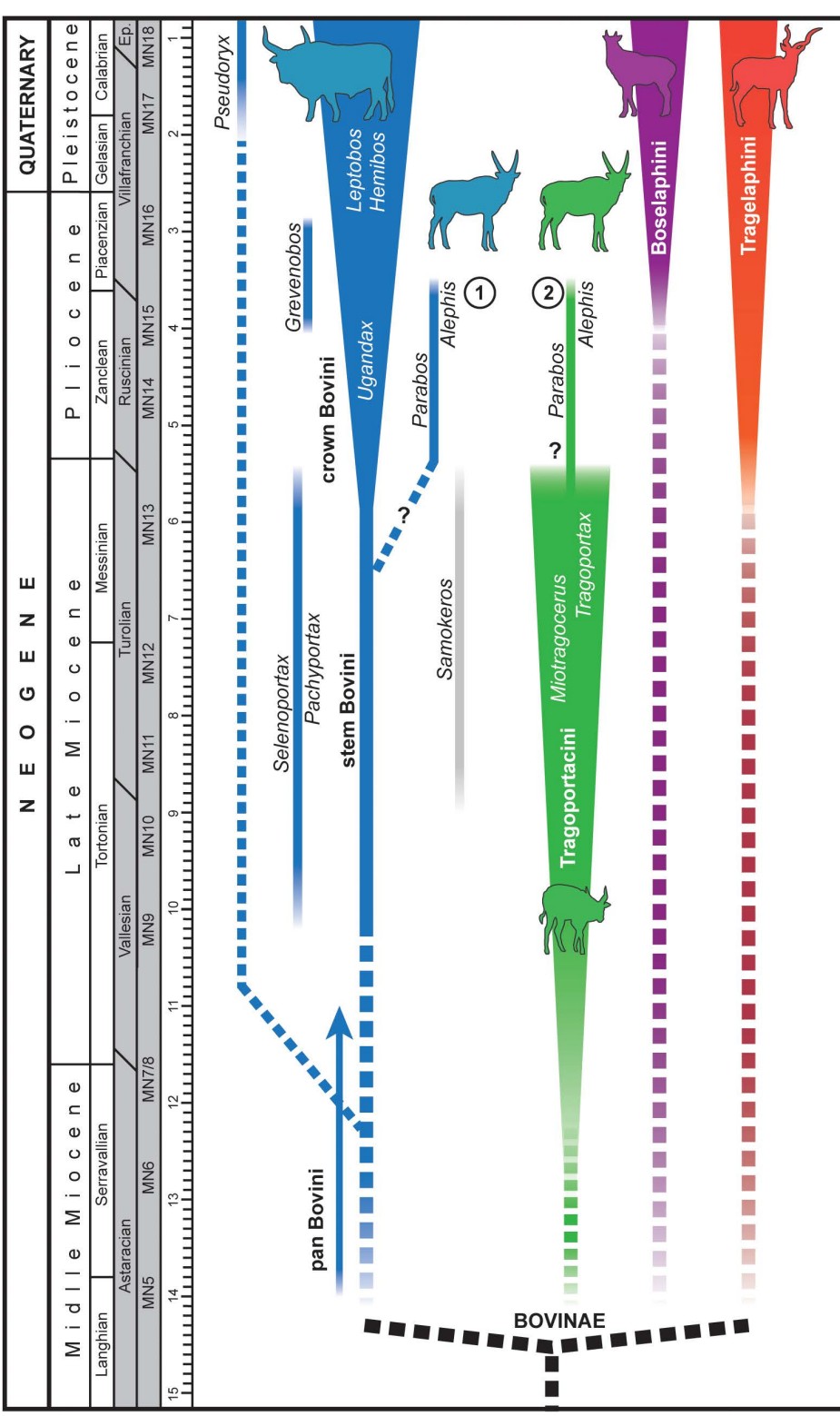

**Fig 25. Suggested phylogeny of Bovinae showing the possible relationships and chronological ranges of *Parabos*, *Alephis* and other Mio-cene to Pliocene bovids.** Alternate scenarios: 1, *Parabos* and *Alephis* are part of a lineage diverging from the stem group of Pan-Bovini; 2, *Parabos* and *Alephis* represent a late surviving lineage of Tragoportacini.

in the Pliocene, with *Parabos* and *Alephis*. Whether the more humid and forested habitats of Western Europe [106,113] were responsible for the delay in the appearance of these traits in the continent is still to be demonstrated. Crown Bovini with fully developed bovin dentition appears in Africa and South Asia by the earliest Pliocene (e.g., *Ugandax coryndonae*, *Proamphibos* spp.) while in Europe this is only with the arrival of the first *Leptobos* from the Eastern Palaearctic at the very end of Pliocene [22,28 and reference therein]. According to the hypsodonty index proposed by Fortelius et al. [29] for large mammals the values found for CN *P. tigneresi* corresponds to a mesodont dentition (i.e., 0.8–1.2). The index computed for the lower molars of the specimen IPHES.CN'12-B7-1 is below the values typical for extant hypsodont tribes such as Alcelaphini, Caprini and Bovini, and higher than the ones of more brachiodont bovids including Boselaphini, Cephalophini and Tragelaphini (Fig 21). The same index calculated for early Bovini from Asia provided values that fit within the variation found in Bovini, higher than in the specimen from CN. *Tragoportax* and *Miotragocerus* from the Miocene of Pikermi, show low crown dentition, especially in m1 and m3, similar to the one observed in Boselaphini and Tragelaphini (Fig 21). The intermediate position of the CN specimen between stem bovins and tragoportacins remarks the presence of an incipient hypsodonty in *P. tigneresi* which, however, still retain a certain degree of plesiomorphic characteristics.

According to Köhler [108], *Parabos cordieri* was well-adapted to wooded, humid habitats, exhibiting limb bone proportions and body mass like those of modern Tragelaphini and some Bovini. Its relatively short radius and tibia suggest a slow-paced mode of locomotion suited for forested environments, rather than cursorial adaptations for open landscapes which, on the contrary, are commonly associated with longer appendicular elements [e.g., 108, 114, 115, 116]. Köhler [108] also noted that the morphology of the metacarpal—characterized by a low distal articulation surface, weak trochlear ridges, and an inflated distal diaphysis—is typical of species inhabiting highly humid forests (metacarpal type A2 in [108]). The CN material shows that *Parabos tigneresi* does not significantly differ in general morphology and proportions from *P. cordieri* with the exception of the larger size. Phalanges are extremely informative elements for the type of habitat [108,117]. The proximal phalanx of the bovid from CN presents a rough surface on its axial side and its distal articulation is visible in anterior (=dorsal) view. The intermediate phalanx has a deep sagittal groove in its posterior (=palmar) side, there is an elongated post-articular plateau on the proximal section and the axial outline of the distal articulation is subtriangular. Lastly, the distal phalanx does not have a process above (i.e., anteriorly) the articulation and the latter is elevated from the ground level due to a conspicuous bone growth below (i.e., posteriorly) to it. All these characters fit with the morphology expected for bovids adapted to densely covered habitats [108,117]. Therefore, it is likely that both *P. cordieri* and *P. tigneresi* inhabited forested, relatively humid ecosystems. The analyses of Fig 22 show interesting results regarding other large Bovids of European Plio-Pleistocene. The limb lengths of *Alephis* are almost identical to *Parabos* suggesting that this taxon did not undergo significant specialization in its postcranial skeleton during the late Ruscinian (Fig 22). Notably, also *Leptobos*, despite being characterized by more derived dentognathic traits, exhibits limb bone proportions which are divergent from the variability observed in younger fossil and extant Bovini, especially in the metacarpal, positioning in between the groups formed by these Bovini and the *Parabos-Alephis* cluster (Fig 22B). This suggests that, at least during the earliest stages of the Quaternary, European Bovini did not yet display a significant shift toward stouter builds or gigantic body size; a transformation that occurred much later with the arrival of the first *Bison*.

Body mass estimates calculated on the basis of 11 dental and postcranial measures (S31 Table) for nine individuals from CN yields an average weight of 419±31 kg for *Parabos tigneresi*, which makes it the largest bovid in Europe prior to the end Pliocene (Fig 26; S32 Table). The individuals with the smallest horncores are also the lightest with an estimated mass of 378±60 kg (IPHES.CN'19-B15) and 397±33 kg (IPHES.CN'04-B1) and therefore were interpreted as females. The rest of the sample, due to the size of the horns and postcranial, was referred to male individuals. In particular, the individuals bearing the largest horns, were estimated at 420±37 kg (IPHES.CN'05-B2), 456±55 (IPHES.CN'12-B10) and 408±80 kg (IPHES.CN'17-B14). The difference between the minimum and maximum body mass estimations is approximately 100 kg (min=378±60 kg; max=481±51 kg) with the lightest specimen weighing slightly 20% less than the

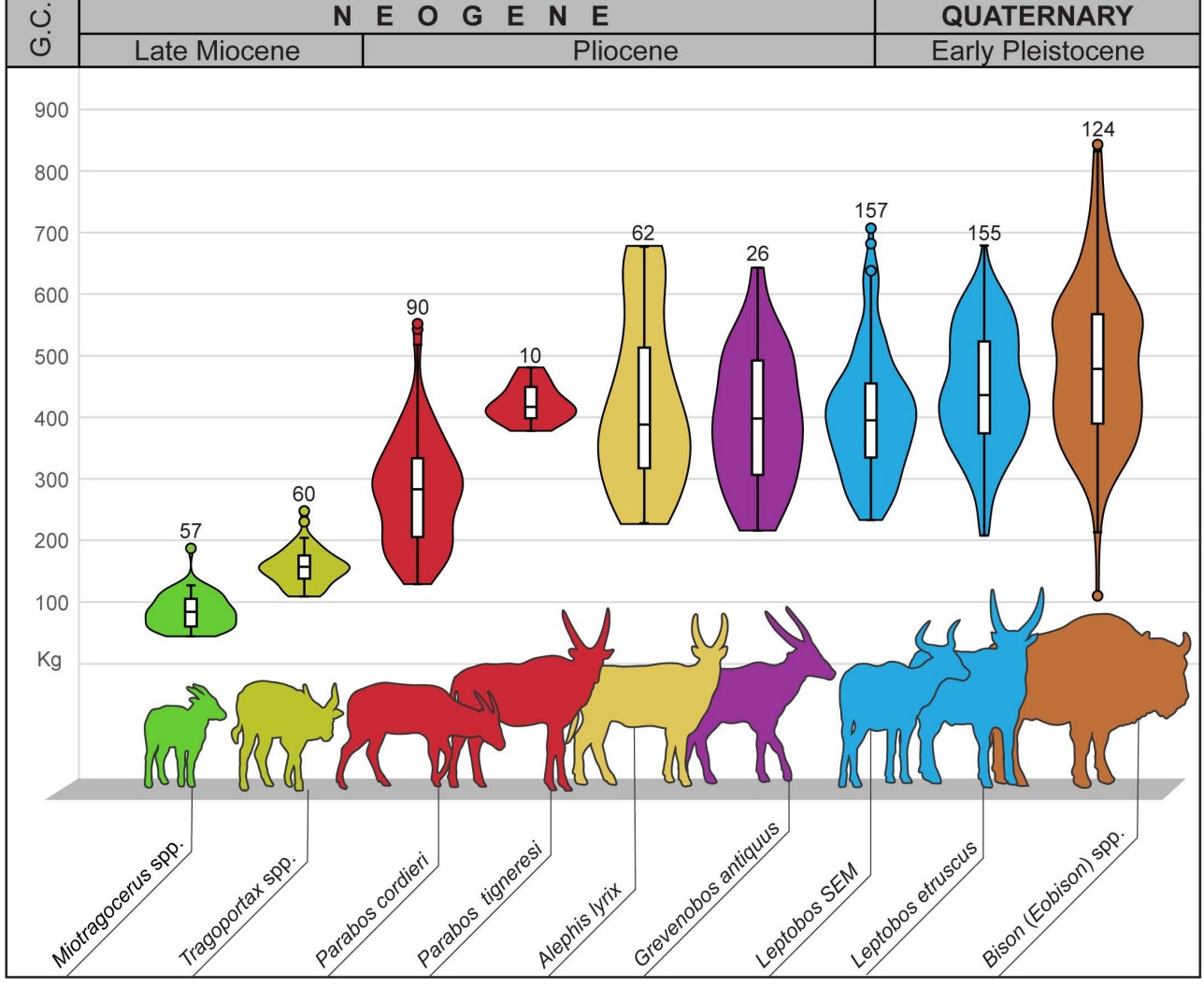

**Fig 26. Violin plot of estimated body mass for selected large-bodied late Neogene and early Quaternary Bovidae from Europe.** Number of specimens shown above each sample. See S32 Table for specimens, localities and references.

heaviest. In extant large bovins, females are typically smaller than males, often averaging 60–70% of male mass [118]. Therefore, it appears that sexual body mass dimorphism in *P. tigneresi* was not as pronounced as in extant bovins. Rather this is more similar to the condition in Alcelaphini and Hippotragini, in which males and females show larger mass overlap. This relative monomorphism is well-reflected in the low degree of horn size variation within the CN sample (see Figs 6–8; Table 1). An example of this can be found in IPHES.CN'11-B4 which, despite being the heaviest individual, and tentatively referred to a male, does not possess large horns. According to Packer [119], large-bodied animals rely less on rapid flight or crypsis as primary anti-predator strategies and instead tend to develop prominent cornual appendages as a defensive adaptation, especially in open environments where concealment within vegetation is less viable. Consequently, females of large-sized bovids living in such habitats are more likely to possess horns [120]. Despite its large size, the CN bovid was likely living in forested habitats and not in open landscapes, partially deviating from this predictive model. A comparable case is observed in another large Bovidae, the bongo (*Tragelaphus eurycerus*), where females possess impressive horns

despite inhabiting dense forests [120]. This trait has been linked to strong intraspecific social competition; an explanation that may also apply to *P. tigneresi*, considering also its inferred ecological and morphological similarities to *T. eurycerus*. On the other hand, Estes [121] proposed that horns in female bovids may evolve not for defence or competition per se, but as a form of andromimicry (i.e., mimicking male traits) to shield male offspring from despotic aggression by dominant males, thus enhancing their survival. It must be remembered, however, that phylogenetic constraints play a significant role in horn development in females [120], thus the ancestry of *Parabos* may have influenced the retention of large horns in females more than natural selection. It is indeed true that also in *P. cordieri* the sexual dimorphism is quite reduced as stated by Duvernois [56] who found difficult distinguishing male and female metapodials. This condition, however, is not common in any other European Tragoportacini or Pliocene Bovini, as in *Miotragocerus* and *Tragoportax* the females are either hornless or showing clearly different morphology and size in the horns, and in *Leptobos* the females are noticeably smaller than the males and hornless [38,56,57,122].

The body mass of *A. lyrix* was estimated to be similar to that of *P. tigneresi*, i.e., in the 300–500 kg range (422±31 kg; Fig 26; S32 Table). Among extant bovids, only the bongo and small ecomorphs of the African buffalo (e.g., *Syncerus caffer 'nanus'* and '*brachyceros*') are in this size range. While smaller than most extant Bovini (>500 kg), *P. tigneresi* is still larger than *Parabos cordieri* (281 ± 89 kg), *Miotragocerus* (85 ± 28 kg) and *Tragoportax* (156 ± 29 kg) (Fig 26; Table S32). Except for the caprine-like antelopes *Protoryx* and *Palaeoryx*, the largest bovid of Europe during this period was the obscure and relatively rare *Samokeros minotaurus*. The clear trend toward large body size (exceeding 400 kg) among multiple lineages of European Bovinae emerged only at the beginning of Pliocene with *Parabos tigneresi*, *Alephis lyrix* (421 ± 135 kg), and *Grevenobos antiquus* (398 ± 107 kg), and continued into the Pleistocene with *Leptobos* ex gr. SEM (404 ± 99 kg), *Leptobos* ex gr. EV (446 ± 98 kg) and *Bison* (*Eobison*) (489 ± 129 kg; Fig 26; S32 Table). The underlying causes of the shift toward larger sizes in European Bovinae remain unclear, though they might be linked, as it is possible for the increased hypsodonty, to the long-term cooling and aridification trend that affected the Eurasian continent culminating at the Quaternary and eventually leading to the replacement of forested, subtropical ecosystems by drier and more open habitats [123, 124, 125 and references therein] The relationship between habitat shifts and body mass fluctuations in herbivores has been widely studied showing that animals' size is indirectly influenced by climate changes through vegetation [e.g., 126, 127]. The Jarman–Bell principle predicts larger body sizes in herbivores allow greater digestive efficiency when consuming coarse, low-nutrient vegetation such as grasses [128,129]. Global cooling and the spread of grasslands during the Neogene-Quaternary could have driven increases in body mass [130]. Such a hypothesis predicts increasing proportions of grasses in the diets of bovids from the Miocene onwards, as is also suggested by the higher hypsodonty index values. The diet of the CN *Parabos tigneresi*, however, is not known, and it remains to be seen if it consumed any appreciable quantity of grass.

## Conclusion

The remarkable skeletons of *Parabos tigneresi* from Camp dels Ninots (CN) allow us to reassess the taxonomy and evolutionary relationships of the enigmatic Bovinae genera *Parabos* and *Alephis*. Historically hindered by poorly preserved and inadequately described material, these taxa of large bovids have long remained taxonomically problematic. We recognize *Parabos cordieri*, *P. soriae*, *P. savelisi*, *P. tigneresi*, and *Alephis lyrix* as valid and provide a new, updated diagnosis for *Parabos* and *P. tigneresi*. While agreeing with previous workers that *Alephis* and *Parabos* are likely closely related, we recognize the need for focused phylogenetic analysis to help determine their placement, as either a late-surviving lineage of Miocene 'boselaphins' (i.e., Tragoportacini), or stem Bovini that persisted in Europe during the Early Pliocene while crown taxa had likely already appeared in Africa and Asia.

Palaeobotanical and palynological data from CN indicate that *P. tigneresi* inhabited a relatively humid, vegetated habitat (Fig 27). While its limb proportions are unspecialized, some discrete features of phalanges and distal metapodials suggest it might have inhabited environment with soft, moist soil, as expected in a water-rich environment such as in Camp dels

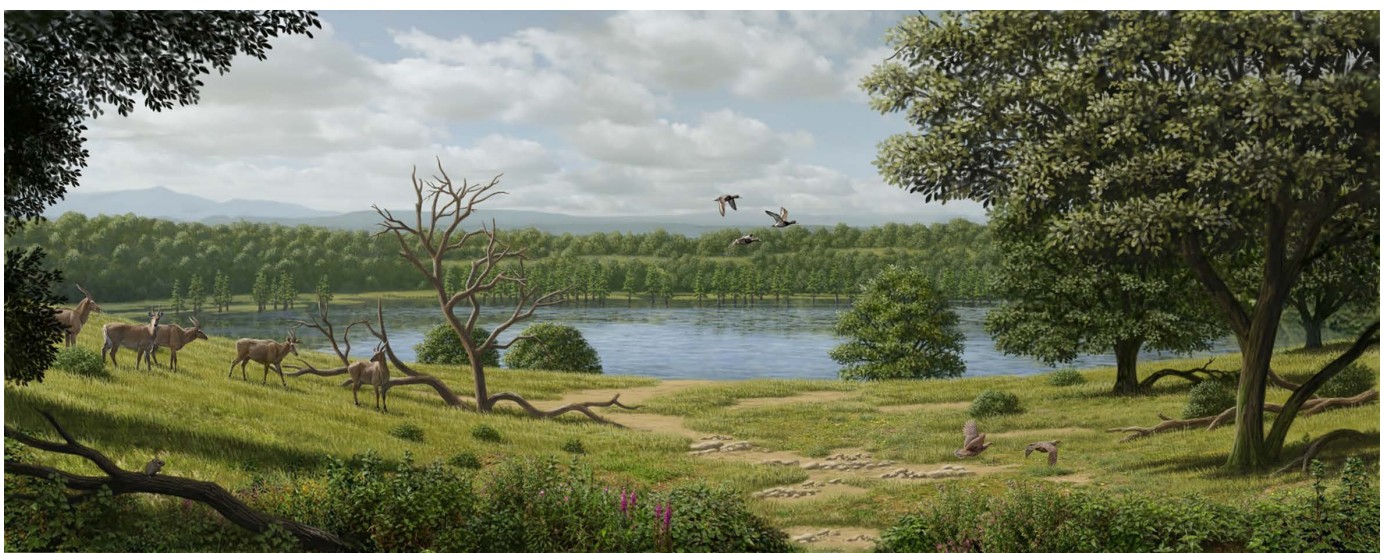

**Fig 27. Palaeoartistic reconstruction of the environment in the surrounding of Camp dels Ninots maar lake during the early Pliocene.** Artwork by Mauricio Antón.

Ninots. Conversely, its large body size and derived dental characters suggest an increased intake of grasses, pointing to potential adaptations to mixed or transitional ecosystems.

Camp dels Ninots thus offers valuable insights into Early Pliocene ecosystems and bovid evolution. Further research into the ecomorphology and diet of *P. tigneresi* will help clarify whether its traits reflect local adaptation or broader evolutionary patterns within Bovinae.

## Supporting information

**S1 Table. Complete list of *Parabos tigneresi* specimens from Camp dels Ninots referred to *Parabos tigneresi* studied in this work.**
(XLSX)

**S2 Table. Measurements (mm) of the atlas of *Parabos tigneresi* from Camp dels Ninots. Estimated measurements are in italics.**
(XLSX)

**S3 Table. Measurements (mm) of the axis of *Parabos tigneresi* from Camp dels Ninots. Estimated measurements are in italics.**
(XLSX)

**S4 Table. Measurements (mm) of the cervical vertebrae (III to VII) of *Parabos tigneresi* from Camp dels Ninots. Estimated measurements are in italics.**
(XLSX)

**S5 Table. Measurements (mm) of the thoracic vertebrae of *Parabos tigneresi* from Camp dels Ninots. Estimated measurements are in italics.**
(XLSX)

**S6 Table. Measurements (mm) of the lumbar vertebrae (III to VII) of *Parabos tigneresi* from Camp dels Ninots. Estimated measurements are in italics.**
(XLSX)

**S7 Table. Measurements (mm) of the scapula of *Parabos tigneresi* from Camp dels Ninots. Estimated measurements are in italics.**
(XLSX)

**S8 Table. Measurements (mm) of the humerus of *Parabos tigneresi* from Camp dels Ninots. Estimated measurements are in italics.**
(XLSX)

**S9 Table. Measurements (mm) of the radius of *Parabos tigneresi* from Camp dels Ninots. Estimated measurements are in italics.**
(XLSX)

**S10 Table. Measurements (mm) of the ulna of *Parabos tigneresi* from Camp dels Ninots. Estimated measurements are in italics.**
(XLSX)

**S11 Table. Measurements (mm) of the magnum of *Parabos tigneresi* from Camp dels Ninots. Estimated measurements are in italics.**
(XLSX)

**S12 Table. Measurements (mm) of the pisiform of *Parabos tigneresi* from Camp dels Ninots. Estimated measurements are in italics.**
(XLSX)

**S13 Table. Measurements (mm) of the pyramidal of *Parabos tigneresi* from Camp dels Ninots. Estimated measurements are in italics.**
(XLSX)

**S14 Table. Measurements (mm) of the scaphoid of *Parabos tigneresi* from Camp dels Ninots. Estimated measurements are in italics.**
(XLSX)

**S15 Table. Measurements (mm) of the semilunar of *Parabos tigneresi* from Camp dels Ninots. Estimated measurements are in italics.**
(XLSX)

**S16 Table. Measurements (mm) of the unciform of *Parabos tigneresi* from Camp dels Ninots. Estimated measurements are in italics.**
(XLSX)

**S17 Table. Measurements (mm) of the metacarpal of *Parabos tigneresi* from Camp dels Ninots. Estimated measurements are in italics.**
(XLSX)

**S18 Table. Measurements (mm) of the hemipelvis of *Parabos tigneresi* from Camp dels Ninots. Estimated measurements are in italics.**
(XLSX)

**S19 Table. Measurements (mm) of the femur of *Parabos tigneresi* from Camp dels Ninots. Estimated measurements are in italics.**
(XLSX)

**S20 Table. Measurements (mm) of the patella of *Parabos tigneresi* from Camp dels Ninots. Estimated measurements are in italics.**
(XLSX)

**S21 Table. Measurements (mm) of the tibia of *Parabos tigneresi* from Camp dels Ninots. Estimated measurements are in italics.**
(XLSX)

**S22 Table. Measurements (mm) of the malleolus of *Parabos tigneresi* from Camp dels Ninots. Estimated measurements are in italics.**
(XLSX)

**S23 Table. Measurements (mm) of the astragalus of *Parabos tigneresi* from Camp dels Ninots. Estimated measurements are in italics.**
(XLSX)

**S24 Table. Measurements (mm) of the calcaneum of *Parabos tigneresi* from Camp dels Ninots. Estimated measurements are in italics.**
(XLSX)

**S25 Table. Measurements (mm) of the cubonavicular of *Parabos tigneresi* from Camp dels Ninots. Estimated measurements are in italics.**
(XLSX)

**S26 Table. Measurements (mm) of the cuneiform of *Parabos tigneresi* from Camp dels Ninots. Estimated measurements are in italics.**
(XLSX)

**S27 Table. Measurements (mm) of the metatarsal of *Parabos tigneresi* from Camp dels Ninots. Estimated measurements are in italics.**
(XLSX)

**S28 Table. Measurements (mm) of the proximal phalanx of *Parabos tigneresi* from Camp dels Ninots. Estimated measurements are in italics.**
(XLSX)

**S29 Table. Measurements (mm) of the intermediate phalanx of *Parabos tigneresi* from Camp dels Ninots. Estimated measurements are in italics.**
(XLSX)

**S30 Table. Measurements (mm) of the distal phalanx of *Parabos tigneresi* from Camp dels Ninots. Estimated measurements are in italics.**
(XLSX)

**S31 Table. Body mass estimation equations used in this paper.**
(XLSX)

**S32 Table. Body mass estimations (kg) computed for several taxa of Bovidae from the Miocene to the Pleistocene of Europe.**
(XLSX)

**S33 Table. Measurements (mm) of the horncores diameters at the base (anteroposterior HAPD and transversal HTD) of various Bovidae from the Miocene to the Pleistocene of Eurasia and Africa.** Abbreviations: HAPD, horncore anteroposterior diameter; HTD, horncore transversal diameter.
(XLSX)

**S34 Table. Measurements (mm) of the lower tooth row lengths (molar, premolar, total) of various Bovidae from the Miocene to the Pleistocene of Eurasia and Africa.**
(XLSX)

**S35 Table. Measurements (mm) of the supraoccipital of various Bovidae from the Miocene to the Pleistocene of Eurasia and Africa.** Abbreviations: SOL, supraoccipital length (anteroposterior diameter); SOW, supraoccipital width (transversal diameter).
(XLSX)

**S36 Table. Measurements (mm) of the basioccipital tuberosities transversal diameters (=widths) of various Bovidae from the Miocene to the Pleistocene of Eurasia and Africa.** Abbreviations: BATW, basioccipital anterior tuberosities width; BPTW, basioccipital posterior tuberosities width.
(XLSX)

**S37 Table. Measurements (mm) of the upper teeth (P4, M1, M3) of various Bovidae from the Miocene to the Pleistocene of Eurasia and Africa.**
(XLSX)

**S38 Table. Measurements (mm) of the lower teeth (p3, p4, m3) of various Bovidae from the Miocene to the Pleistocene of Eurasia and Africa.**
(XLSX)

## Acknowledgments

We are deeply grateful to Prof. Jan van der Made for his pioneering studies on these specimens, which laid the groundwork for our current research. We also extend our sincere thanks to the landowners for granting us permission to conduct fieldwork in the area. The authors wish to thank the curator technicians of the IPHES Núria Ibáñez López and Laura Hernando Folch for the help in the collections of the IPHES where the specimens from CN are housed. We acknowledge the support of the medical staff at the Hospital de Sant Pau i Santa Tecla (Tarragona, Spain) in the use of their facilities during the scanning of the CN specimens. We are grateful to Maria Dolors Guillén Espínola (IPHES-CERCA) for the pictures taken of the specimens in Figs 11–15. No permits were required for the described study, which complied with all relevant regulations.

The authors thank the CERCA Programme/Generalitat de Catalunya. IPHES-CERCA has received grant funding through the "María de Maeztu" program for Units of Excellence (CEX2024–01485-M, funded by MICIU/AEI/10.13039/501100011033), which supported FG, EMR, GC, and BGS.

We thank the institutions that allowed us to study the comparative material used in this work including: Museum of Geology, Palaeontology and Palaeoanthropology of the Aristotle University of Thessaloniki (LGPUT, Greece), Museo di Storia Naturale, Sezione di Geologia e Paleontologia, Università di Firenze (IGF, Italy), Museum für Naturkunde, Berlin (MfN, Germany). L.S. wants to thank Dimitris Kostopoulos and George Konidaris for the interesting discussions regarding

the evolution of Miocene bovids of Europe and their hospitality at LGPUT during his research stay. L.S. thanks also Elpiniki Maria Parparousi for the help provided during the data collection at LGPUT. The authors thank the review of Dimitris Kostopoulos which greatly improved the first version of the manuscript.

## Author contributions

**Conceptualization:** Leonardo Sorbelli, Faysal Bibi, Joan Madurell-Malapeira, Gerard Campeny, Bruno Gómez de Soler.

**Data curation:** Leonardo Sorbelli, Faysal Bibi, Joan Madurell-Malapeira, Federica Grandi, Elena Moreno-Ribas, Oriol Oms, Gerard Campeny, Bruno Gómez de Soler.

**Formal analysis:** Leonardo Sorbelli, Faysal Bibi.

**Funding acquisition:** Gerard Campeny, Bruno Gómez de Soler.

**Investigation:** Leonardo Sorbelli, Faysal Bibi, Joan Madurell-Malapeira.

**Methodology:** Leonardo Sorbelli, Faysal Bibi, Joan Madurell-Malapeira.

**Project administration:** Joan Madurell-Malapeira, Gerard Campeny, Bruno Gómez de Soler.

**Resources:** Gerard Campeny, Bruno Gómez de Soler.

**Supervision:** Faysal Bibi, Joan Madurell-Malapeira, Gerard Campeny, Bruno Gómez de Soler.

**Validation:** Leonardo Sorbelli, Faysal Bibi, Joan Madurell-Malapeira, Gerard Campeny, Bruno Gómez de Soler.

**Visualization:** Leonardo Sorbelli.

**Writing – original draft:** Leonardo Sorbelli.

**Writing – review & editing:** Leonardo Sorbelli, Faysal Bibi, Joan Madurell-Malapeira, Federica Grandi, Elena Moreno-Ribas, Oriol Oms, Gerard Campeny, Bruno Gómez de Soler.

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
