## [Decision Letter · Decision Letter 0]

28 Oct 2025

PONE-D-25-47590First of a Line or Last of a Dynasty? Parabos tigneresi and the Evolution of Eurasian Bovinae in the Early PliocenePLOS ONE?

Dear Dr. Sorbelli,

Thank you for submitting your manuscript to PLOS ONE. After careful consideration, we feel that it has merit but does not fully meet PLOS ONE’s publication criteria as it currently stands. Therefore, we invite you to submit a revised version of the manuscript that addresses the points raised during the review process.

The submitted manuscript represents a high-quality and valuable contribution to the understanding of Early Pliocene Bovinae in Europe, based on the remarkable material from Camp dels Ninots. It is a well-structured, rigorous, and clearly written study that significantly enriches current knowledge of the paleobiodiversity of this period and region.

The editorial process took longer than usual, mainly due to the difficulty in finding suitable and available reviewers with expertise in this specific field. Nevertheless, the received evaluation confirms the scientific merit of the work and highlights the strength of the analyses, the precision of the illustrations, and the clarity of the descriptions.

The reviewer emphasized the overall consistency and scientific rigor of the manuscript, as well as the up-to-date bibliography and the coherence between results and conclusions. Only minor adjustments and clarifications were suggested, as indicated in the annotated PDF. These mainly concern certain paleoenvironmental and biogeographical interpretations, which could be presented in a slightly more cautious manner—preferably framed as working hypotheses for future research rather than as definitive statements.

Recommendations:

1- Revise the discussion and conclusions to express the paleoenvironmental and biogeographical interpretations in a more tentative or exploratory tone.

2- Address the specific corrections and comments noted in the annotated PDF.

3- Perform a final language and consistency check to ensure clarity and uniform terminology throughout the manuscript.

We look forward to receiving your revised manuscript.

Kind regards,

Wesley D. Colombo

Academic Editor

PLOS ONE

Journal Requirements:

2. In your manuscript, please provide additional information regarding the specimens used in your study. Ensure that you have reported human remain specimen numbers and complete repository information, including museum name and geographic location.

For more information on PLOS One's requirements for paleontology and archeology research, see https://journals.plos.org/plosone/s/submission-guidelines#loc-paleontology-and-archaeology-research .

“This research has received funding from the Catalan Government (Generalitat de Catalunya) through the Departament de Cultura (project CLT009/22/000043) and the research group 2021 SGR 01238 (AGAUR), as well as from the Universitat Rovira i Virgili (grant 2023-URV-01238). Additional funding is provided by the Ministry of Science and Innovation of the Spanish Government through project PID2021-123092NB-C21. L.S. was supported by a Humboldt Research Fellowship provided by the Alexander von Humboldt Foundation and by the CERCA Programme of the Generalitat de Catalunya. The research conducted by F.G., E.M., G.C., and B.G.S. is funded by the CERCA Programme of the Generalitat de Catalunya.”

“The CN project is supported by the Caldes de Malavella Town Hall. This research has received funding from the Catalan Government (Generalitat de Catalunya) through the Departament de Cultura (project CLT009/22/000043) and the research group 2021 SGR 01238 (AGAUR), as well as from the Universitat Rovira i Virgili (grant 2023-URV-01238). Additional funding is provided by the Ministry of Science and Innovation of the Spanish Government through project PID2021-123092NB-C21. L.S. was supported by a Humboldt Research Fellowship provided by the Alexander von Humboldt Foundation and by the CERCA Programme of the Generalitat de Catalunya. The research conducted by F.G., E.M., G.C., and B.G.S. is funded by the CERCA Programme of the Generalitat de Catalunya.”

“This research has received funding from the Catalan Government (Generalitat de Catalunya) through the Departament de Cultura (project CLT009/22/000043) and the research group 2021 SGR 01238 (AGAUR), as well as from the Universitat Rovira i Virgili (grant 2023-URV-01238). Additional funding is provided by the Ministry of Science and Innovation of the Spanish Government through project PID2021-123092NB-C21. L.S. was supported by a Humboldt Research Fellowship provided by the Alexander von Humboldt Foundation and by the CERCA Programme of the Generalitat de Catalunya. The research conducted by F.G., E.M., G.C., and B.G.S. is funded by the CERCA Programme of the Generalitat de Catalunya.”

5. We note that Figure 1 and 2 in your submission contain [map/satellite] images which may be copyrighted. All PLOS content is published under the Creative Commons Attribution License (CC BY 4.0), which means that the manuscript, images, and Supporting Information files will be freely available online, and any third party is permitted to access, download, copy, distribute, and use these materials in any way, even commercially, with proper attribution. For these reasons, we cannot publish previously copyrighted maps or satellite images created using proprietary data, such as Google software (Google Maps, Street View, and Earth). For more information, see our copyright guidelines: http://journals.plos.org/plosone/s/licenses-and-copyright.

1. You may seek permission from the original copyright holder of Figure 1 and 2 to publish the content specifically under the CC BY 4.0 license.

Reviewers' comments:

Reviewer's Responses to Questions

**Comments to the Author**

1. Is the manuscript technically sound, and do the data support the conclusions?

Reviewer #1: Yes

2. Has the statistical analysis been performed appropriately and rigorously?

Reviewer #1: Yes

3. Have the authors made all data underlying the findings in their manuscript fully available?

Reviewer #1: Yes

4. Is the manuscript presented in an intelligible fashion and written in standard English?

Reviewer #1: Yes

Reviewer #1: A truly excellent and important piece of work that provides valuable information about the relatively unknown Bovinae of the Early Pliocene in Europe based on the impressive material from Camp dels Ninots. The descriptions are detailed and accurate, the comparisons comprehensive and thorough, the diagrams and visual material impeccable, as usual in works signed by the first author, the bibliography is fully up to date, and the conclusions are interesting and in complete harmony with the results. I have very few comments/corrections and suggestions to make (see annotated pdf). Some of the environmental/biozoogeographical interpretations may be somewhat exaggerated (see comments in pdf) and would be better formulated as working hypotheses for future studies.

**Do you want your identity to be public for this peer review?** For information about this choice, including consent withdrawal, please see our Privacy Policy

Reviewer #1: **Yes:** Dimitris S Kostopoulos

---

## [Author Response · Author response to Decision Letter 1]

9 Dec 2025

Dear PLOS ONE Editors and reviewer,

We thank you and reviewer D. S. Kostopoulos for the thorough evaluation of our submitted manuscript. We appreciate that the paper has been well received and are grateful for the reviewer’s particularly kind and constructive comments.

We have revised the main text, supporting information, and figures to address all requested changes and requirements for publication. Included with this resubmission are the three required files: (1) a marked-up copy of the revised manuscript (“Revised Manuscript with Track Changes”), (2) a clean version of the revised manuscript (“Manuscript”), and (3) the rebuttal document containing our detailed responses directly on the commented PDF provided by the reviewer (“Response to Reviewers”). As shown in both the rebuttal and the marked-up manuscript, all changes suggested by the reviewer have been fully implemented.

Below, we address each point raised by the editor in the journal’s requirements section:

1. Please ensure that your manuscript meets PLOS ONE's style requirements […]

Completed.

2. In your manuscript, please provide additional information regarding the specimens used in your study […]

No additional information was required.

In the acknowledgments we added the phrase: “No permits were required for the described study, which complied with all relevant regulations.”.

If this sentence should be placed elsewhere in the manuscript, please let us know.

3. Thank you for stating the following financial disclosure […]

The founders had no role in study design, data collection and analysis, decision to publish, or preparation of the manuscript.

4. Thank you for stating the following in the Acknowledgments Section of your manuscript […]

The funding-related text has been removed from the acknowledgments section.

Please implement the following amended version of the funding acknowledgments:

“The CN project is supported by the Caldes de Malavella Town Hall. This research has received funding from the Catalan Government (Generalitat de Catalunya) through the Departament de Cultura (project CLT009/22/000043) and the research group 2021 SGR 01238 (AGAUR), as well as from the Universitat Rovira i Virgili (grant 2023-URV-01238). Additional funding is provided by the Ministry of Science and Innovation of the Spanish Government through project PID2024-157622NB-I00. L.S. was supported by a Humboldt Research Fellowship provided by the Alexander von Humboldt Foundation and by the CERCA Programme of the Generalitat de Catalunya. The research conducted by F.G., E.M., G.C., and B.G.S. is funded by the CERCA Programme of the Generalitat de Catalunya. The IPHES-CERCA has received financial support through the “María de Maeztu” program for Units of Excellence (CEX2024-01485-M/funded by MICIU/AEI/10.13039/501100011033).”.

5. We note that Figure 1 and 2 in your submission contain [map/satellite] images which may be copyrighted […]

Both figures 1 and 2 have been modified accordingly. The map in Figure 1 has been redrawn by us using original data in our possession, and the map in Figure 2 has been produced using data from one of the recommended platforms (http://www.naturalearthdata.com/)

6. Please include captions for your Supporting Information files at the end of your manuscript […]

Completed.

7. If the reviewer comments include a recommendation to cite specific previously published works […]

No suggested additional references were given by the reviewer

8. Please review your reference list to ensure that it is complete and correct […]

The reference list was checked to ensure that is correct. We note one issue regarding the reference for the new dating of the Camp dels Ninots site. The study establishing its Early Pliocene age is still under review. For the moment, we have kept the citation as “(in prep.)”. We are unsure when it will be published, but the results supporting the age are robust and consistent with the site’s biochronological framework. Please advise if an alternative approach is required.

Sincerely,

Leonardo Sorbelli

on behalf of all co-authors

---

## [Editor Report · Decision Letter 1]

18 Dec 2025

First of a Line or Last of a Dynasty? Parabos tigneresi and the Evolution of Eurasian Bovinae in the Early Pliocene

PONE-D-25-47590R1

Dear Dr. Sorbelli,

We’re pleased to inform you that your manuscript has been judged scientifically suitable for publication and will be formally accepted for publication once it meets all outstanding technical requirements.

Kind regards,

Wesley D. Colombo

Academic Editor

PLOS One
---

## [Editor Report · Acceptance letter]

PONE-D-25-47590R1

PLOS One

Dear Dr. Sorbelli,

I'm pleased to inform you that your manuscript has been deemed suitable for publication in PLOS One. Congratulations! Your manuscript is now being handed over to our production team.

Kind regards,

on behalf of

Dr. Wesley D. Colombo

Academic Editor

PLOS One